# Modeling surface water dynamics in the Amazon Basin using MOSART-Inundation-v1.0: Impacts of geomorphological parameters and river flow representation

Xiangyu Luo[1], Hong-Yi Li[1,*], L. Ruby Leung[1], Teklu K. Tesfa[1], Augusto Getirana[2], Fabrice Papa[3,4], Laura L. Hess[5]

[1] Pacific Northwest National Laboratory, Richland, Washington 99352, United States
[2] NASA Goddard Space Flight Center, Greenbelt, Maryland 20771, United States
[3] LEGOS/IRD, Universite de Toulouse, IRD-CNRS-CNES-UPS, Toulouse 31400, France
[4] Indo-French Cell for Water Sciences, IISc-NIO-IITM–IRD Joint International Laboratory, IISc, Bangalore, India
[5] University of California, Santa Barbara, California 93106, United States
[*] Now at Montana State University, Bozeman, Montana 59717, United States

*Correspondence to*: L. Ruby Leung (Ruby.Leung@pnnl.gov)

## Abstract

In the Amazon Basin, floodplain inundation is a key component of surface water dynamics and plays an important role in water, energy and carbon cycles. The Model for Scale Adaptive River Transport (MOSART) was extended with a macro-scale inundation scheme to represent floodplain inundation. The extended model, named "MOSART-Inundation", was used to simulate surface hydrology of the entire Amazon Basin. Previous hydrologic modeling studies in the Amazon Basin identified and addressed a few challenges in simulating surface hydrology of this basin, including uncertainties of floodplain topography and channel geometry, and the representation of river flow in reaches with mild slopes. This study further addressed four aspects of these challenges. First, the spatial variability of vegetation-caused biases embedded in the HydroSHEDS DEM data was explicitly addressed. A vegetation height map of about 1-km resolution and a land cover dataset of about 90-m resolution were used in a DEM correction procedure that resulted in an average elevation reduction of 13.2 m for the entire basin and led to evident changes in the floodplain topography. Second, basin-wide empirical formulae for channel cross-sectional dimensions were refined for various subregions to improve the representation of spatial variability in channel geometry. Third, the channel Manning roughness coefficient was allowed to vary with the channel depth, as the effect of riverbed resistance on river flow generally declines with increasing river size. Lastly, backwater effects were accounted for to better represent river flow in mild-slope reaches. The model was evaluated against in situ streamflow records and remotely sensed Envisat altimetry data

and GIEMS inundation data. In a sensitivity study, seven simulations were compared to evaluate the impacts of the four modeling aspects addressed in this study. The comparisons showed that representing floodplain inundation could significantly improve the simulated streamflow and river stages. Refining floodplain topography, channel geometry and Manning roughness coefficients, as well as accounting for backwater effects had notable impacts on the simulated surface water dynamics in Amazonia. The understanding obtained in this study could be helpful in improving modeling of surface hydrology in basins with evident inundation, especially at regional to continental scales.

**Keywords**

surface hydrology, hydrologic modeling, floodplain inundation, Amazon Basin, geomorphology, DEM biases, channel cross-sectional geometry, Manning roughness coefficient, backwater effects

# 1 Introduction

The terrestrial surface water dynamics have significant impacts on the water, energy and carbon cycles of the planet, as they influence energy and material exchanges between the land surface and the atmosphere. For instance, surface water bodies are important natural sources of greenhouse gases (e.g., carbon dioxide and methane) (Bousquet et al., 2006; Richey et al., 2002). Extreme events such as river inundation have extraordinary effects on land surface – groundwater interactions and sediment and nutrient exchanges between rivers and floodplains, and thereby influence land and aquatic ecosystems as well as their feedback to the atmosphere. Therefore, improving parameterizations of surface water dynamics is meaningful in studying the land – climate linkage.

Many previous studies of surface-hydrology modeling were conducted for the Amazon River, which is the largest river of the globe and accounts for about 18% of the total continental freshwater discharge to oceans (Dai and Trenberth, 2002). Seasonal floods occur every year and wetlands occupy a considerable fraction of the total area in the basin (Hess et al., 2003, 2015). River and inundation dynamics were simulated by using 2-D hydrodynamic models at the central Amazonia (e.g., Baugh et al., 2013; Wilson et al., 2007). Using fine-resolution grid cells (e.g., ~ 300 m) as computation units, 2-D hydrodynamic models could represent water flow over floodplains. They were not applied at regional or larger scales due to computational cost. On the other hand, some computationally efficient macro-scale inundation schemes were used in a few continental-scale hydrologic models for the entire Amazon Basin (Coe et al., 2008; Decharme et al., 2008; Getirana et al., 2012; Paiva et al., 2013a; Vörösmarty et al., 1989; Yamazaki et al., 2011), which could capture some aspects of surface water dynamics fairly well. These previous studies also identified and addressed a number of modeling challenges, including uncertainties in model inputs of floodplain and channel morphology, flow parameterization for gentle-gradient reaches, etc.

The Model for Scale Adaptive River Transport (MOSART) was developed to simulate terrestrial surface water flow from hillslopes to the basin outlet (Li et al., 2013). It was designed to be applicable at the local, regional or continental scale. Some details of this model are provided in Sect. 2.1. In this study, the MOSART model was extended with a macro-scale inundation scheme to represent floodplain inundation. The extended model, named "MOSART-Inundation", was applied to the entire Amazon Basin. In addition, some efforts were made to further address four aspects of the aforementioned challenges: (1) while alleviating the vegetation-caused biases embedded in the DEM data, we explicitly considered the spatial variability of those biases; (2) the

approach for estimating channel cross-sectional dimensions was refined to improve its representation of the spatial variability in channel geometry; (3) the Manning roughness coefficient of the channel was allowed to vary with the channel depth; and (4) backwater effects were accounted for to better represent river flow in gentle-gradient reaches.

Topography data are essential inputs in hydrologic modeling. At present the common practice is to use the digital elevation model (DEM) to represent topography. Because the coverage of high-accuracy DEM data (e.g., with elevation errors less than 1 m) is limited, hydrologic modeling at regional or larger scales uses DEM data obtained by spaceborne sensors. The Shuttle Radar Topography Mission (SRTM) DEM data have been widely used for hydrologic modeling, but some factors limit their accuracy. In forested regions such as the Amazon
Basin, primary biases in the SRTM DEM data were caused by vegetation cover because the radar signal was not able to penetrate the vegetation canopy (Sanders, 2007). Previous studies in the Amazon Basin adopted various approaches to alleviate the vegetation-caused biases embedded in the SRTM data. In some modeling studies, elevation values were lowered by a constant in forested areas of the entire basin, so the spatial variability of vegetation heights was ignored (Coe et al., 2008; Paiva et al., 2013a). In a few hydrodynamic modeling studies
for the central Amazonia, the vegetation-caused biases in the SRTM elevations were derived from spatially varying vegetation heights. For example, Wilson et al. (2007) estimated the vegetation-height distribution based on their surveyed heights of various vegetation types and a map of vegetation types by Hess et al. (2003) and Baugh et al. (2013) utilized a global dataset of spatially distributed vegetation heights developed by Simard et al. (2011). These two studies estimated the vegetation-caused biases as products of spatially varying vegetation
heights and a fixed percentage. In this study, we used the HydroSHEDS DEM data derived from the SRTM data and inherited the vegetation-caused biases. To alleviate those biases, we used a method similar to that of Baugh et al. (2013). Besides the vegetation height map by Simard et al. (2011), we also used a land cover dataset for wetlands of the lowland Amazonia developed by Hess et al. (2003, 2015). A "bare-earth" DEM of the Amazonia was created and employed in the hydrologic modeling for the entire basin. To our knowledge, this was the first
time that the spatial variability of vegetation-caused biases in the DEM data was explicitly considered in hydrologic modeling for the entire Amazon Basin.

Channel cross-sectional geometry affects the channel conveyance capacity in modeling of surface water dynamics. Distributed hydrologic modeling at regional or larger scales needs cross-sectional dimensions of all the channels that constitute the river network in the study domain. Channel cross-sectional dimensions obtained from
in situ measurements are reliable, but limited to a small number of locations. Therefore channel cross-sectional

dimensions were usually estimated based on available basin characteristics by using empirical formulae. Modeling studies in the Amazon Basin employed relationships between channel geometry and streamflow statistics (Getirana et al., 2012; Yamazaki et al., 2011) or upstream drainage areas (Beighley et al., 2009; Coe et al., 2008; Paiva et al., 2013a). Those relationships are also referred to as "channel geometry formulae" in this

article. In most of the previous studies, cross-sectional dimensions of all the channels spread over the Amazon Basin were estimated by using one set of channel geometry formulae and corresponding parameters, which represent the average characteristics of the entire basin. So for different subregions of the basin, channel cross-sectional dimensions derived from the same formulae and parameters contained biases of various magnitudes. Hydrologic modeling results were demonstrated to be sensitive to channel cross-sectional dimensions and shapes

(Getirana et al., 2013; Neal et al., 2015; Paiva et al., 2013a; Yamazaki et al., 2011) so improving the representation of channel morphology could be important. In this study, the basin-wide parameters for the channel geometry formulae were refined for various subregions of the Amazon Basin based on local channel morphology information to better represent the spatial variability in channel morphology.

The Manning formula has been used for estimating flow velocities of rivers in many continental scale

hydrologic models. In this formula, the Manning roughness coefficient (also abbreviated to "Manning coefficient" hereinafter; in this article, the "Manning roughness coefficient" discussed is for river channels) is a key and sensitive parameter (Paiva et al., 2013a; Yamazaki et al., 2011) that can only be estimated empirically. In previous studies of the Amazon Basin, the Manning coefficient was determined using various approaches: (a) a constant value for the entire basin (Beighley et al., 2009; Yamazaki et al., 2011); (b) different values for different

subregions as a result of calibration using hydrographs of major rivers (Paiva et al., 2013a); (c) diverse values dependent on the channel cross-sectional dimensions that vary spatially (Getirana et al., 2012, 2013). For natural river channels, the Manning coefficient depends on many factors, including riverbed roughness, cross-sectional geometry and channel sinuosity (Arcement and Schneider, 1989). The significant variations of these factors within a basin undermine the rationale of using a uniform Manning coefficient across the entire basin or using a

few Manning coefficients for different subregions of the basin. The approaches of the aforementioned category (c) reflect the general phenomenon that the relative importance of riverbed friction in river flow becomes smaller for larger rivers, and can be used to represent the dominant spatial variability of the Manning coefficients. We adopted a method of category (c), similar to those of Decharme et al. (2010) and Getirana et al. (2012), to estimate the spatially varying Manning coefficients for different channels of the Amazon Basin.

The Amazonia is characterized by flat gradients and backwater effects are evident in river flow (Meade et al., 1991). Trigg et al. (2009) analyzed the characteristics of flood waves and conducted hydraulic modeling for reaches of the central Amazonia. They demonstrated that it was necessary to account for backwater effects and the diffusion wave method was valid for modeling the Amazon flood waves. Backwater effects were also

represented in some continental-scale models applied in the Amazon Basin. Yamazaki et al. (2011) used both the kinematic wave and diffusion wave methods to simulate river flow, with the latter capable of representing backwater effects. Paiva et al. (2013a, 2013b) used the full Saint-Venant equations (or the dynamic wave method) to represent water flow of river reaches with gentle riverbed slopes and large floodplains. These studies showed that accounting for backwater effects could evidently improve the modeling of surface water dynamics in

this basin. In this study, river flow was modeled with the diffusion wave method that could represent backwater effects. Moreover, the impacts of backwater effects on surface hydrology of the Amazon Basin were investigated through numerical experiments in a comprehensive manner.

The four factors described above could have important impacts on modeling surface hydrology in the Amazonia and were accounted for in the simulations conducted with the MOSART-Inundation model. The model

performance was evaluated against gauged streamflow data, as well as river-stage and inundation data obtained by satellites. In a sensitivity study, the roles of the following factors in hydrologic modeling for the Amazon Basin were separately examined and demonstrated: (1) representing floodplain inundation; (2) alleviating vegetation-caused biases in the DEM data; (3) refining channel cross-sectional geometry; (4) adjusting Manning roughness coefficients; and (5) representing backwater effects. The results of this study were also compared with

those of a few previous studies on modeling surface hydrology in the Amazonia.

## 2 Methods and data

### 2.1 MOSART model

In the MOSART model, each computation unit (subbasin or grid cell) has a major channel (or main channel)

and a tributary subnetwork that represents the combined equivalent transport capacity of all the tributaries within the computation unit (Li et al., 2013). Two simplified forms of the one-dimensional Saint-Venant equations (i.e., kinematic wave or diffusion wave methods) are used to represent water flow over hillslopes, in the tributary subnetwork, or in main channels. The MOSART model is driven by runoff estimates from the land surface

model. Surface runoff is treated as input of overland flow, which is represented with the kinematic wave method and enters the tributary subnetwork, while subsurface runoff directly enters the tributary subnetwork. Water flow in the tributary subnetwork is also represented with the kinematic wave method and the outflow finds its way to the main channel. Either the diffusion wave method or kinematic wave method could be used to simulate water

flow in main channels. The two methods use the same continuity equation, and differ in the momentum equation and Manning's equation.

The continuity equation is expressed as (Chow et al., 1988):

$$\frac{\partial(v \cdot y \cdot w)}{\partial x} + \frac{\partial(y \cdot w)}{\partial t} = q \tag{1}$$

where $v$ is the flow velocity [unit: m s$^{-1}$]; $y$ is the water depth in the channel [unit: m]; $w$ is the channel width [unit: m]; $x$ is the distance along the river [unit: m]; $t$ is time [unit: s]; and $q$ is the lateral inflow per unit length

of channel [unit: m$^2$ s$^{-1}$].

In the diffusion wave method, the momentum equation is expressed as (Chow et al., 1988):

$$\frac{\partial y}{\partial x} - S_0 + S_f = 0 \tag{2}$$

where $S_0$ is the riverbed slope [dimensionless] and $S_f$ is the friction slope [dimensionless], which could be positive or negative.

The Manning's equation is expressed as:

$$v = \frac{S_f}{\left|S_f\right|} n^{-1} R^{\frac{2}{3}} \left|S_f\right|^{\frac{1}{2}} \tag{3}$$

where $n$ is the Manning roughness coefficient [unit: s $\cdot$ m$^{-1/3}$] and $R$ is the hydraulic radius [unit: m].

The continuity equation, momentum equation and Manning's equation are combined to determine the flow velocity, channel water depth and friction slope. The friction slope depends on water depth variation along the channel so it is affected by the river stage of the downstream channel. This way, backwater effects are represented. One extreme phenomenon caused by backwater effects is that when the downstream river stage is

higher than the river stage of the current channel and hence $S_f$ is negative, the flow velocity from Eq. (3) is also negative, so water flows from downstream to upstream.

In the kinematic wave method, the term $\dfrac{\partial y}{\partial x}$ is neglected from the momentum equation. With this simplification, the friction slope equals the riverbed slope and backwater effects are not represented.

In this model, the equations are solved with the explicit finite difference method. Either square grid cells or irregular subbasins can be used as computation units. The Courant–Friedrichs–Lewy (CFL) condition can be used to obtain a preliminary estimate of the time-step size (Cunge et al., 1980). In order to satisfy the CFL condition, the time-step size should be reduced with decreasing computation-unit length or increasing water depth. However, the CFL condition may not be sufficient to guarantee a stable numerical simulation (e.g., Hunter et al.,

2005). In practice, the final time-step size is determined through sensitivity tests to ensure numerical stability.

## 2.2 Macro-scale inundation scheme

In this study the MOSART model was extended with a macro-scale inundation scheme and the extended model was named "MOSART-Inundation". Floodplain inundation dynamics were represented by macro-scale

inundation schemes in a few previous studies (Coe et al., 2008; Decharme et al., 2008; Getirana et al., 2012; Paiva et al., 2013a; Yamazaki et al., 2011). Those studies used relatively coarse computation units with the area magnitude ranging from 100 to 10,000 km$^2$. The main feature of their macro-scale inundation schemes was that the water level–inundated area relationship for the computation unit was used to estimate flood extent. The inundation scheme of this study is similar to those of Yamazaki et al. (2011) and Getirana et al. (2012). In this

scheme, each computation unit (a square grid or a subbasin) has a main channel and a floodplain reservoir (Fig. 1a). Flooding water can spill out of the main channel and enter the floodplain reservoir, or recede from the floodplain reservoir to the main channel. The lateral flow between adjacent computation units is restricted to the main channel, namely it is assumed that there is no water exchange between floodplains of different computation units. The water volume within each computation unit is used with an "elevation profile" (i.e., the relationship

between the water stage and the inundated fraction of a computation unit) to estimate the surface water area within the computation unit.

The brown solid line in Fig. 1b is the original elevation profile which is developed from all the elevations of the fine-resolution DEM within the computation unit. The channel area is implicitly included in the original elevation profile. Getirana et al. (2012) proposed an amended elevation profile in which the channel area was distinguished from the non-channel area. Their method was adopted in this study. It is assumed that the main channel consists of the lowest pixels of the DEM within a computation unit. So the main channel and the rest of the computation unit, including the floodplain and the upland, are represented by the lower part and the upper part of the elevation profile, respectively (Fig. 1b). The dividing point corresponds to the fraction of channel area, which is estimated as the product of the channel length derived from DEM data and the channel width calculated with empirical formulae in Sect. 2.5. The elevation of the dividing point corresponds to the channel bank top ($E_t$ in Fig. 1b). If the channel cross-sectional shape is assumed to be a rectangle, the channel part of the elevation profile changes to follow the green dashed line in Fig. 1b. The channel bed elevation $E_b$ equals the difference of the bank top elevation $E_t$ and the channel depth, which is estimated in Sect. 2.5. The channel bed could be lower than the lowest DEM pixel of the computation unit, as shown in Fig. 1b, because the DEM does not reflect the channel bed elevation.

In each time step of the simulation, the channel – floodplain water exchange in each computation unit is calculated before the channel routing computation for the entire basin. The channel – floodplain exchange is assumed to be instantaneous, namely the pixels within a computation unit are inundated in the reversed order of elevations (from lower to higher locations). Specifically, the exchange calculation is described as follows.

At the beginning of each time step, the total water volume $V_{total}$, including the channel water volume and floodplain water volume, is compared with the channel storage capacity $S_{channel}$ (i.e., the product of the channel length, width and depth):

(1) If $V_{total}$ is less than or equal to $S_{channel}$: After the exchange process, all the water remains in the channel and the floodplain is not inundated according to the assumption of instantaneous exchange. That is, the river stage does not exceed the bank top ($E_t$ in Fig. 1b). When the rectangular channel cross-section is used, the surface water area does not change with the river stage and is always equal to the channel area (Fig. 1b).

(2) If $V_{total}$ is greater than $S_{channel}$: After the exchange process, the floodplain is inundated, and the water stage over the floodplain and the water stage in the channel are level due to the assumption of instantaneous

exchange, as shown by the blue water stage line in Fig. 1b. The final water stage can be derived from $V_{total}$ and the amended elevation profile because $V_{total}$ is equal to the product of the computation unit area and the area of the polygon surrounded by the amended elevation profile, water stage line and the y-axis in Fig. 1b. The channel water volume is calculated as the product of the channel length, width and the water depth in the channel (i.e., the difference between the final water stage and the channel bed elevation $E_b$). The intersection of the final water stage and the elevation profile indicates the fraction of the surface water area which includes the channel area and the inundated floodplain area.

The amount of the channel – floodplain water exchange is revealed by the change in the channel water volume during the exchange process, and is used in the channel routing computation described in Sect. 2.1. Specifically, the exchange amount is incorporated with the outflows from the tributary channels as the lateral inflow term in the continuity equation (i.e., Eq. (1)). The lateral inflow term could be positive (the channel receives water) or negative (the channel loses water), depending on the amount and direction of the channel – floodplain exchange, and the amount of tributary-channel outflows.

**2.3 Application in the Amazon Basin**

The MOSART-Inundation model was applied to the entire Amazon Basin. The 3 arc-seconds HydroSHEDS DEM data developed by United States Geological Survey (USGS) was used in this study. The hydrologically conditioned HydroSHEDS DEM was used to generate the digital river network and subbasins. Relatively coarse resolution subbasins were adopted as MOSART-Inundation is intended for global earth system modeling, which is constrained by computational cost. The study domain of 5.89 million km$^2$ was divided into 5395 subbasins (the average area is 1091.7 km$^2$ and the standard deviation is 921.5 km$^2$), which were used as computation units (Figs. 2a and 2b). Each subbasin has a main channel and the entire river network consists of 5395 main channels (Fig. 2a). To ensure stable computation, the time-step size was determined based on the Courant-Friedrichs-Lewy condition and sensitivity tests. The time step of one minute was used for all the simulations.

In order to analyze the spatially varying characteristics of inundation results, the Amazon Basin was divided into 10 subregions (Fig. 2c). Twenty eight large tributary catchments were first delineated and then aggregated to nine tributary subregions. Initially, seven major catchments (i.e., Xingu, Tapajos, Madeira, Purus, Jurua, Japura and Negro) were selected as subregions or the major part of a subregion. Then the Upper-Solimoes catchments

were combined as one subregion, the northeast catchments were combined as another subregion, and the remaining five large catchments were incorporated into their adjacent tributary subregions. This way, nine tributary subregions were delineated. Lastly, all the small tributary catchments and the area draining directly to the mainstem were aggregated to be the tenth subregion (i.e., the mainstem subregion).

The inputs of surface and subsurface runoff, which were of 1-degree resolution, were produced by the ISBA land surface model (Getirana et al., 2014) driven by the ORE-HYBAM precipitation dataset (Guimberteau et al., 2012). The area-weighted averaging method was used to convert the grid based runoff data to subbasin based runoff data for driving the simulations of this study.

## 2.4 Vegetation-caused biases in DEM

In the previous section, it is mentioned that the digital river network and subbasins were derived from the hydrologically conditioned HydroSHEDS DEM. However, this conditioned DEM was not suitable for representing floodplain topography and generating elevation profiles. In the DEM conditioning process, the elevation values of pixels for river channels and their buffer zones were lowered by non-negligible amounts that could, for example, be larger than 20 m in the lower mainstem area of the Amazon Basin. So the channels and their adjacent areas in the conditioned DEM could hold more water than the actual counterparts, which would lead to underestimation of flood extent. Therefore, the void-filled HydroSHEDS DEM, which was not altered by the conditioning process, is more appropriate for use to generate the elevation profiles.

However, the void-filled HydroSHEDS DEM was derived from the SRTM data so it inherited the vegetation-caused biases. Before being used for producing elevation profiles, the void-filled HydroSHEDS DEM was processed to alleviate the biases caused by vegetation. The vegetation height data with ~ 1-km resolution developed by Simard et al. (2011) was used. For vegetated areas, the original void-filled DEM represented elevations of locations within the vegetation canopy. So, part of the vegetation height needed to be deducted from the original elevation. Baugh et al. (2013) found that deducting 50 – 60% of the vegetation height of the Simard et al. (2011) data from the original DEM achieved the greatest improvements to hydrodynamic model accuracy in the Amazon floodplain. A deduction ratio of 50% was used for the vegetated area in this study.

The resolution of the vegetation height data was coarser than that of the DEM data. It might not be appropriate to assume a uniform vegetation height for all the DEM pixels within the grid cell of the vegetation

height dataset. Hess et al. (2003, 2015) developed a high resolution (3 arc-seconds) land cover dataset for floodplains (or wetlands) located in the lowland Amazon Basin (i.e., areas with elevations lower than 500 m). This land cover dataset was used in our DEM correction process. In the floodplains of the lowland Amazon Basin, vegetation height removal was conducted differently for different land cover classes. For DEM pixels with forest or woodland classes, 50% of the vegetation height was deducted from the original DEM. In the high resolution land cover dataset, shrubs were defined to be less than 5 m tall (Junk et al., 2011). So for DEM pixels with shrubs, the vegetation height was determined by the vegetation height data, but with an upper limit of 5 m. After this correction, the elevations were lowered by 50% of the vegetation heights for shrub DEM pixels. For DEM pixels with other land cover classes (e.g., open water, bare soil, etc.), the elevations were not modified. For areas outside of the floodplains of the lowland basin, a uniform vegetation height was applied for all the DEM pixels within each vegetation height pixel. This approximation was not expected to have obvious effects on inundation modeling since most inundation occurred within the floodplains of the lowland basin.

The DEM correction obviously changed the topographic features in the DEM data. The average elevation deduction in each subbasin ranges from 0 to 21 m (Fig. 2d). After the DEM correction, the average elevation in each subbasin ranges from 0 to 4772 m (Fig. 2e). For all the subbasins, the ratio of the average elevation deduction to the subbasin elevation difference (i.e., the difference between the highest and lowest elevations in the subbasin) ranges from 0 to 52.9% (average: 9.2%; standard deviation: 7.1%). The average elevation profile of the Amazon Basin was generated for the original DEM and corrected DEM, respectively (Fig. 2f). At first, the normalized elevation profile was produced for each subbasin. For each DEM pixel within a subbasin, the elevation relative to the lowest pixel of the subbasin was divided by the subbasin elevation difference to give the normalized elevation, which was used to generate the normalized elevation profile. Then the normalized elevation profiles of all subbasins were averaged to give the average elevation profile of the entire basin. Figure 2f illustrates that the DEM processing evidently lowers the average elevation profile.

O'Loughlin et al. (2016) estimated the vegetation-caused biases in the SRTM DEM data based on vegetation height data, canopy density data and the distribution of five climatic zones (i.e., Tropical, Arid, Temperate, Cold and Polar). They created the first global 'Bare-Earth' high resolution (3 arc-seconds) DEM from the SRTM DEM data. They compared their method with the static correction method (i.e., estimating the vegetation-caused bias as the product of vegetation height and a fixed percentage) used by Baugh et al. (2013) and this study, and noted that the static correction method was effective but moderately worse than their method.

## 2.5 Channel geometry

At regional or larger scales, channel cross-sectional shape is usually simplified to be a rectangle since the channel top width is much larger than the channel depth (or bank height). The channel cross-section can be determined by channel width and channel depth. Beighley and Gummadi (2011) presented a methodology for estimating channel cross-sectional dimensions (i.e., channel width and channel depth) at stream-gauge locations by using stage – discharge relationship data and Landsat imagery. They implemented the approach to derive channel cross-sectional dimensions of 82 streamflow gauging locations spread over the Amazon Basin, which were further used to develop the general relationships between channel cross-sectional dimensions and upstream drainage area (or channel geometry formulae) for the entire basin. Their channel geometry formulae are listed as follows.

$$w = 1.956 \, A^{0.413} \quad (A < 10,000 \, km^2) \tag{4}$$

$$w = 0.403 \, A^{0.600} \quad (A \geq 10,000 \, km^2) \tag{5}$$

$$d = 0.245 \, A^{0.342} \tag{6}$$

where $w$ is channel width (unit: m); $d$ is channel depth (unit: m); $A$ is upstream drainage area (unit: $km^2$). Beighley and Gummadi (2011) showed that the channel cross-sectional dimensions estimated from their channel geometry formulae agreed well with those from the formulae by Coe et al. (2008). Based on extensive river morphology data obtained from stations spread throughout the Amazon and Tocantins basins, Coe et al. (2008) derived the general channel geometry formulae for the Amazon Basin and, in their formulae, channel cross-sectional dimensions were also expressed as power functions of upstream drainage area.

The channel geometry formulae of Beighley and Gummadi (2011) were obtained through regression analysis of data from 82 locations over the Amazon Basin, and reflected the average feature of the basin. Directly applying the same formulae and parameters to the entire basin could cause large biases in the estimated channel cross-sectional dimensions for some subregions. In order to reduce those biases, in this study the coefficients in the basin-wide channel geometry formulae of Beighley and Gummadi (2011) were adjusted for a majority of the 10 subregions (Fig. 2c) based on channel cross-sectional dimensions of local locations. The 82 streamflow

gauging locations scattered over the Amazon Basin and each subregion contained a few streamflow gauging locations. For the streamflow gauging locations of the same subregion, the root-mean-square error (RMSE) between the channel cross-sectional dimensions estimated with the channel geometry formulae and the corresponding dimensions presented in Beighley and Gummadi (2011) could be calculated. During the adjustment process, the coefficient of the channel geometry formula (i.e., 1.956, 0.403 or 0.245 in Eqs. (4)–(6)) was multiplied by a factor to reduce the RMSE. The factor values for the 10 subregions are listed in Table 1. The ranges for the channel width and depth of each subbasin are shown in Figs. 2g and 2h, respectively.

It is worth mentioning that Paiva et al. (2013a) also accounted for spatial variability of channel geometry formulae and used various coefficients in their formulae for six zones of the Amazon Basin. In this study, we used both the basin-wide channel geometry formulae and the diverse formulae for various subregions, and investigated the effects of refining channel geometry on modeled surface water dynamics.

In order to convert the calculated channel water depths to river stages, we estimated the riverbed elevations by using the following equation since observed data were not available.

$$E_c = E_{mouth} + \sum_{i=1}^{n} L_i S_i + \frac{1}{2} L_c S_c \tag{7}$$

where $E_c$ is the average riverbed elevation of the current channel [unit: m]; $E_{mouth}$ is the riverbed elevation at the mouth of the Amazon River [unit: m]; $n$ is the total number of downstream channels; $L_i$ is the flow length of a downstream channel $i$ [unit: m]; $S_i$ is the average riverbed slope of a downstream channel $i$ [dimensionless]; $L_c$ is the flow length of the current channel [unit: m] and $S_c$ is the average riverbed slope of the current channel [dimensionless]. $E_{mouth}$ is assumed to be the negative channel depth at the mouth of the Amazon River, which is calculated with Eq. (6). The riverbed slopes were extracted from the DEM and could contain uncertainties since the DEM did not reflect the actual riverbed elevations.

## 2.6 Manning roughness coefficients for channels

The Manning roughness coefficient for channels reflects the resistance to water flows in channels and is determined by many factors, such as roughness of riverbed and riverbank, shape and size of channel cross-

sections and channel meanderings. In general, within a basin these factors have considerable spatial heterogeneities. Therefore it is more reasonable to use spatially varying coefficients estimated based on these factors than using a constant coefficient. However, distributed hydrologic modeling requires a channel Manning coefficient for each subbasin. It is not realistic to separately estimate each of the Manning coefficients given the

lack of information. For continental scale studies, the river network consists of river channels of distinct magnitude orders. Riverbed resistance plays a relatively smaller role in water flows of larger channels. Assuming that the Manning coefficient decreases linearly with the channel top width, Decharme et al. (2010) showed that the assumed relationship produced acceptable variation in flow velocity in a global application of the ISBA-TRIP continental hydrologic modeling system. Getirana et al. (2012) expressed the Manning coefficient as a power

function of the channel depth in their study of inundation dynamics in the Amazon Basin. In our study, the Manning coefficient also depended on the channel depth and was estimated using the following function:

$$n = n_{min} + (n_{max} - n_{min}) \left( \frac{h_{max} - h}{h_{max} - h_{min}} \right) \tag{8}$$

where the maximum Manning coefficient $n_{max}$ is for the channel with the shallowest channel depth and the minimum Manning coefficient $n_{min}$ is for the channel with the largest channel depth. Following Getirana et al. (2012), $n_{max}$ and $n_{min}$ were set as 0.05 and 0.03, respectively. In addition, a few other studies of the Amazon

Basin adopted similar values around the range of 0.03 – 0.05 for the Manning coefficient (Beighley et al., 2009; Paiva et al., 2013a; Yamazaki et al., 2011). In Eq.(8), $h_{max}$ and $h_{min}$ are the maximum and minimum channel depths in all the channels, and were estimated to be 50.64 and 0.96 m, respectively, using the method described in Sect. 2.5. The variable $h$ is the depth of the current channel. The spatial distribution of the channel Manning coefficient is shown in Fig. 2i.

In this study, the function of the Manning coefficient (i.e., Eq. (8)) was compared to those of Decharme et al. (2010) and Getirana et al. (2012). In general, compared to the equations of the two previous studies, Eq. (8) gave smaller Manning coefficients and resulted in better simulation of hydrographs, which suggested that Eq. (8) was more appropriate for the simulations of this study.

## 2.7 Control simulation

The aforementioned factors could have important impacts on modeling surface hydrology of the Amazon Basin. We configured a control simulation (abbreviated as "CTL") using the preferred methodologies for five aspects: (1) the inundation scheme was turned on; (2) vegetation-caused biases in the DEM data were alleviated; (3) the basin-wide channel geometry formulae were refined for different subregions; (4) the Manning coefficient varied with the channel size; (5) the diffusion wave method was used to represent river flow in channels. The control simulation was run for 14 years (1994 – 2007) and the results of 13 years (1995 – 2007) were evaluated against gauged streamflow data and remotely sensed river stage and inundation data.

## 3 Model evaluation

### 3.1 Streamflow

The observed daily streamflow data for model evaluation were from 13 stream gauges operated by the Brazilian Water Agency. Eight of the 13 gauges either control the major area of a tributary subregion or are typical gauges in their tributary subregions. None of the 13 gauges is located in the tributary subregion "Upper-Solimoes tributaries" in the western Amazon Basin. Most of this subregion is controlled by the Santo antonio do ica gauge at the upper mainstem. The remaining four gauges are located along the middle or lower mainstem.

The simulated daily streamflow results were compared with the observed data for a 12-year period (1995 – 2006) at the 13 stream gauges (Fig. 3). The Nash–Sutcliffe efficiency coefficient (NSE) and the relative error of mean annual streamflow (RE) were calculated for each gauge (Fig. 3). For the majority of the 13 gauges, daily streamflow values were reproduced fairly well. The NSE value is higher than 0.62 at seven gauges. The four gauges with NSE values lower than 0.5 have high absolute values of RE (i.e., > 0.20), which suggests that large biases in runoff inputs for the areas upstream of those gauges degrade the streamflow results. Overall, runoff inputs have large negative biases in the western portion of the Amazon Basin, and large positive biases in the southern and southeastern portions. The runoff biases could be caused by errors in the precipitation forcing dataset or errors in the land surface water fluxes calculated by the land surface model (e.g., canopy evaporation, plant transpiration, and soil evaporation). In general, the simulated streamflow results are comparable to those of a few previous studies (e.g., Getirana et al., 2012; Yamazaki et al., 2011) and slightly worse than those of Paiva et al. (2013a).

## 3.2 River stage

The observed river stages were based on altimetry data obtained by the Envisat satellite. The altimetry data were stored in the Hydroweb server (http://ctoh.legos.obs-mip.fr/products/hydroweb). This study utilizes river stages of 11 virtual stations which correspond to 11 of the 13 stream gauges used in Sect. 3.1. Each of the 11 virtual stations is close to one gauge: the virtual station and the gauge are located in either the same subbasin or two neighboring subbasins. There is no virtual station close to the Altamira or Cach da porteira-con gauges. The simulated river stages are relative elevations as they were calculated from the riverbed elevation and the channel water depth. The method for estimating the riverbed elevation is described in Sect. 2.5. Considerable uncertainties in the riverbed elevation are expected due to the large uncertainties in the riverbed elevation at the mouth and the riverbed slopes. Therefore the simulated river stage of a channel is negatively affected by parameter biases of downstream channels and cannot be directly compared to the observations. The timing and magnitude of simulated river stage fluctuations were compared to those of observed data. The comparison was conducted at the daily scale during a 6-year period (2002 – 2007) for the 11 subbasins containing the 11 virtual stations (Fig. 4). For better visual comparison, the simulated river stages of the same subbasin were shifted by a uniform height to coincide with the observations. The Pearson correlation coefficient between the simulated river stages and the observed data were calculated. The timing of the simulated river stage fluctuations is in good agreement with the observations in all 11 subbasins, with Pearson correlation coefficients ranging from 0.830 to 0.960. Moreover, the standard deviations for the simulated and observed river stages were also calculated. The river stage fluctuations are captured well in the majority of the 11 subbasins, and overestimated for the subbasins of 4 gauges (i.e., Canutama, Acanaui, Serrinha and Santo antonio do ica): the standard deviation of the simulated river stages is much larger than that of the observed data, which could be primarily due to a few reasons: (1) overestimation of streamflow peaks (e.g., Canutama and Acanaui), which could be caused by biases of runoff inputs or underestimation of flood extent in the upstream area; (2) uncertainties in model parameters of channel cross-sectional geometry, channel Manning coefficients, etc. Overall, in terms of the timing and magnitude of fluctuations, the modeled river stages of this study are comparable to those reported in some previous investigations (Coe et al., 2008; Getirana et al., 2012; Paiva et al., 2013a).

### 3.3 Flood extent

The simulated flood extent results were evaluated using the Global Inundation Extent from Multi-Satellite (GIEMS) data (Papa et al., 2010; Prigent et al., 2007, 2012). The GIEMS data contained monthly surface water area during a 15-year period (1993 – 2007) for each of the land pixels of equal area (i.e., 773 km$^2$). The area-weighted averaging method was used to convert the grid based surface water extent data to subbasin based data for using in this study. Lake area was not deducted from the GIEMS data because in the Amazon Basin the lakes usually were located in the low portion of one subbasin and the simulated inundated area also contained lake areas.

The simulated monthly flood extent results (including channel surface area and flooded area over floodplains) were compared to the GIEMS data during a 13-year period (1995 – 2007) for 10 subregions and the entire Amazon Basin (Fig. 5). The Pearson correlation coefficient and the mean annual relative difference between the simulated flood extent results and the observations were calculated. The timing of inundation was reproduced well for most area of the Amazon Basin: the Pearson correlation coefficient is equal to or larger than 0.727 at seven of the ten subregions and the entire basin. The mean annual value of simulated flood extent is comparable to that of the GIEMS observations for major portion of the basin: the absolute value of the mean annual relative difference is less than 0.23 at seven of the ten subregions and the entire basin.

The spatial pattern of simulated flood extent was also compared to that of the GIEMS observations for high-water and low-water seasons (Fig. 6). For each subbasin, the simulated or observed flooded fractions of 13 years (1995 – 2007) were averaged for the high-water season (April, May and June) and low-water season (October, November and December), respectively. Both the observations and the simulated results show evident inundation in the regions near the middle and lower mainstem. The observed inundation in the upper Madeira subregion and middle Negro subregion is partially captured by the model. The comparison also shows spatially varying differences between the modeled and observed flood extent (Figs. 6e and 6f). The modeled flood extent exceeds the observations in the lower Madeira subregion near the mainstem and around the major reaches in the middle Negro subregion. At the same time, the modeled flood extent is lower than the observations for some subbasins in the mainstem, upper Madeira, Upper-Solimoes and middle Negro subregions.

The aforementioned discrepancies between the simulated flood extent and the GIEMS data could be related to biases of runoff inputs, which have important effects on the streamflow simulation, as noted earlier. The runoff biases (i.e., the differences between runoff inputs and "actual" runoff) in the upstream area of a stream gauge

could be inferred from the long-term mean streamflow errors. Comparing the annual streamflow errors to the flood extent errors upstream of the gauge from year 1995 to 2006 (Fig. 7) shows that runoff biases could be the partial cause for the flood extent discrepancies. For three of the ten gauges (i.e., (b) Itaituba, (g) Tabatinga and (h) Acanaui), the upstream flood-extent discrepancies are consistent with the streamflow errors (i.e., both are positive or negative) in all 12 years. For the other seven gauges, upstream flood-extent discrepancies and streamflow errors are consistent for some years, but contradictory for other years. This result suggests that flood extent discrepancies were also caused by other factors such as (1) uncertainties in model parameters including floodplain topography, channel cross-sectional geometry, channel Manning coefficients, the riverbed slope, etc.; (2) surface water bodies (e.g., lakes and swamps) not represented by the model were lumped into the inundated floodplains; (3) subsurface processes and wetlands sustained by groundwater were not simulated; and (4) inundation could be underestimated or overestimated in the GIEMS data which were of comparatively low resolution (Hess et al., 2015; Prigent et al., 2007). The effects of model parameters (including floodplain topography, channel cross-sectional geometry and channel Manning coefficients) on the inundation results were investigated in the sensitivity study.

Although the GIEMS data have non-negligible uncertainties, it is useful to check how our results may differ from those of previous studies using the GIEMS data as the common benchmark. Overall compared to the GIEMS data, the spatial inundation patterns of this study were slightly better than those of Getirana et al. (2012), and comparable to those of Yamazaki et al. (2011) and Paiva et al. (2013a). In terms of monthly total flooded areas, Getirana et al. (2012), Paiva et al. (2013a) and this study were comparable at the whole-basin scale, while the results from Getirana et al. (2012) and this study were closer to the GIEMS data than those of Paiva et al. (2013a) at the subregion scale.

**4 Sensitivity study**

A sensitivity study was carried out to investigate the roles of the following factors in modeling of surface hydrology of the Amazon Basin: (1) representing floodplain inundation; (2) alleviating vegetation-caused biases in the DEM; (3) refining channel geometry; (4) adjusting Manning coefficients; and (5) accounting for backwater effects. Six scenario simulations were so designed that for each simulation only one of the above five factors was changed from the control simulation described in Sect. 2.7 (Table 2). All simulations were run for 14 years (1994

– 2007) and the results of 13 years (1995 – 2007) were analyzed. The results of the control simulation were compared with those of each scenario simulation to separately examine the impacts of each factor on the modeled streamflow, river stages and inundation.

The inundation scheme was turned off (i.e., river water could not spill out of the main channel and enter the floodplain) in the second simulation (abbreviated as "NoInund") of Table 2. The results of the control simulation were compared to those of the simulation "NoInund" to reveal the role of the inundation scheme in improving the modeled streamflow and river stages (Sect. 4.1).

The original HydroSHEDS DEM data without the correction of vegetation-caused biases were used in the third simulation (abbreviated as "OriDEM"); the basin-wide channel geometry formulae were not refined for different subregions and were directly used for the entire basin in the fourth simulation (abbreviated as "OriSec"). The results of these two simulations were contrasted with those of the control simulation to show the effects of geomorphological parameters on modeling surface water dynamics (Sect. 4.2 and 4.3).

A few previous studies at the Amazon Basin used a constant Manning coefficient for all the channels (e.g., 0.04 was used by Beighley et al., 2009; and 0.03 was used by Yamazaki et al., 2011). A constant Manning coefficient of 0.03 and 0.04 was used in the fifth and sixth simulations, respectively (abbreviated as "n003" and "n004"). The diffusion wave method was replaced by the kinematic wave method for representing water flow through channels in the seventh simulation (abbreviated as "KW"). These three simulations were compared with the control simulation to reveal the impacts of river flow representations on modeled surface hydrology (Sect. 4.4 and 4.5).

In the comparisons between the control simulation and the contrasting scenario simulations, we examined the model results of various locations spread over the Amazon Basin, including streamflow at 13 major mainstem or tributary gauges (Fig. 8), river stages near 11 major gauges (Fig. 9), the mainstem water surface profile (Fig. 10), inundation of 10 subregions (Fig. 11), and spatial patterns of inundation differences for the entire basin (Fig. 12). In the following discussions, Figs. 8 – 12 are used jointly to reveal the impacts of the five factors on surface water dynamics.

## 4.1 Representing floodplain inundation

The comparison of streamflow results between the control simulation "CTL" and the simulation "NoInund" shows that incorporating the inundation scheme evidently improves the modeled streamflow. More specifically, streamflow peaks are reduced and delayed, and the streamflow hydrographs become smoother (Fig. 8). The impacts are especially prominent in the subregions with evident inundation (e.g., Fig. 8c) and at the gauges on the middle and lower mainstem (Figs. 8j – 8m). This result demonstrates that floodplains play a significant role in regulating streamflow of the Amazon Basin.

Fig. 9 shows that incorporating the inundation scheme has prominent impacts on the modeled river stages of most of the 11 subbasins examined in this study: the river-stage peaks are attenuated and delayed, and the river-stage timing and fluctuation magnitude are improved. The impacts are most obvious in the subregions with evident inundation (e.g., Fig. 9b) and in the middle and lower mainstem (Figs. 9h – 9k). One exception is that the large improvement of river stages near the Itaituba gauge (Fig. 9a) is primarily caused by the improvement of mainstem river stages because the Itaituba gauge is close to the lower mainstem and its river stages are influenced by the mainstem through backwater effects.

Including the inundation scheme brings about changes of the mainstem water surface profile and the changes are more evident in the rising-flood season than in other seasons (Fig. 10). In the rising-flood season, the average water surface profile is lowered for the entire mainstem section examined here and the large river-stage differences occur in the middle mainstem with magnitude up to more than 5 m (Fig. 10a). In the high-water season, the average water surface profile is also lowered (Fig. 10b). However, Figure 10c shows that in the falling-flood season the mainstem river stages are raised because water stored in the floodplains returns to the river channels. Similar to the rising-flood season, large river-stage differences appear in the middle mainstem with magnitude of about 3 m. In the low-water season, the average water surface profile is slightly lowered (Fig. 10d). It should be noted that the mainstem river stages are first raised and then lowered during the three months (Figs. 9h – 9k).

The above comparisons and analyses reveal that incorporating the inundation scheme into hydrologic modeling has prominent impacts on the simulated surface hydrology in the Amazon Basin and significantly improves both the streamflow and the river-stage hydrographs, especially at reaches whose upstream area involves large floodplains. This result suggests that floodplain inundation is an important component of the surface water dynamics in the Amazon Basin and should be represented in hydrologic modeling for this basin.

Some previous studies also examined and reported the impacts of representing the floodplain inundation on the modeled surface hydrology in the Amazon Basin (Getirana et al., 2012; Paiva et al., 2013a; Yamazaki et al., 2011). Yamazaki et al. (2011) showed the impacts of floodplain inundation on the streamflow, water depths, and flow velocities at the Obidos gauge (in their Fig. 5) and the mainstem water surface profile (in their Fig. 7). Getirana et al. (2012) demonstrated the effects of floodplain inundation on streamflow of a few mainstem gauges (in their Fig. 16). When investigating the impacts of floodplain inundation on surface hydrology, these two studies used the kinematic wave river routing method that could not represent the important backwater effects in the Amazonia, while we used the diffusion wave river routing method that captured backwater effects. Backwater effects were also represented in the dynamic wave river routing method used by Paiva et al. (2013a) when they studied the impacts of floodplain inundation on streamflow of a few major tributary or mainstem gauges including Obidos and Manacapuru (in their Table 2 and Fig. 14). Besides streamflow, in this study we also examined and revealed the prominent impacts of floodplain inundation on the river stages near 11 major gauges or along the mainstem.

## 4.2 Correcting DEM

The vegetation-caused biases in the HydroSHEDS DEM data were alleviated via DEM correction. This lowered the floodplain elevations and changed the slope of the elevation profile, which could lead to changes in simulated flood extent. Figure 11 shows that the DEM correction increases flood extent in all 10 subregions. The increase of inundation postpones and lowers streamflow peaks in the downstream channels, especially in the middle and lower mainstem (Figs. 8 j – m).

The increase of inundation also brings about changes in river stages: the magnitude of river stage fluctuations is reduced in the 11 subbasins (Fig. 9). In the middle mainstem, the river stages averaged over three months is lowered in the rising-flood and high-water seasons (Figs. 10a and 10b) and elevated in the falling-flood and low-water seasons (Figs. 10c and 10d), with magnitude up to about 1 m.

Figures 12a and 12b show that DEM correction leads to inundation changes in many subbasins: while flood extent is mostly enlarged, DEM correction could increase the slope of the elevation profile in some subbasins and reduce flood extent.

The vegetation-caused biases in DEM data were alleviated with various approaches in a few previous studies modeling the surface hydrology in the Amazon Basin (Baugh et al., 2013; Coe et al., 2008; Getirana et al., 2012; Paiva et al., 2011, 2013a; Wilson et al., 2007; Yamazaki et al., 2011). Most of these studies did not examine and explicitly report the effects of the DEM correction on the modeled results. Baugh et al. (2013) demonstrated that alleviating vegetation-caused biases in DEM could improve the modeled water levels and inundation over floodplains adjacent to a 280-km reach of the central Amazon (in their Figs. 2 and 5).

## 4.3 Refining channel geometry

Adjusting channel cross-sectional geometry could evidently affect the simulated surface water area (Fig. 11) and the changes are caused by two mechanisms: (1) reducing channel cross-sectional area, which is equivalent to reducing channel conveyance capacity, could increase flooded area over floodplains, and vice versa; (2) broadening the channel width, hence increasing channel surface area, and vice versa. The nine tributary subregions can be placed in five categories according to the changes of channel cross-sectional area, the channel width and the total surface water area (Table 3). The channel geometry of the mainstem is not adjusted. The inundation changes in the tributary subregions affect streamflow in the mainstem and slightly delays and attenuates the inundation peak there (Fig. 11j).

Figure 8c shows that channel geometry changes significantly postpone and lower the streamflow peak at the gauge in the lower Madeira subregion. The reason is that the channel cross-sectional area is multiplied by a factor of 0.36 (Table 1), which evidently increases inundation in this subregion (Fig. 11c). Similar phenomenon is observed at the gauge "Cach da porteira-con" in the Northeast subregion (Fig. 8h), where the channel cross-sectional area is multiplied by a factor of 0.48. Inundation changes caused by refining channel geometry in other subregions are comparatively smaller than those of the Madeira and Northeast subregions, and do not result in significant alterations in streamflow (Fig. 8).

Adjustment of channel geometry could have evident effect on the river stage of the local channel. The mechanism for channel geometry changes to affect river stages is not straightforward. For instance, reducing the channel width could raise the river stage and hence increase the flow velocity or inundation, which, in turn tend to lower the river stage (Fig. 13). The simulated results of this study show that, in most circumstances, reducing the channel width raises the river stage of the local channel (Figs. 9b, 9c and 9d) and vice versa (Figs. 9e and 9f).

In Fig. 9a, this rule does not apply from about day 160 to350, which could be caused by backwater effects: the river stage of this channel is influenced by that of the mainstem section downstream of the Obidos gauge.

Channel geometry changes could also influence river stages of remote downstream channels. The channel morphology of the mainstem is not adjusted. So the river stage changes along the mainstem are caused by inundation changes in the upstream area. The channel-geometry adjustment of this study increases inundation in the major portion of the Amazon Basin, which influences river stages along the mainstem, particularly in the middle reaches: the river stages averaged over three months are lowered in the rising-flood and high-water seasons (Figs. 10a and 10b) and elevated in the falling-flood and low-water seasons (Figs. 10c and 10d), with magnitude up to about 1 m. The phenomenon can also be observed in Figs. 9h–k.

The sensitivities of modeled surface hydrology to channel geometry were also investigated by some former studies (Paiva et al., 2013a; Yamazaki et al., 2011). Yamazaki et al. (2011) perturbed the channel width or depth by a uniform percentage for all the channels and examined the effects of these channel-geometry changes on streamflow of the Obidos gauge and the flooded area over the central Amazon region (in their Fig. 13). Paiva et al. (2013a) perturbed the channel width by a uniform percentage or perturbed the channel-bottom level by a uniform height, which was equivalent to perturbing the channel depth by a uniform value, and investigated the effects of these channel-geometry changes on streamflow of the Obidos gauge, channel water depths of the Manacapuru gauge, and the total flooded area of the entire Amazon Basin (in their Fig. 10). These two studies showed the sensitivities of modeled surface hydrology to channel geometry, as well as the interactions between streamflow, water depths and inundation. They pointed out the importance of channel geometry and provided a foundation to this study. Here, channel-geometry changes were caused by the process of refining the channel cross-sections, and the changes varied spatially (Table 1). We examined the effects of channel-geometry changes on inundation of 10 subregions, streamflow of 13 gauges, river stages near 11 gauges, as well as the mainstem water surface profile. In addition, the effects of channel-geometry changes on modeled surface water dynamics were analyzed with approaches of which some were different from those of the former studies.

## 4.4 Varying Manning roughness coefficients

A few studies for the Amazon Basin (e.g., Paiva et al., 2013a; Yamazaki et al., 2011) revealed some sensitivities of surface hydrology to the Manning coefficient. Yamazaki et al. (2011) perturbed the Manning coefficient by a uniform percentage for all the channels and examined the effects on streamflow of the Obidos

gauge and the flooded area over the central Amazon region (in their Fig. 13). Using a similar approach, Paiva et al. (2013a) investigated the effects of the Manning coefficient on streamflow of the Obidos gauge, channel water depths of the Manacapuru gauge, and the total flooded area of the entire Amazon Basin (in their Fig. 10). These studies revealed that increasing the Manning coefficient could raise the river stage, expand the flooded area, and

reduce and delay the flood peak. Instead of a uniform perturbation, we varied the Manning coefficient with the channel depth and examined the effects on flood extent of 10 subregions, streamflow of 13 gauges, river stages near 11 gauges, and the mainstem water surface profile.

The streamflow Nash–Sutcliffe efficiency coefficients (NSEs) of "CTL" were compared with those of "n003" and "n004" (Table 4). The NSEs of "CTL" are higher than those of "n004" at 10 of the 13 gauges (except

Fazenda vista alegre, Itapeua and Manacapuru) and higher than those of "n003" at 12 of the 13 gauges (except Obidos). These results suggest that the spatially varying Manning coefficients are more appropriate than the uniform Manning coefficient of 0.03 or 0.04 for the simulations of this study.

The spatially varying Manning coefficients range from 0.03 to 0.05 and are equal to or larger than the Manning coefficient of 0.03. The spatially varying Manning coefficients result in larger flood extent than the

uniform coefficient of 0.03 (Fig. 11). The larger Manning coefficient leads to the lower flow velocity, larger wet cross-sectional area and thereby higher river stage (Fig. 9), which increase local inundation, as well as upstream inundation due to backwater effects. Inundation increases in the upstream area postpone and attenuate flood waves at the downstream gauges (Fig. 8).

Increases of the Manning coefficients not only affect local and upstream river stages as discussed above, but

also influence downstream river stages. Inundation increases in the upstream area have impact on streamflow rates and hence river stages in the downstream channels. Therefore river stages are influenced by not only downstream and local Manning coefficients, but also upstream Manning coefficients. Figure 9 shows that the Manning coefficient increases result in rise of river stages in most circumstances, which suggests that the local and downstream effects play a dominant role: increases of Manning coefficients reduce flow velocities, enlarge

wet cross-sectional area and hence elevate river stages. However, in the lower mainstem the upstream effects may overwhelm the local and downstream effects. For instance, Fig. 9k shows that, during the rising-flood period (before about the day 150), the Manning coefficient increases reduce river stages at the Obidos gauge. The main reason is that the larger Manning coefficient promotes inundation in the upstream area, which results in smaller streamflow rates in the lower mainstem for the rising-flood period.

## 4.5 Backwater effects

Besides the above factors, backwater effects also play a significant role in the surface water dynamics of the Amazon Basin, particularly in the middle and lower portions of this basin that have very mild topography (e.g., Fig. 10e). In this study, backwater effects were represented in the diffusion wave routing method for six of the seven simulations (including the control simulation). In the remaining simulation (i.e., KW), the diffusion wave method was replaced with the kinematic wave method that could not represent backwater effects. The results of the control simulation were compared with those of the simulation KW to reveal backwater effects on surface water dynamics.

(1) Backwater effects on flood extent

In the diffusion wave method, backwater effects could decrease the friction slope and hence reduce the flow velocity (Eqs. (2) and (3)), and vice versa. For the same streamflow rate, reduction of the flow velocity leads to larger wet cross-sectional area and thereby higher river stage, which could increase local inundation if the river stage exceeds the bank top, as well as increase upstream inundation due to backwater effects. This mechanism is similar to the aforementioned mechanism that increases of the Manning coefficients could promote local and upstream inundation. Using the same reasoning, backwater effects also could increase the flow velocity and eventually reduce inundation. Figure 11 shows that the flood extent of the control simulation is evidently larger than that of the simulation KW for nine of the ten subregions and the entire Amazon Basin, which suggests that the dominant role of backwater effects is to increase inundation for this basin. However, backwater effects also could reduce inundation as demonstrated in the subregion "Upper-Solimoes tributaries" (Fig. 11f). Figures 12j and 12k illustrate that backwater effects tend to increase inundation in the middle and lower mainstem, lower Negro and lower Madeira subregions, where the topography is flat and the streamflow rate is comparatively high. Yamazaki et al. (2011) showed the backwater effects on the flooded area over the central Amazon region (in their Fig. 9). In their results, backwater effects promoted the flooded area to a lesser extent compared to our study, which may be due to the differences in the channel or floodplain geomorphology data used in the two studies. Paiva et al. (2013b) used the dynamic wave method to represent river flow in the Solimoes River basin, which is the western upstream portion of the Amazon Basin. They discussed the important role of backwater effects in the inundation dynamics of the Amazon. In this study, we examined the impacts of backwater effects on flood extent

in the 10 subregions constituting the Amazon Basin (Fig. 11), and demonstrated the spatial pattern of flood extent changes caused by backwater effects (Figs. 12j and 12k).

(2) Backwater effects on streamflow

Backwater effects bring about inundation increases in the subbasins of the upstream area, which have impact on streamflow in the downstream channels. Inundation increases in the upstream area could delay and attenuate hydrographs in the middle and lower mainstem (Figs. 8k–m). These results agree with Paiva et al. (2013a, 2013b) which demonstrated the important role of the backwater effects in streamflow of the mainstem and tributaries of the Amazon Basin (Table 2 and Fig. 14 of Paiva et al., 2013a; Table 2 and Figs. 3, 4 and 9 of Paiva et al., 2013b).

Backwater effects could increase the friction slope and hence increase the flow velocity, which resulted in changes of the hydrograph. For instance, Fig. 8c shows that at the lower Madeira River the flow peak of the control simulation is about 20 days earlier than that of the simulation 'KW'. The Madeira River reaches its highest stage about 1 – 2 months earlier than the mainstem (compare Fig. 9b and Fig. 9j; also see Meade et al., 1991). This time difference in peak stage makes the slope of the river surface steep in the rising-flood period of the Madeira River, which increases the flow velocity and leads to an earlier timing of the streamflow peak. This phenomenon of backwater effects on the streamflow timing cannot be captured in the simulation 'KW' because in the kinematic wave method the flow velocity depends on the riverbed slope instead of the river surface slope.

(3) Backwater effects on river stages

It is discussed above that backwater effects could influence local and upstream river stages by changing the local flow velocity, but they could also affect downstream flow rates, which consequently influence downstream river stages. Therefore the river stage of a channel is influenced by not only the local and downstream backwater effects, but also the backwater effects in the upstream area. The combined impact significantly attenuates both temporal (Fig. 9) and spatial (Fig. 10) river stage fluctuations. This result is consistent with that of Yamazaki et al. (2011), which primarily discussed the water depths at the Obidos gauge (in their Fig. 5b) and the mainstem water surface profile in one month (in their Fig. 7a), while this study examined river stages near 11 major gauges on tributaries or the mainstem (Fig. 9), and the mainstem water surface profiles in four seasons (Fig. 10). Moreover, in the results of Yamazaki et al. (2011), the backwater effects on river stages were not as prominent as those simulated in this study, which may be due to the discrepancies in channel geometry or floodplain

topography between the two studies. In addition, the result of this study agreed with Paiva et al. (2013b), which discussed the backwater effects on river stages in the Solimoes River basin.

Figure 10 also shows that the river stages of the middle and lower mainstem drop significantly when backwater effects are not represented, especially during the rising-flood, falling-flood and low-water periods (Figs. 10a, 10c and 10d). The sea level was used as the boundary condition at the basin outlet when the diffusion wave method was employed to simulate water flow in channels. The river stages of the middle and lower mainstem were influenced by the sea level via backwater effects. In this study the sea level was assumed to be fixed, which was similar to the approach of Yamazaki et al. (2011). In reality, the sea level rises and falls regularly, which exerts varying impact on river flow (e.g., Yamazaki et al., 2012). The effect of sea level variation on river hydrology can be represented when the surface-water transport model is coupled with an Earth system model. Furthermore, this modeling framework could be used to investigate the potential impact of sea level rise on the terrestrial hydrologic cycle due to climate change.

## 5 Summary and discussion

Floodplain inundation is a key component of surface water dynamics in the Amazon Basin. A macro-scale inundation scheme for representing floodplain inundation was incorporated into the Model for Scale Adaptive River Transport (MOSART) and the extended model was applied to the entire Amazonia. Efforts were made to deal with a few challenges in continental-scale modeling of surface hydrology in this vast basin: (1) we refined the floodplain topography by alleviating the spatially varying vegetation-caused biases in the HydroSHEDS DEM data. To our knowledge, this was the first time that the spatial variability of vegetation-caused biases in the DEM data was explicitly considered in hydrologic modeling for the entire Amazon Basin; (2) we improved the representation of spatial variability in channel cross-sectional geometry by refining the basin-wide channel geometry formulae for various subregions; (3) the Manning roughness coefficient varied with the channel depth to reflect the general rule that the relative importance of riverbed resistance in river flow declined with the increase of river size; (4) we accounted for the backwater effects in the river routing method to better represent river flow in gentle-slope reaches.

The model results were evaluated against in situ streamflow data as well as remote sensing river-stage and inundation data. The simulated streamflow results were compared with the observed data from 13 major stream

gauges (Fig. 3). The streamflow hydrographs were reproduced fairly well for the majority of the 13 gauges. The simulated river stages were compared to the altimetry data obtained by the Envisat satellite for the 11 subbasins containing or close to 11 of the 13 gauges (Fig. 4). The timing of river stage fluctuations was captured well for all 11 subbasins and the magnitude of river stage fluctuations was reproduced well for most of the 11 subbasins. The simulated monthly flood extent results were compared against the GIEMS satellite data for the 10 subregions and the entire basin (Fig. 5). For the time series of the lumped flood extent, the model results were comparable to the GIEMS observations in most subregions of the basin. The spatial pattern of modeled inundation was also contrasted with that of the GIEMS observations (Fig. 6). While the model results resemble the overall spatial pattern of the observed inundation, the comparison also shows spatially varying flood extent discrepancies between the simulation and observations which could be partially explained by the biases of runoff inputs (Fig. 7). Those discrepancies could also be due to uncertainties in geomorphological parameters, missing representations of some potentially important hydrologic processes, as well as biases of the GIEMS data.

In the sensitivity study, the results of the control simulation were compared with those of a few scenario simulations for investigating the roles of the following factors in the hydrologic modeling for the Amazon Basin.

(1) Representing floodplain inundation. It was shown that representing floodplain inundation could evidently improve the modeled streamflow at 13 major gauges (Fig. 8). It was also demonstrated that representing floodplain inundation could improve the river-stage timing and fluctuation magnitude near 11 major gauges (Fig. 9), and have prominent impacts on the modeled water surface profile along the mainstem (Fig. 10). These results showed that floodplain inundation played an important role in surface hydrology of the Amazon Basin and should be represented in the hydrologic modeling for this basin.

(2) Alleviating vegetation-caused biases in the DEM. The DEM correction leaded to evident inundation changes, of which most were inundation increases, in many subbasins (Figs. 11, 12a and 12b). The DEM correction could lower and postpone streamflow peaks, especially at the mainstem (Fig. 8) and attenuate river-stage fluctuations in the tributaries and the mainstem (Figs. 9 and 10). To our knowledge, for hydrologic modeling of the entire Amazon Basin, the impacts of correcting vegetation-caused biases in the DEM on the modeled surface hydrology were not reported in the past.

(3) Refining channel cross-sectional geometry. The channel geometry refinements could evidently increase or decrease the inundation area for various locations of the basin (Figs. 11, 12d and 12e). Those refinements could obviously improve the streamflow hydrograph (Figs. 8c and 8h), and raise or lower river stages in the

tributaries and the mainstem (Figs. 9 and 10). These results demonstrated the importance of improving the representation of spatial variability in channel geometry.

(4) Adjusting Manning coefficients. The streamflow hydrographs of the scenario simulations suggested that the spatially varying Manning coefficients were more appropriate than the uniform Manning coefficient of 0.03 or 0.04 for the hydrologic modeling of this study. The comparison between the control simulation, where the Manning coefficient varied from 0.03 to 0.05, and the simulation using the uniform Manning coefficient of 0.03 revealed that increasing the value of the Manning coefficient could obviously promote inundation (Figs. 11, 12g and 12h), reduce and delay streamflow peaks (Fig. 8), and mostly raise river stages (Figs. 9 and 10). One exception was that an increase in the Manning coefficient could lower the river stages in the lower mainstem during the rising-flood period (Fig. 9k).

(5) Representing backwater effects. The comparison between scenario simulations showed that the backwater effects could prominently increase inundation in most of the 10 subregions, especially in the area near the middle and lower mainstem and in the lower Negro basin (Figs. 11, 12j and 12k), and reduce inundation in some circumstances (Figs. 11f, 12j and 12k). Representing backwater effects could evidently lower and delay streamflow peaks, improve the hydrographs in the middle and lower mainstem (Figs. 8k–m), as well as increase the flow velocity and lead to an earlier timing of the streamflow peak (e.g., Fig. 8c). It was also illustrated that representing backwater effects could significantly attenuate the modeled river stage fluctuations in the mainstem and tributaries (Fig. 9), and smooth the mainstem water surface profile (Fig. 10).

The understanding obtained in this study could be helpful to improving the modeling of terrestrial surface water dynamics at the global scale. Besides the Amazon Basin, alleviating the vegetation-caused biases in the DEM data is also worthwhile for other basins with considerable inundation and extensive forested area, such as the Congo Basin. The DEM correction can be tested globally for its impacts on surface hydrologic modeling. It is shown that a simple method can improve the representation of channel cross-sectional geometry and consequently the modeled surface hydrology, which implies that representing the spatial variability of channel morphology should be emphasized in applications for other regions. The future Surface Water and Ocean Topography (SWOT) mission (Alsdorf et al., 2007) is expected to bring notable advancement in this aspect. It is also demonstrated that, in general, spatially varying Manning coefficients depending on the channel depth result in higher Nash–Sutcliffe efficiency coefficients of streamflow, as compared to the uniform Manning coefficient of 0.03 or 0.04. It is worth investigating the application of this method to other regions, although the Manning

coefficient is empirical and model dependent. Besides the Amazon River, backwater effects also play a significant role in many other rivers, such as the Yangtze River and Mississippi River (Meade et al., 1991). Therefore backwater effects should be accounted for in the global applications where river stages, inundation extent or river flow velocities are investigated. These factors may have impacts on surface hydrology to different degrees for various regions. For instance, DEM correction and backwater effects are expected to have larger impacts on surface hydrology in regions with milder topography.

Subbasins are used as computation units in this study. Surface hydrologic simulations using subbasins as computation units are less scale-dependent than those using square grids as computation units (e.g., Getirana et al., 2010; Tesfa et al., 2014a, 2014b; Yamazaki et al., 2011). For instance, the shape of the digital river network is less dependent on the computation unit size when subbasin units are used, as compared to applications using grid units (e.g., Getirana et al., 2010). In this study, the simulated hydrologic results are comparable to observations, although the subbasin units are relatively coarse (with an average area of 1091.7 km$^2$). For continental or global scale applications, using subbasin units could represent surface water transport more realistically than using grid units when the subbasin size is comparable to the grid size.

At the same time, some aspects of the model could be improved, such as the representation of water exchange between channels and floodplains. In this study, instantaneous channel–floodplain exchange is assumed, which could overestimate flooded area during the rising-flood period, and vice versa during the receding-flood period. The modeling of this exchange process could be improved by including a mechanistic representation of water flow over floodplains. For instance, Alsdorf et al. (2005) demonstrated that the floodplain drainage could be simulated using a linear diffusion model and Miguez-Macho and Fan  (2012) used diffusion wave method to simulate two dimensional flow over floodplains. Moreover, the mechanistic representation of floodplain flow could be used to simulate water exchange over floodplains between neighboring subbasins, which was not accounted for in this study.

In addition, the modeling of surface water dynamics could benefit from integrating the surface-water transport model with land surface models or climate models by representing the interactions between surface hydrology and subsurface water fluxes as well as atmospheric processes. Such interactions could potentially have important effects on surface fluxes, with important implications to modeling of land – atmosphere interactions and tropical forest response to floods and droughts.

**Code availability**

The MOSART code including the inundation parameterization described herein will be distributed through a git repository and made available upon request.

5 **Data availability**

This study used the following datasets, which can be either accessed from the internet or acquired from the corresponding institution or person.

(1) The HydroSHEDS DEM datasets were developed by United States Geological Survey and are available on-line (http://hydrosheds.cr.usgs.gov/ ).

10 (2) The dataset "Global 1km Forest Canopy Height (Simard et al., 2011)" is available on-line (http://webmap.ornl.gov/wcsdown/dataset.jsp?ds_id=10023) from the Oak Ridge National Laboratory Distributed Active Archive Center, Oak Ridge, Tennessee, USA.

(3) Hess, L.L., J.M. Melack, A.G. Affonso, C.C.F. Barbosa, M. Gastil-Buhl, and E.M.L.M. Novo. 2015. LBA-ECO LC-07 Wetland Extent, Vegetation, and Inundation: Lowland Amazon Basin. ORNL DAAC, Oak 15 Ridge, Tennessee, USA. http://dx.doi.org/10.3334/ORNLDAAC/1284

(4) The surface and subsurface runoff inputs of 1-degree resolution were produced by Bertrand Decharme at CNRM/Météo-France (Getirana et al., 2014) and can be acquired by contacting Augusto Getirana (augusto.getirana@nasa.gov).

(5) The streamflow data of the stream gauges can be acquired by contacting the Brazilian Water Agency 20 ANA (Agencia Nacional de Aguas).

(6) The river water levels are mainly based on altimetry data from the Envisat satellite and available from the HydroWeb data base (http://ctoh.legos.obs-mip.fr/products/hydroweb) maintained by CTOH (Center for Topographic studies of the Ocean and Hydrosphere) at LEGOS, France.

(7) The dataset GIEMS (Global Inundation Extent from Multi-Satellite) was developed by Catherine Prigent 25 (Observatoire de Paris), Filipe Aires (Estellus and Observatoire de Paris) and Fabrice Papa (IRD, LEGOS), and can be acquired by contacting Fabrice Papa (fabrice.papa@ird.fr).

## Competing interests

The authors declare that they have no conflict of interest.

## Acknowledgements

We would like to thank the two anonymous reviewers and the editor for their constructive comments and suggestions. We also thank the institutions/researchers for making the datasets described in the "Data availability" section available for this study. This research was supported by the Office of Science of the U.S. Department of Energy as part of the Earth System Modeling program through the Accelerated Climate Modeling for Energy (ACME) project. The Pacific Northwest National Laboratory is operated by Battelle for the U.S. Department of Energy under Contract DE-AC05-76RLO1830.

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

**Table 1**

**Table 1. Coefficients in channel geometry formulae for the 10 subregions.**

| No. | Subregion name | Factor for adjusting channel width ($\alpha_w$) | Channel width coefficient | | Factor for adjusting channel depth ($\alpha_d$) | Channel depth coefficient | Factor for adjusting cross-sectional area ($\alpha_A = \alpha_w \cdot \alpha_d$) |
|---|---|---|---|---|---|---|---|
| | | | $A_u < 10000 \ \text{km}^2$ | $A_u \geq 10000 \ \text{km}^2$ | | | |
| 1 | Xingu | 1.0 | 1.956 | 0.403 | 1.0 | 0.245 | 1.00 |
| 2 | Tapajos | 1.6 | 3.130 | 0.645 | 0.7 | 0.172 | 1.12 |
| 3 | Madeira | 0.6 | 1.174 | 0.242 | 0.6 | 0.147 | 0.36 |
| 4 | Purus | 0.8 | 1.565 | 0.322 | 1.4 | 0.343 | 1.12 |
| 5 | Jurua | 0.7 | 1.369 | 0.282 | 1.5 | 0.368 | 1.05 |
| 6 | Upper-Solimoes tributaries | 1.0 | 1.956 | 0.403 | 1.0 | 0.245 | 1.00 |
| 7 | Japura | 1.8 | 3.521 | 0.725 | 0.7 | 0.172 | 1.26 |
| 8 | Negro | 1.7 | 3.325 | 0.685 | 0.5 | 0.123 | 0.85 |
| 9 | Northeast | 0.6 | 1.174 | 0.242 | 0.8 | 0.196 | 0.48 |
| 10 | Mainstem | 1.0 | 1.956 | 0.403 | 1.0 | 0.245 | 1.00 |

Note: $A_u$ is the upstream drainage area.

**Table 2**

**Table 2. Setup of seven simulations.**

| No. | Inundation scheme | DEM | Channel cross-sectional geometry | Manning roughness coefficients of channels | Method for representing river flow | Abbrevia-tions |
|-----|-----|-----|-----|-----|-----|-----|
| 1 | On | Corrected | Refined | Spatially varying | Diffusion wave method | CTL |
| 2 | Off | Corrected | Refined | Spatially varying | Diffusion wave method | NoInund |
| 3 | On | Original | Refined | Spatially varying | Diffusion wave method | OriDEM |
| 4 | On | Corrected | No refining | Spatially varying | Diffusion wave method | OriSec |
| 5 | On | Corrected | Refined | 0.03 | Diffusion wave method | n003 |
| 6 | On | Corrected | Refined | 0.04 | Diffusion wave method | n004 |
| 7 | On | Corrected | Refined | Spatially varying | Kinematic wave method | KW |

**Table 3**

**Table 3. Refining the channel cross-sectional geometry affects inundated area in tributary subregions. [a]**

| Category | Cross-sectional area [b] | Inundated area over floodplains | Channel width [c] | Channel area | Total surface water area [d] | Subregions |
|---|---|---|---|---|---|---|
| A | − | + | + | + | + | h) Negro |
| B | − | + | − | − | + | c) Madeira; i) Northeast |
| C | + | − | + | + | + | b) Tapajos; g) Japura |
| D | + | − | − | − | − | d) Purus; e) Jurua |
| E | No refining | No change | No refining | No change | No change | a) Xingu; f) Upper-Solimoes tributaries |

Note: a. '+' means increase; '−' means decrease;

b. This variation depends on the factor $\alpha_A$ in Table 1: $\alpha_A > 1$: '+'; $\alpha_A < 1$: '−'; $\alpha_A = 1$: 'No refining';

c. This variation depends on the factor $\alpha_w$ in Table 1: $\alpha_w > 1$: '+'; $\alpha_w < 1$: '−'; $\alpha_w = 1$: 'No refining';

d. This change is shown by inundation results in Fig. 11.

**Table 4**

**Table 4. Nash–Sutcliffe efficiency coefficients (NSEs) of modeled daily streamflow of 12 years (1995 – 2006) at the 13 stream gauges for the simulations 'CTL', 'n004' and 'n003'.**

| Gauge index | Gauge name | NSE of Simulation 'CTL' | NSE of Simulation 'n004' | NSE of Simulation 'n003' | Subregion of the gauge |
|---|---|---|---|---|---|
| a | Altamira | - 0.677 | - 0.765 | - 0.889 | Xingu |
| b | Itaituba | - 0.310 | - 0.354 | - 0.420 | Tapajos |
| c | Fazenda vista alegre | 0.782 | 0.796 | 0.701 | Madeira |
| d | Canutama | 0.678 | 0.659 | 0.567 | Purus |
| e | Gaviao | 0.512 | 0.482 | 0.389 | Jurua |
| f | Acanaui | - 0.160 | - 0.312 | - 0.604 | Japura |
| g | Serrinha | 0.748 | 0.694 | 0.546 | Negro |
| h | Cach da porteira-con | 0.767 | 0.725 | 0.674 | Northeast |
| i | Santo antonio do ica | 0.428 | 0.413 | 0.297 | Mainstem |
| j | Itapeua | 0.570 | 0.593 | 0.140 | Mainstem |
| k | Manacapuru | 0.623 | 0.653 | 0.407 | Mainstem |
| l | Jatuarana+Careiro | 0.819 | 0.813 | 0.787 | Mainstem |
| m | Obidos | 0.911 | 0.907 | 0.931 | Mainstem |

**Fig. 1**

a) Illustration of river overflow

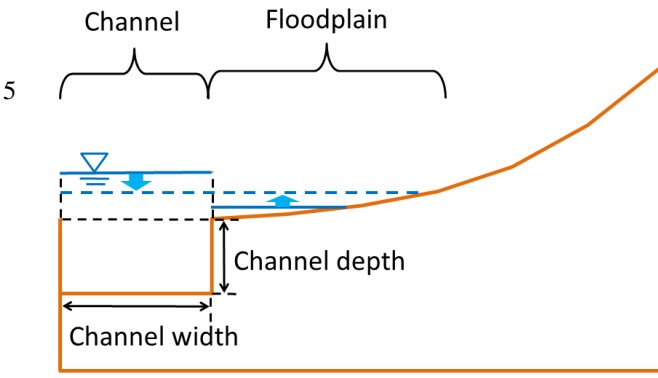

b) Elevation profiles

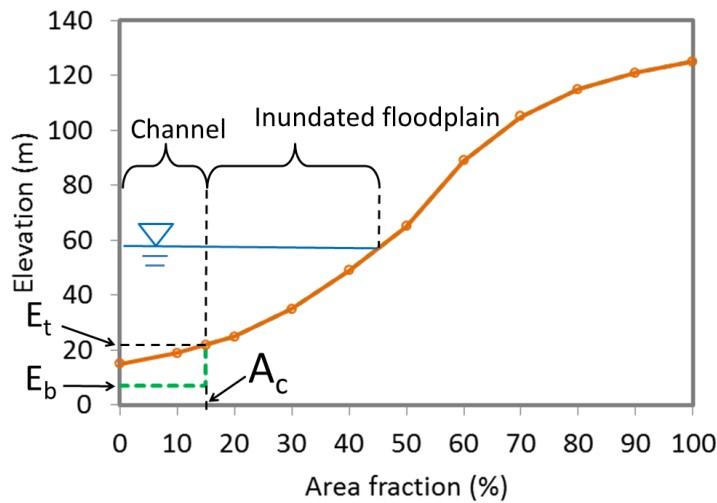

Figure 1. Illustrations of the macro-scale inundation scheme: (a) Illustration of river overflow; (b) Elevation profiles of a computation unit (e.g., a grid cell or subbasin). The brown solid line is the original elevation profile. The green dash line is the amended elevation profile (its non-channel part overlaps with the original elevation profile). $A_c$ is the fraction of the channel area in the computation unit; $E_t$ is the bank top elevation; and $E_b$ is the channel bed elevation.

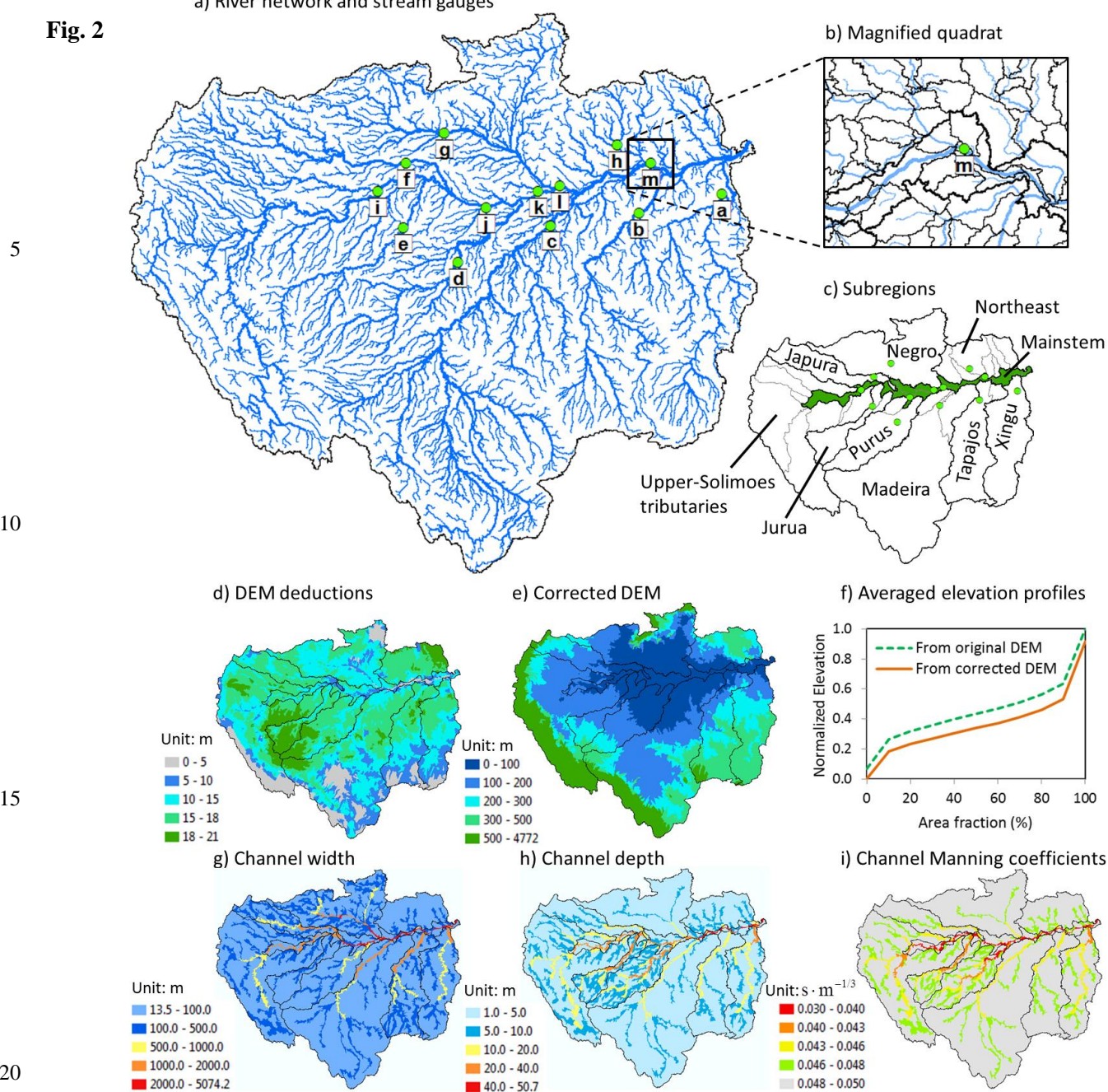

**Fig. 2**

a) River network and stream gauges

b) Magnified quadrat

c) Subregions

d) DEM deductions

e) Corrected DEM

f) Averaged elevation profiles

g) Channel width

h) Channel depth

i) Channel Manning coefficients

**Figure 2. Basin discretization and model inputs. (a) The river network extracted from the DEM overlaps with 13 stream gauges: a) Altamira; b) Itaituba; c) Fazenda vista alegre; d) Canutama; e) Gaviao; f) Acanaui; g) Serrinha; h) Cach da porteira-con; i) Santo antonio do ica; j) Itapeua; k) Manacapuru; l) Jatuarana+Careiro; m) Obidos. (b) Magnified quadrat. The thin (thick) black lines mark boundaries between subbasins (subregions). (c) Delineation of 10 subregions (including 9 tributary subregions and the mainstem subregion indicated by dark green color). (d) Average DEM deductions at each subbasin for alleviating vegetation-caused biases. (e) The corrected DEM. (f) Averaged elevation profiles based on the original and corrected DEMs. (g) Channel widths. (h) Channel depths. (i) Manning roughness coefficients of channels.**

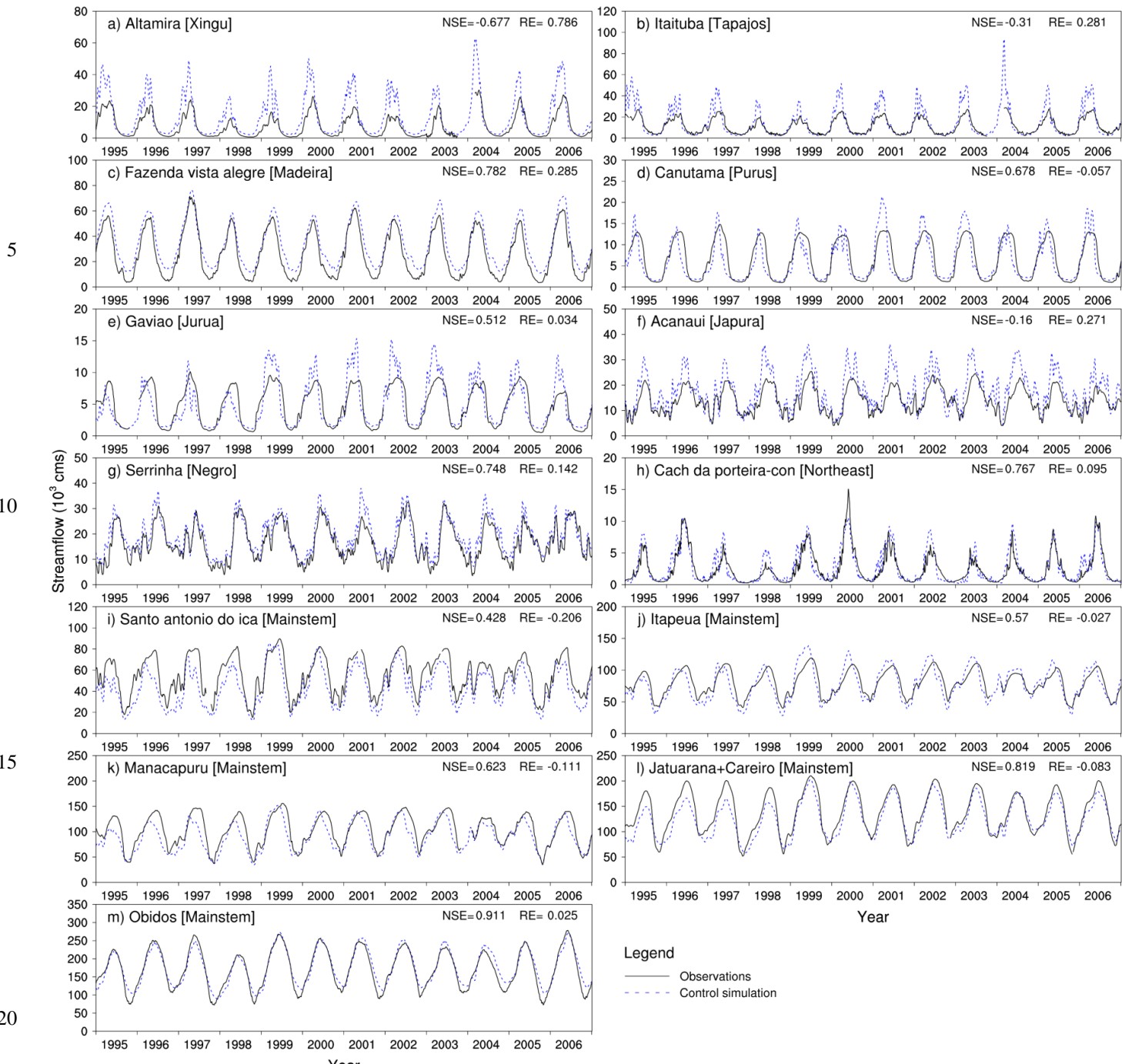

**Figure 3. Comparison between modeled and observed daily streamflow for a 12-year period (1995 – 2006) at 13 stream gauges (the corresponding subregion names are shown in the brackets): a) Altamira [Xingu]; b) Itaituba [Tapajos]; c) Fazenda vista alegre [Madeira]; d) Canutama [Purus]; e) Gaviao [Jurua]; f) Acanaui [Japura]; g) Serrinha [Negro]; h) Cach da porteira-con [Northeast]; i) Santo antonio do ica [Mainstem]; j) Itapeua [Mainstem]; k) Manacapuru [Mainstem]; l) Jatuarana+Careiro [Mainstem]; m) Obidos [Mainstem]. The Nash–Sutcliffe efficiency coefficient and the relative error of mean annual streamflow are indicated at the upper right corner of each panel. Figure 2a shows the stream gauge locations.**

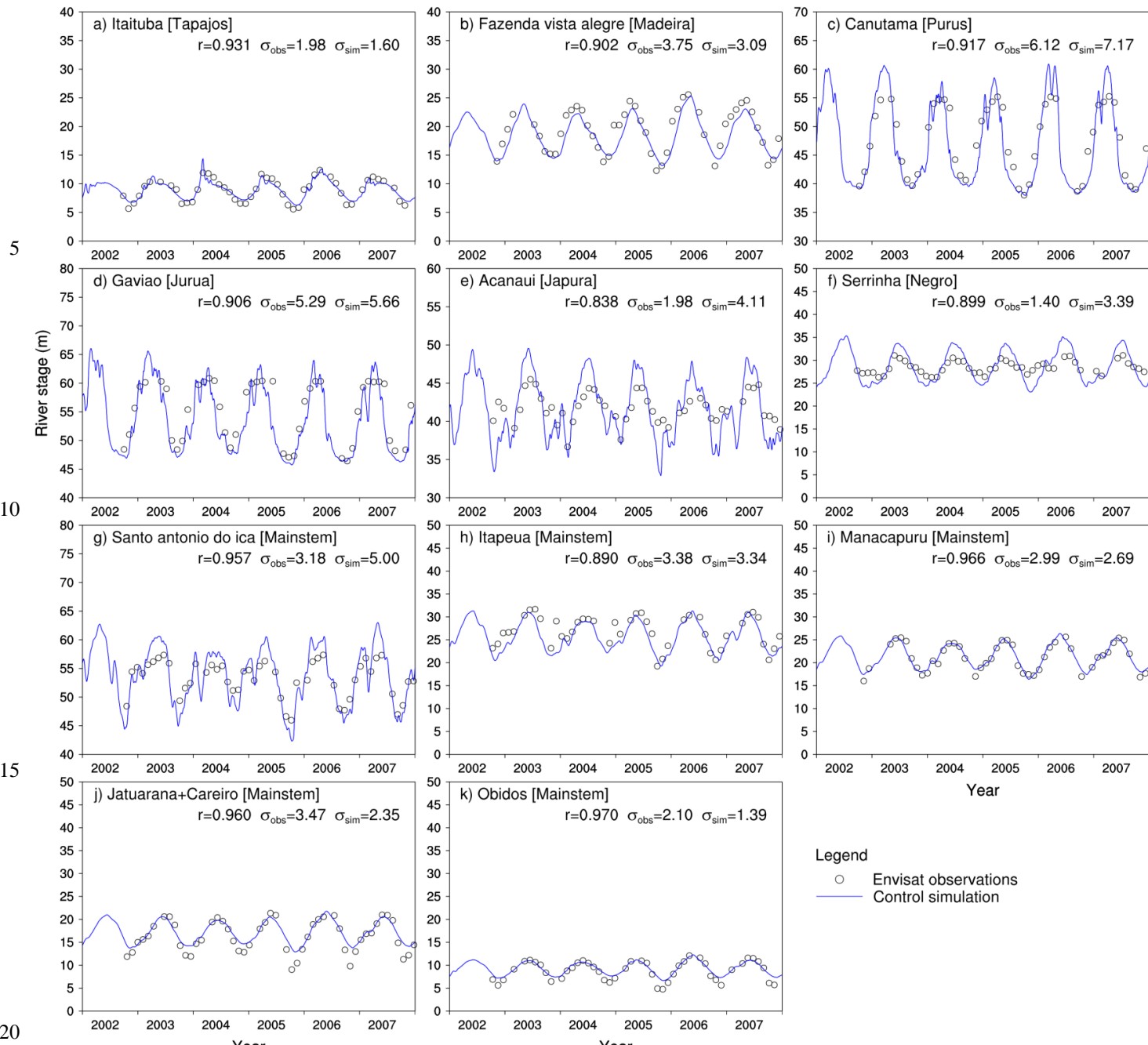

**Figure 4.** Comparison of modeled daily river stages with the observations for a 6-year period (2002 – 2007) at the subbasins containing or close to 11 of the 13 stream gauges (the corresponding subregion names are shown in the brackets): a) Itaituba [Tapajos]; b) Fazenda vista alegre [Madeira]; c) Canutama [Purus]; d) Gaviao [Jurua]; e) Acanaui [Japura]; f) Serrinha [Negro]; g) Santo antonio do ica [Mainstem]; h) Itapeua [Mainstem]; i) Manacapuru [Mainstem]; j) Jatuarana+Careiro [Mainstem]; k) Obidos [Mainstem]. The Pearson correlation coefficient between modeled river stages and the observations, as well as standard deviation for modeled and observed river stages, are indicated in each panel. The simulated river stages are shifted to coincide with the observations for better visual comparison (please see the Sect. 3.2 for the detailed explanation).

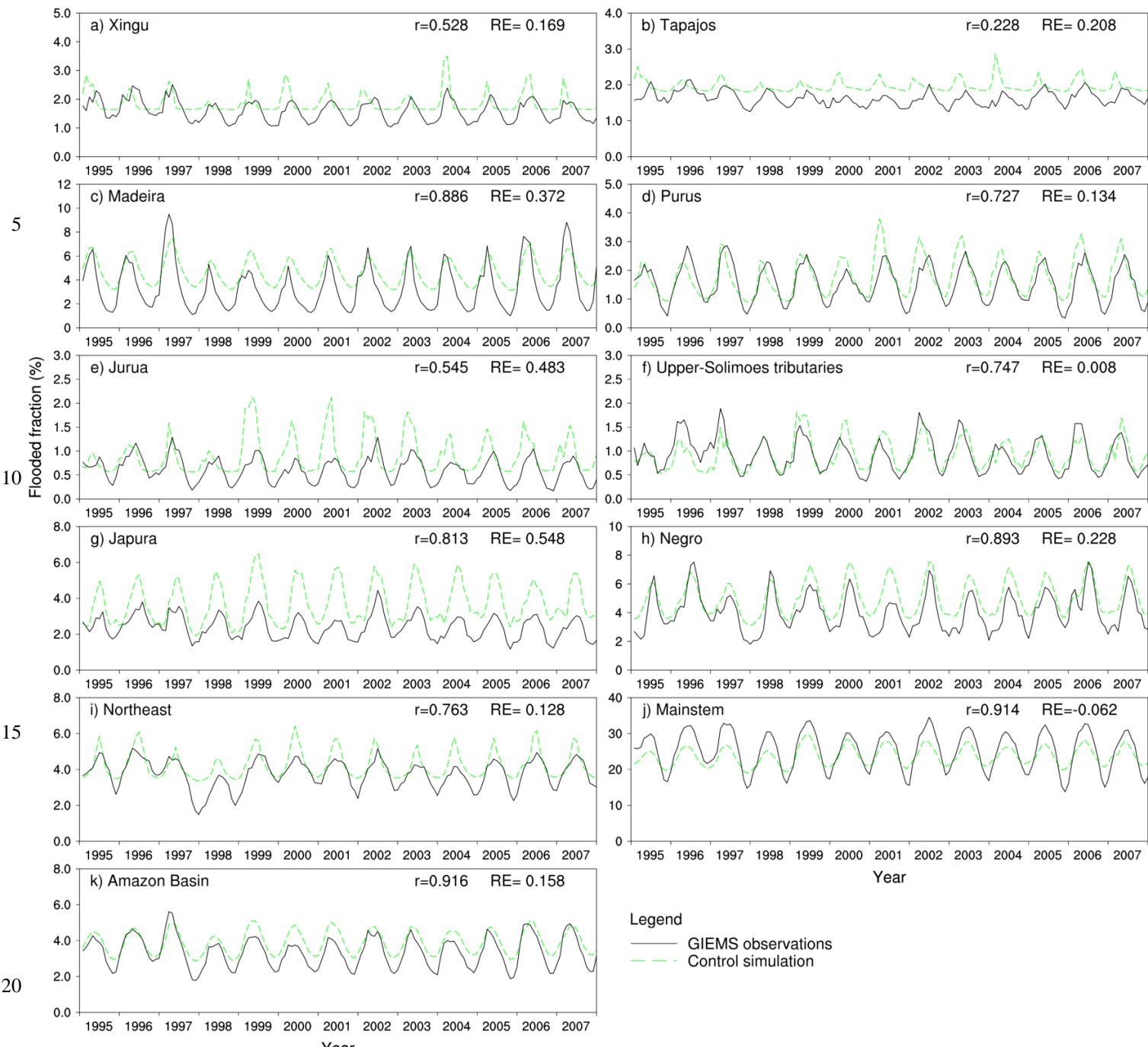

**Figure 5. Comparison of modeled monthly flood extent to the GIEMS satellite observations during a 13-year period (1995 – 2007) for 10 subregions and the entire Amazon Basin: a) Xingu; b) Tapajos; c) Madeira; d) Purus; e) Jurua; f) Upper-Solimoes tributaries; g) Japura; h) Negro; i) Northeast; j) Mainstem; k) Amazon Basin. The Pearson correlation coefficient between the modeled and observed monthly flood extent and the relative error of mean annual flood extent are indicated in each panel.**

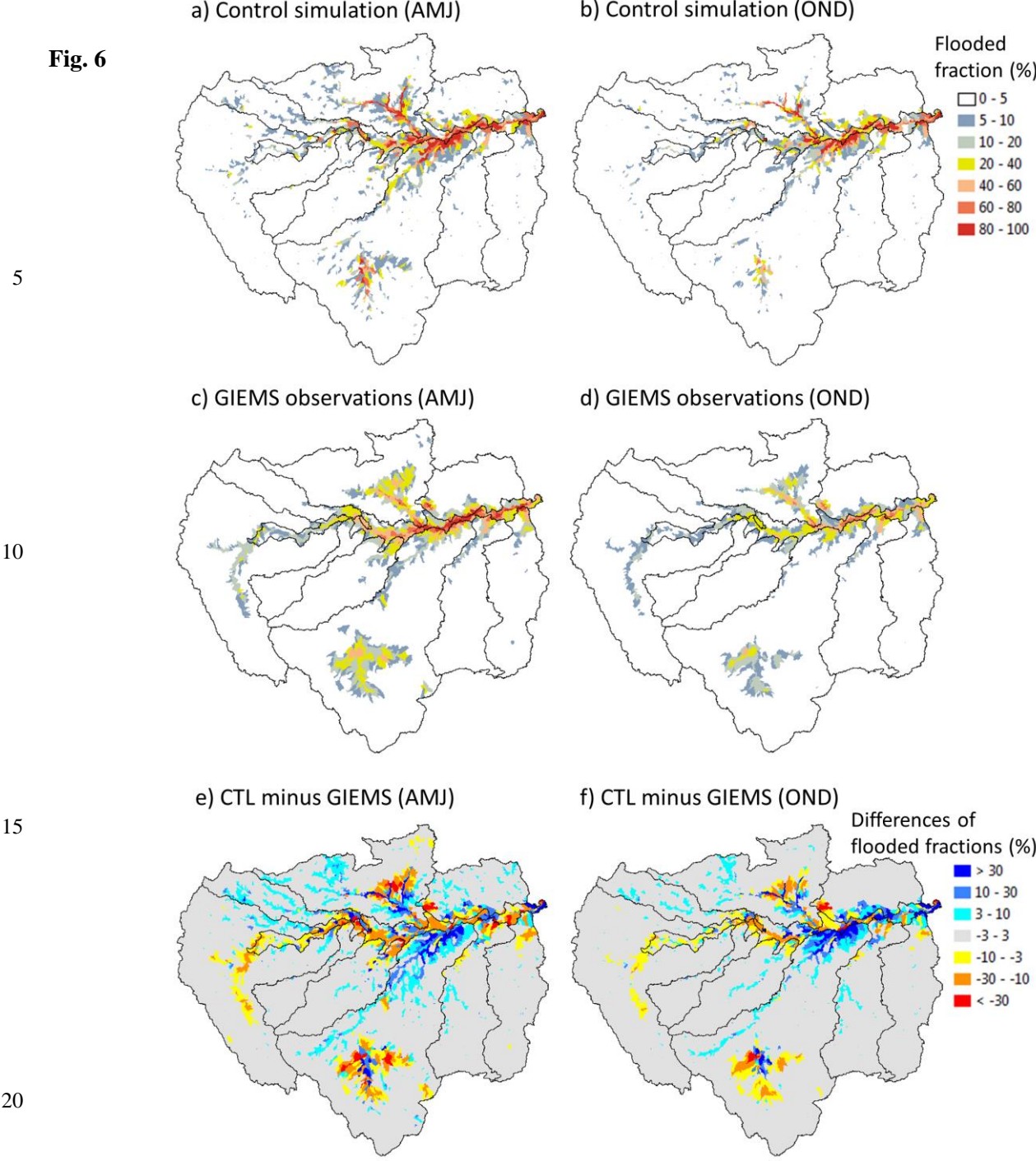

**Figure 6. Average spatial patterns of flooded fractions for all subbasins during 13 years (1995 – 2007): a) Results of the control simulation in the high-water season (AMJ – April, May and June); b) Results of the control simulation in the low-water season (OND – October, November and December); c) GIEMS observations in the high-water season; d) GIEMS observations in the low-water season; e) Differences between the control simulation and GIEMS observations in the high-water season; f) Differences between the control simulation and GIEMS observations in the low-water season.**

**Fig. 7**

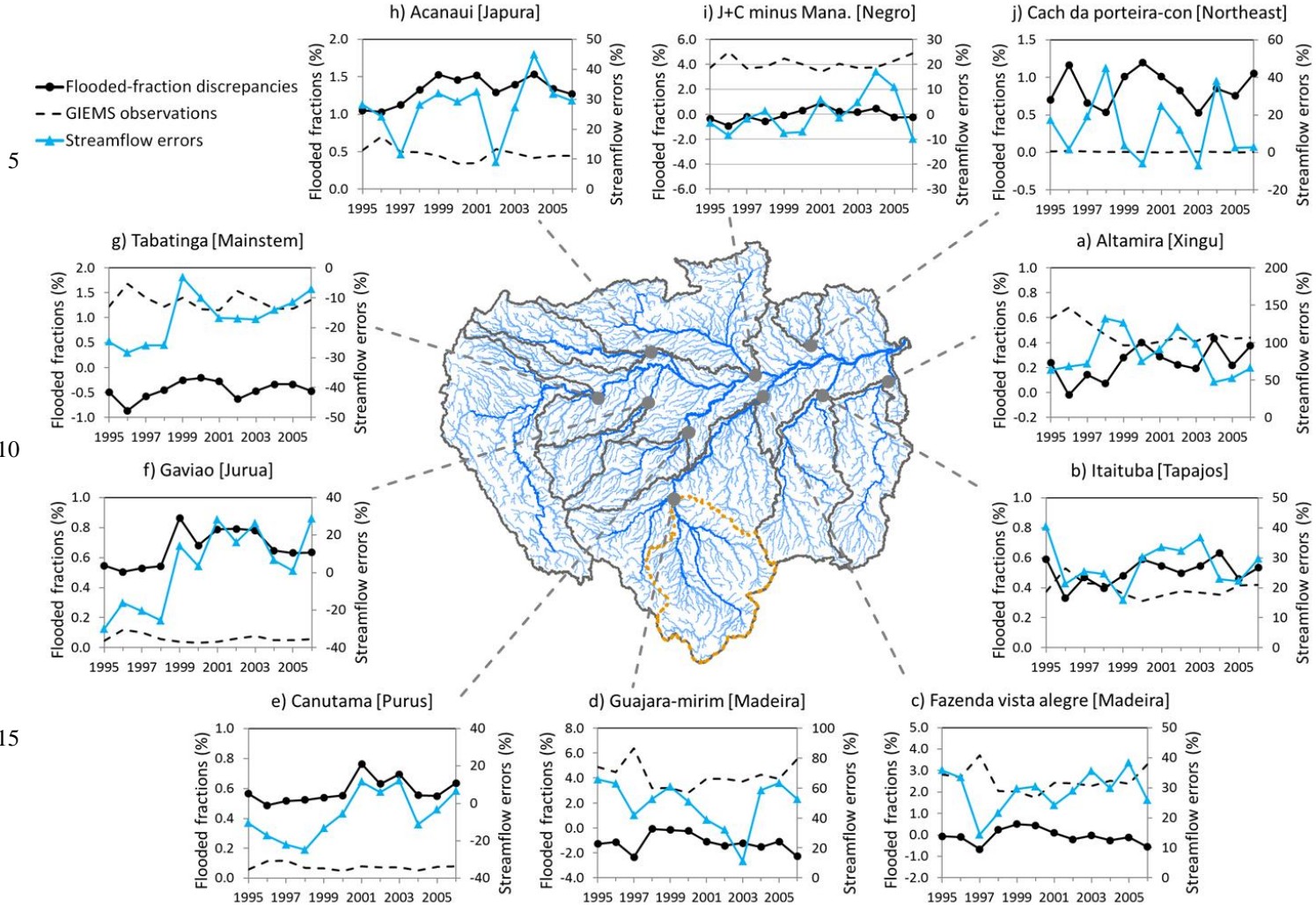

Figure 7. Streamflow errors and the flood extent discrepancies (i.e., the differences between simulated flood extent and the GIEMS data) in the area upstream of the gauge for 10 gauges at the annual scale during 12 years (1995 – 2006). Streamflow of the Negro subregion (panel (i) ) is approximated by the streamflow difference between the Jatuarana+Careiro gauge and the Manacapuru gauge. The upstream area of each gauge is enclosed by the gray lines (or brown dotted lines for the Guajara-mirim gauge) in the basin map.

**Fig. 8**

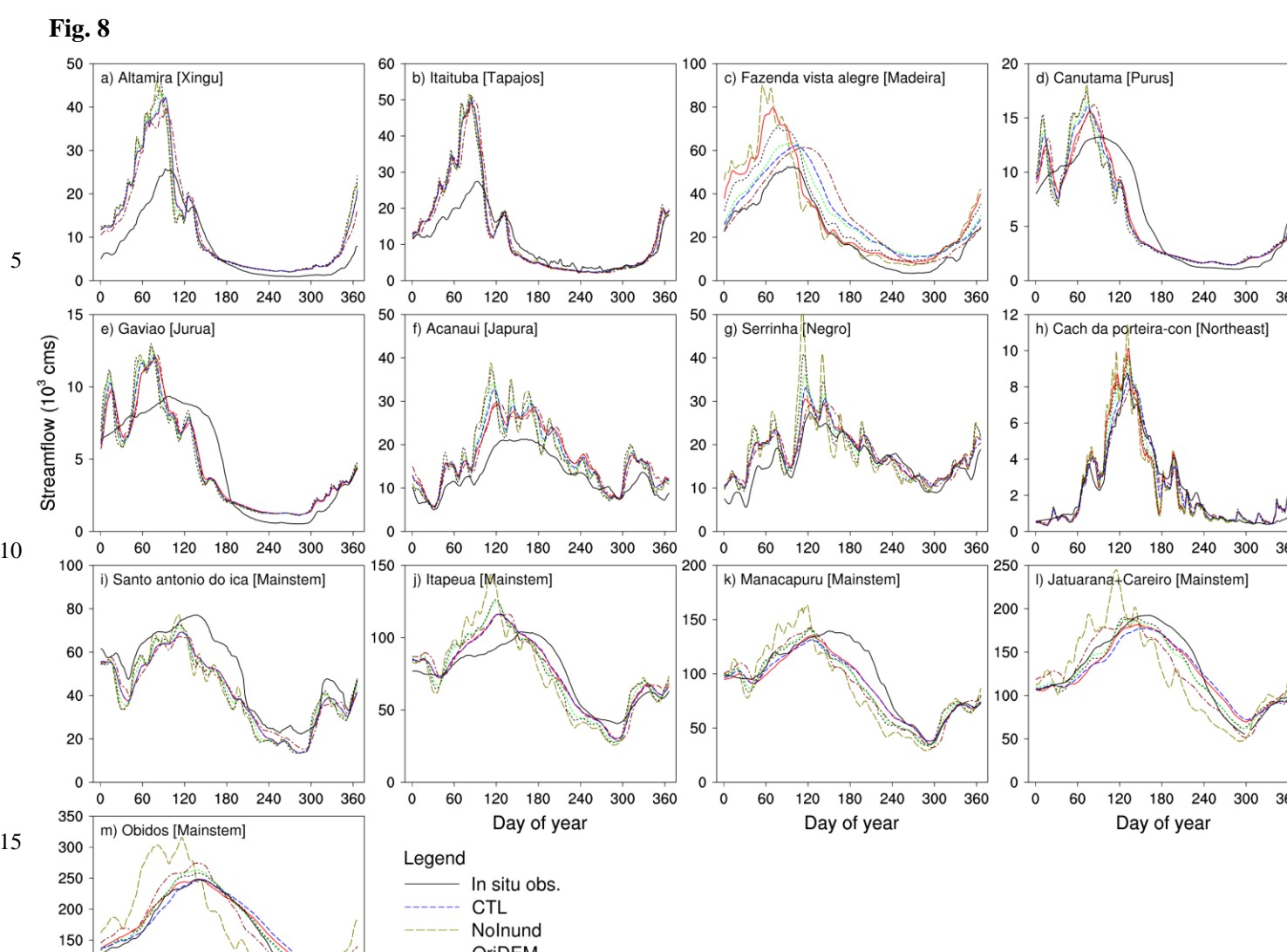

Figure 8. Observed and modeled daily streamflow of year 2005 at 13 stream gauges. Setup of the six simulations is described in Table 2: CTL – Control simulation; NoInund – Without inundation scheme; OriDEM – Using the original DEM (with vegetation-caused biases); OriSec – Using basin-wide channel geometry formulae; n003 – Using a uniform Manning roughness coefficient (i.e., 0.03) for all the channels; KW – Using kinematic wave method to represent river flow.

**Fig. 9**

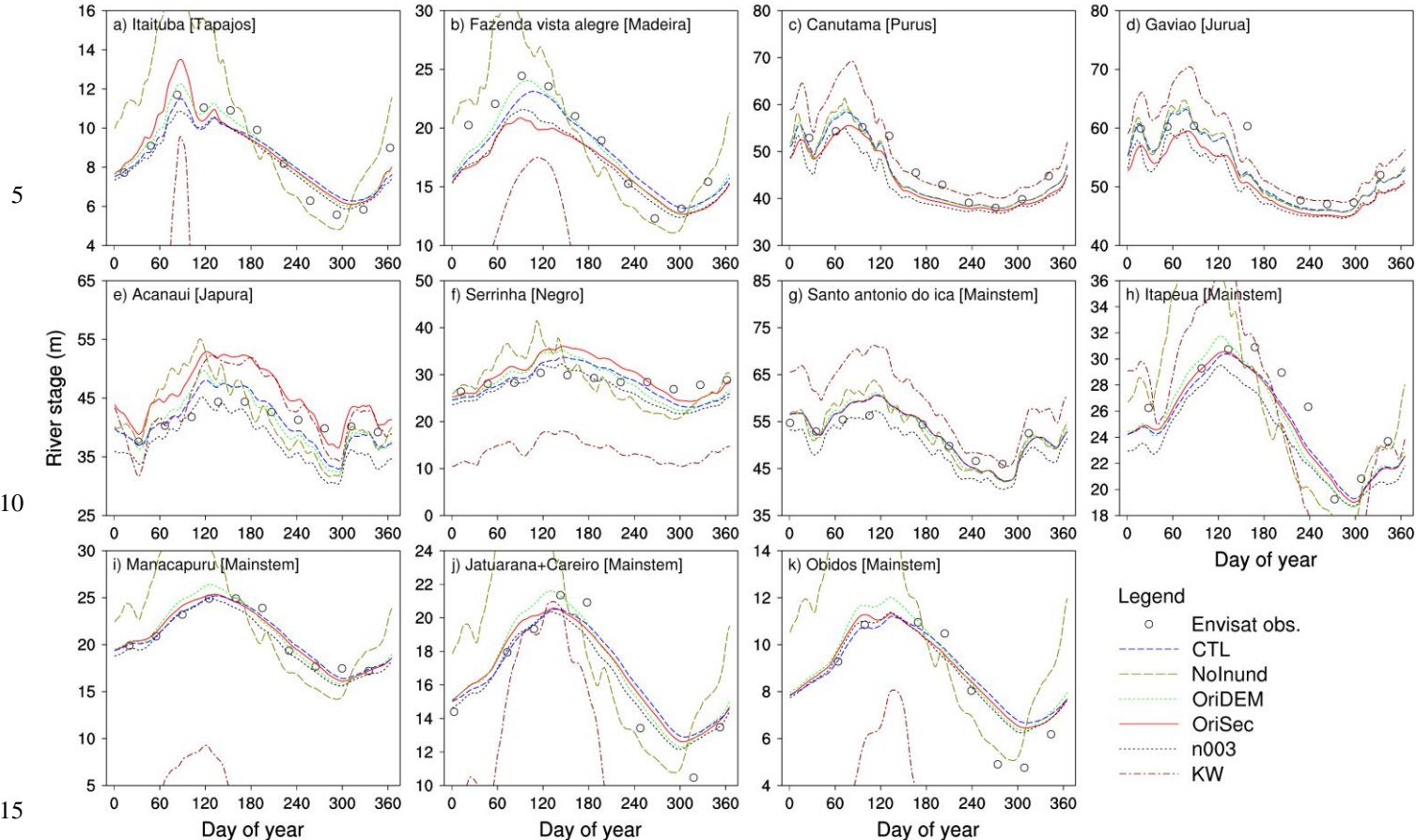

**Figure 9. Observed and modeled river stages at the daily scale in year 2005 for the subbasins containing or close to 11 of the 13 stream gauges. Setup of the six simulations is described in Table 2: CTL – Control simulation; NoInund – Without inundation scheme; OriDEM – Using the original DEM (with vegetation-caused biases); OriSec – Using basin-wide channel geometry formulae; n003 – Using a uniform Manning roughness coefficient (i.e., 0.03) for all the channels; KW – Using kinematic wave method to represent river flow.**

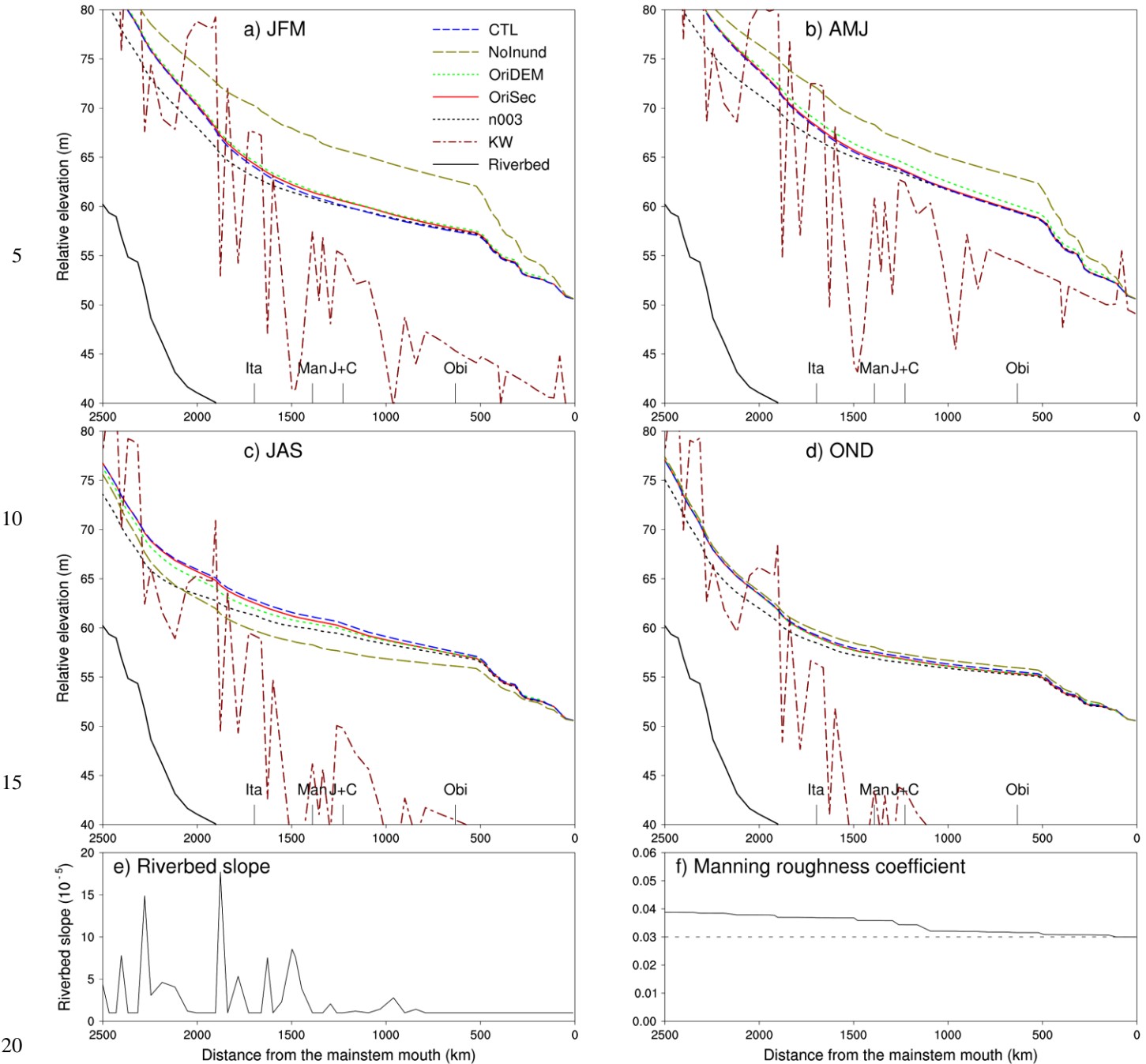

**Figure 10. Modeled average river surface profiles along the middle and lower mainstem in the four seasons of year 2005: (a) JFM (January, February and March; the period of rising flood); (b) AMJ (April, May and June; the period of high water); (c) JAS (July, August and September; the period of falling flood); and (d) OND (October, November and December; the period of low water). Results of six simulations are shown. The four stream-gauge locations are labeled on the x-axis: Ita – Itapeua; Man – Manacapuru; J+C – Jatuarana+Careiro; Obi – Obidos. Riverbed slopes (e) and Manning roughness coefficients (f) along the mainstem are also shown. In the panel (f), the solid curve shows spatially varying Manning coefficients used in five simulations; the dotted line shows the uniform Manning coefficient of 0.03 used in the simulation "n003".**

**Fig. 11**

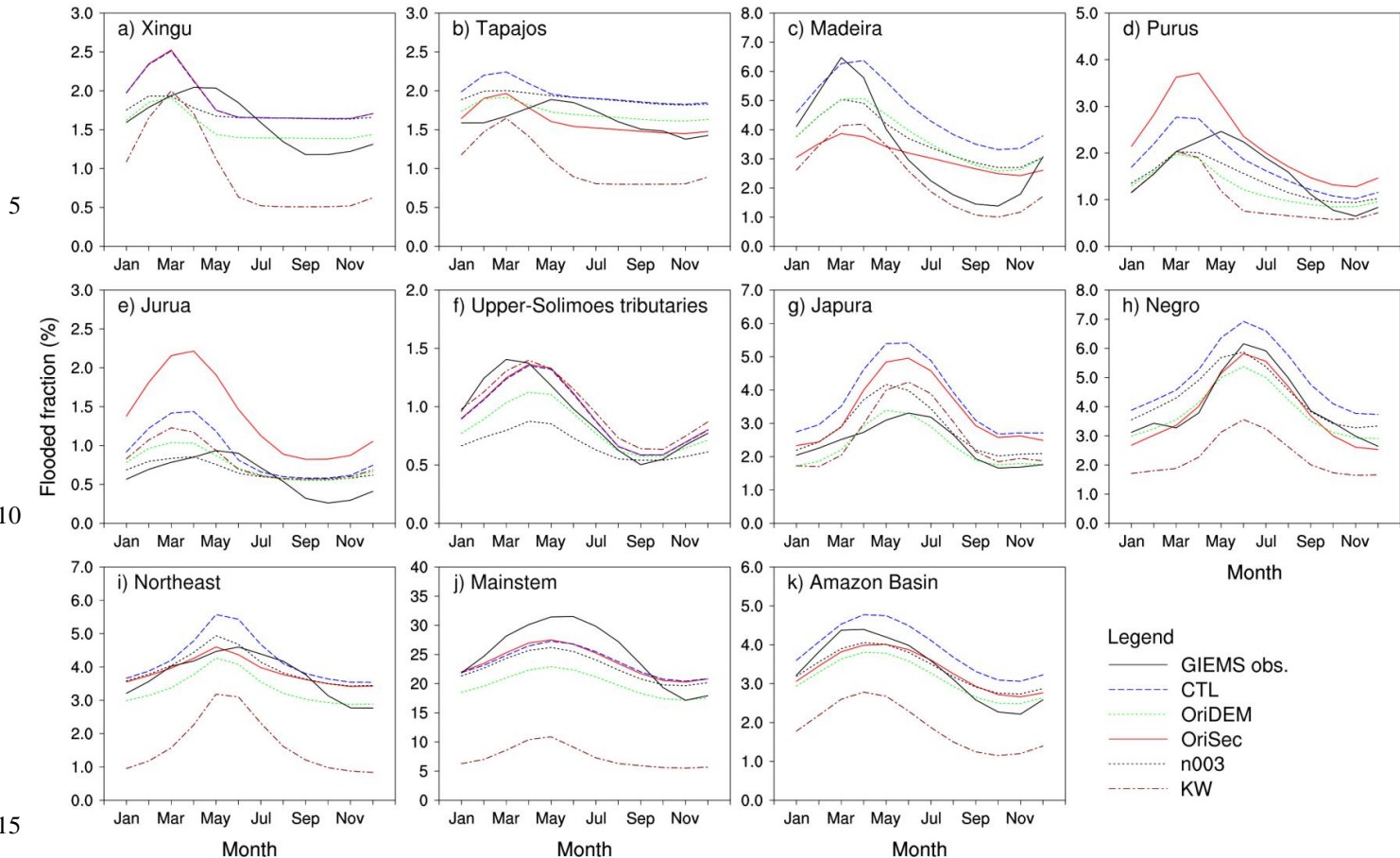

**Figure 11. Observed and modeled average monthly flood extent of 13 years (1995 – 2007) for the 10 subregions and the entire Amazon Basin. Setup of the five simulations is described in Table 2: CTL – Control simulation; OriDEM – Using the original DEM (with vegetation-caused biases); OriSec – Using basin-wide channel geometry formulae; n003 – Using a uniform Manning roughness coefficient (i.e., 0.03) for all the channels; KW – Using kinematic wave method to represent river flow.**

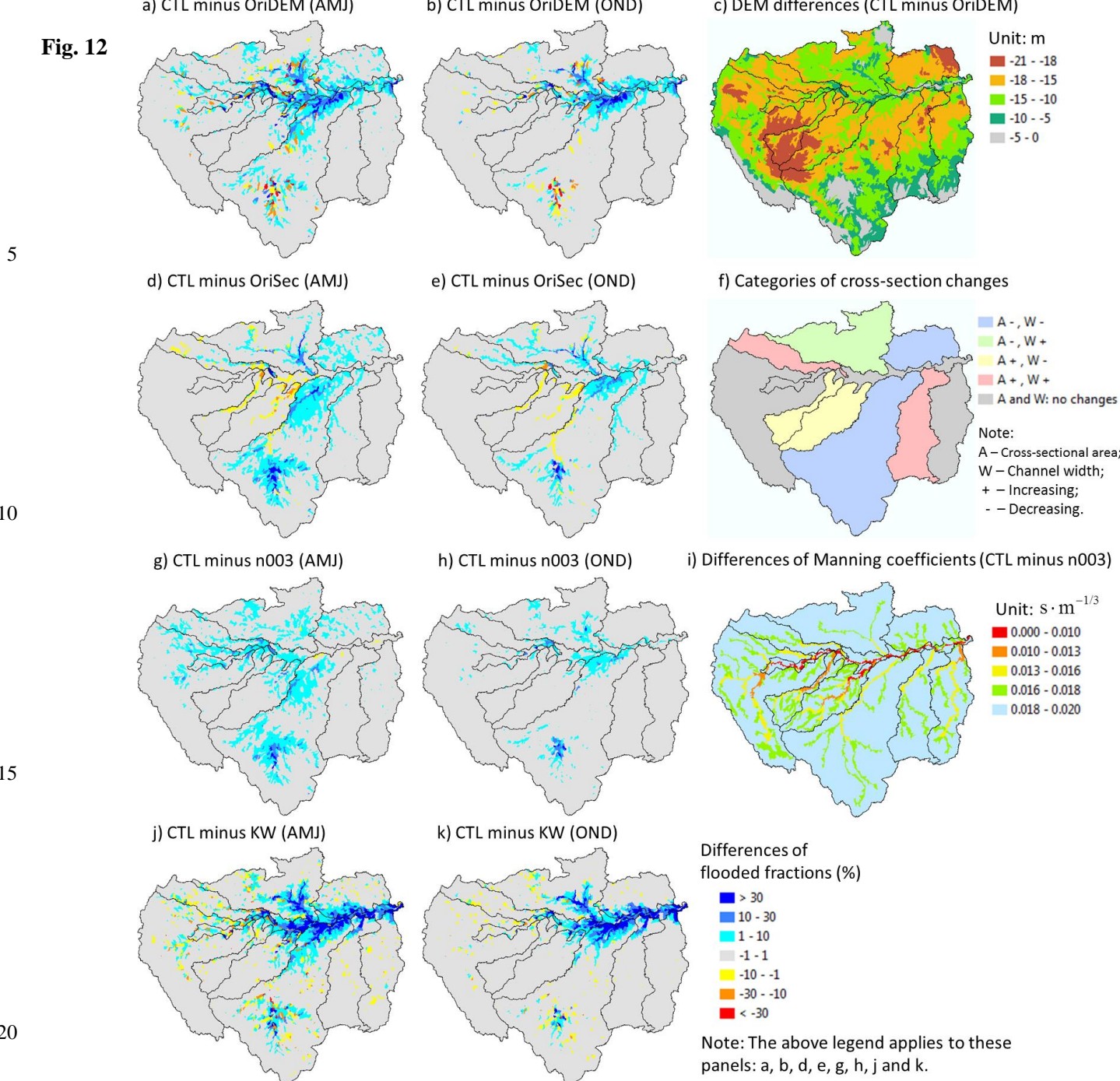

**Figure 12. Differences in subbasin flooded fractions averaged during 13 years (1995 – 2007) between the control simulation (CTL) and the four contrasting simulations (i.e., OriDEM, OriSec, n003 and KW) for the high-water season (AMJ – April, May and June) and low-water season (OND – October, November and December): (a) and (b): CTL minus OriDEM; (d) and (e): CTL minus OriSec; (g) and (h): CTL minus n003; (j) and (k): CTL minus KW. Panel (c) shows DEM differences (CTL minus OriDEM); Panel (f) shows categories of cross-section changes for the 10 subregions; Panel (i) shows Manning coefficient differences (CTL minus n003).**

**Fig. 13**

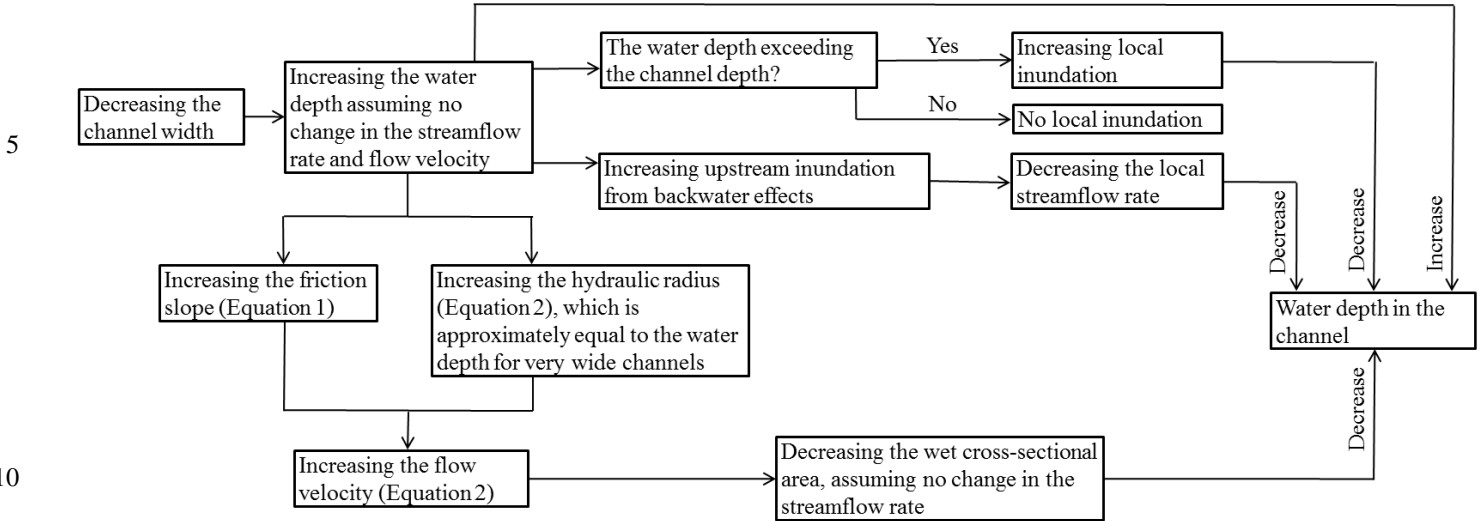

**Figure 13. A diagram illustrating that decreasing the width of the local channel could bring about changes in the water depth of the local channel through various mechanisms. In general the phenomena before and after an arrow have the cause – effect relationship.**