# Peer review of "Modeling surface water dynamics in the Amazon Basin using MOSART-Inundation-v1.0: Impacts of geomorphological parameters and river flow representation"

_Geoscientific Model Development, 2016_

## Referee Comment (RC1) · Anonymous Referee #1 · 14 Sep 2016

This manuscript describes the development of the MOSART river transport model to include a flood inundation scheme which was then tested across the Amazon basin. Excellent detail is given as to the setup of the model including the processing of the DEM and channel geometry parameters. The model is run for a time period longer than 20 years and evaluated against in-situ streamflow observations and remotely sensed satellite data of river stage and flood extent. Results from the evaluation showed good agreement in each of these aspects. A sensitivity analysis was then conducted to assess the impact of the DEM and channel geometry corrections, setting a uniform Manning's n and using a kinematic channel flow equation. Sensitivities were found in

each variable due to the influence they have upon the floodplain elevation, channel capacity and flow velocity.

The manuscript's contribution to model development is the inclusion of an inundation scheme to the MOSART model, however this is not explicitly stated until page 6 therefore leaving the reader unclear about the paper's contribution for most of the introductory sections. The authors should revise the abstract to state much more clearly that this is one major contribution of the manuscript.

One aim of the manuscript is to investigate the importance of geomorphic parameters and river flow representation when modelling the Amazon basin. This is done through the results of the sensitivity analysis, however these mostly back up results from previous papers which also describe the parameterisation of large scale river models in the Amazon basin. Therefore the novel contribution from this aspect is minimised, the greatest contribution from this paper is in describing the model development of the MOSART model.

There is no comparison between the results from the model developed in this manuscript with results from the previous version of the model without the inundation scheme. Clearly it is not possible to compare the results of inundation extent but for a model development paper there needs to be some direct comparison between the results of the developed model and those of its predecessor. In this case it should be possible to compare the results of streamflow and river stage. I believe that the model development in this manuscript is significant and merits eventual publication, however I would suggest that it is reconsidered after major revisions so that the authors can include results from a direct comparison between the two model versions.

Additionally please find below the following minor corrections:

Equation 2 define v

Page 7 line 3 how was it decided to combine the neighbouring catchments?

Page 8 line 9 should read 'lowered to 2.5m'

Page 8 line 9 why was a distinction made between shrubs which were over 5m and those which were lower - why the different treatment when correcting the SRTM?

Page 8 line 13 what was the uniform value that was subtracted from areas located outside the floodplain?

Page 7 line 15 were the elevation profiles not defined from the vegetation corrected DEM?

Page 9 line 13 how were the gauges distributed amongst the 10 regions? Some regions might have only had a few gauges hence the significance of the RMSE value might be low, plus this might override the significance of geomorphological factors in applying this correction

Page 10 line 16 give an example of the literature - a reference to a textbook for example

Page 11 line 15 - can river flow in the upper tributaries really be evaluated using the gauge at Santo Antonio do Ica which is located much further downstream? The steeper gradients of the tributaries are likely to have different flow hydraulics to that in the mainstem, can the authors comment on this and provide further justification for using this gauge to make the evaluation?

Page 11 line 22 there is a positive runoff bias in the Japura basin which goes against the overall trend of negative biases in the western portion of the basin, could the authors explain what may be causing this?

Eq 7 This describes how the simulated river stages are converted into elevations, should this not therefore be included in section 2.5 which describes how the river channel geometry in the model was established?

Page 12 line 12 how were the simulated river stages shifted to coincide with the observations?

Page 12 line 15 should another metric be calculated alongside the correlation coefficient? In the Negro and Japura basins for example Fig 4 shows there is a very high correlation but the differences between the simulations and observations are very large. Perhaps calculating another metric might capture this?

Page 13 line 2 should read 'lake areas'

Figure 6 the four plots should be replaced with two difference plots, one showing the difference between the simulated and observed during high water and the other during low water. This would better visualise the difference between the two simulations.

Page 13 line 13 the statement that the GIEMS data and simulation agree reasonably well is very vague. Figure 6 appears to show that the simulation overestimates the extent in the lowland portion of the basin, especially at low water. This sentence should be expanded to include more details about where the differences occur.

Figure 7 it could be useful to plot the data by seasons e.g AMJ, JAS, OND, JFM as this might show if the errors are concentrated in a particular season e.g. low water.

Figure 8, why does this figure refer to the average seasonal cycle from 1995-2006 whilst figures 9 and 10 refer to 2007 only? Does this explain why the results for the kinematic simulation are so different between figures 8 and 9 & 10? I would expect the kinematic simulation to be very different to the control simulation (as it appears in Figs 9 and 10), yet does this not appear to be the case for streamflow - could the authors explain why streamflow is not sensitive to the kinematic solution or replot Fig 8 for 2007 only so that it is directly comparable to Figs 9 and 10? Figure 10 is confusing with the y-axis reset for 0-1500 km for the simulations but not for the riverbed profile. These graphs should use the same y-axis for the entire river length in order to remove the confusing jump that happens at 1500 km.

Section 4.2, the greatest effect is shown in the Madeira basin, is this most likely because the multiplicative factor (0.36) has the greatest effect on changing the channel

[Figure]

geometry relative to the other basins? This should be stated more explicitly in the second paragraph.

Figure 13 needs to be redone as it is very difficult to follow the decision chain that the authors are trying to imply. For example at the second box there are four options but how is a reader meant to decide between these?

Page 18 line 26 should read 'could have an evident effect'
* * *

---

## Referee Comment (RC2) · Anonymous Referee #2 · 24 Oct 2016

The paper presents improvements on the parametrization of the MOSART surface water model. State of art methods are used to update river model and inundation parametrization. The model is evaluated in the Amazon basin and several simulations were performed to evaluate the role of the DEM, river geometry parameters, and backwater effects. The subject addressed by the paper is important. With new data available for regional/global hydrologic simulations, there are several new efforts to improve hydrological models. And the documentation of new improvements/updates of models, as the MOSART, fits the goal of GMD journal. Also, the study of impact of model errors and different parametrizations are important guide future model developments. The paper is generally clear. However, it seems that most of the conclusions from paper analyses were already provided by the past modelling studies in the Amazon (e.g. Paiva et al., 2013, Getirana et al. 2012, Yamazaki et al., 2011, Beighley et al., 2009; Baugh et al., 2013). For example, the past studies already pointed for the importance backwater effects and flooding, performed sensitivity studies on the role of river geometry errors and DEM errors on amazon simulations. So I guess that it would be better to present the paper as a documentation of the improvements of a specific model (MOSART) to move toward state of art methods. And to clarify that the analyses could reproduce similar conclusions from the past studies. So, as the documentation of model parametrizations fits the GMD journal scope, I think that the paper could be published. But it needs to be reviewed clarify the actual contributions, by addressing the comments above and below.

Introduction: I feel that the main goal of this paper should be to document improvements on the MOSART model. So it is important to provide more details in the intro section.

Page 2. Line 25. Which of these challenges were addressed by this paper in a novel way that was not done by the past efforts?

Page 3. Line 9. Vegetation errors from SRTM DEM were removed globally by F.E. O'Loughlin et al. 2016 RSE. Please review and discuss it in the paper.

Page 3. Line 22. See also analyses from Paiva et al., 2013 WRR.

Objectives. What is the new proposed contribution? If the contributions are limited to updating MOSART model with state of art methods, then I think that you should specify it in the objectives and introduce MOSART in the intro section.

2.1. How the model defines what is main river network and tributary subnetwork?

Eq.1. It seems g can be removed from equation... Continuity equation is not shown.. Please show it.

How these equations (kinematic and diffusive) are solved? Please provide details on the numerical methods. finite difference, finite volumes, implicit, explicit? Criteria for time step, spatial discretization? What is done to avoid mass errors. . ..

2.2. It is not clear how you compute river bed elevation? Is it simply the lowest DEM pixel of the catchment? How the model accounts for the fact that SRTM DEM does not see the river bed? And the fact that the river profile is not flat?

2.3. How the basins are defined? What is the input data? Hydrosheds? Please make it clear.

Pag. 6 Line 30. What is the criteria to define river length? How time step is defined? How these choices affect model errors (model numerical stability, mass errors, numerical dispersion,. . .) ? Please clarify and discuss it.

2.4. Vegetation Errors. Was the corrected DEM validated ? Please justify and compare these methods to the global SRTM DEM product free of veg errors recently developed by F.E. O'Loughlin et al. 2016 RSE.

2.6. Line 16. What literature was used to define Manning at 0.03 and 0.05?

I feel that the parametrization of Manning needs more justification (past studies or calibration). How these choices will impact model results?

2.7. Line 6. Why average manning of 0.03 ? You should use the average Manning from the reference simulation or use other approach to isolate the effect of variable vs constant manning.

2.7. What is optimal combination? Was any calibration performed?

3.1. How the model performance compare to past modelling studies in the Amazon? Please discuss it in the manuscript.

3.2. Equation 7. Do you use this equation to estimate a parameter for simulation? If yes, this explanation should appear o section 2. Why this approach were selected?

How it compares to previous studies? How this choice impact model results? See Paiva et al., 2013 for an analyses of impact of bed elevation errors on simulations.

3.2. How model performance for river elevation compares to previous modelling studies in the amazon? Please discuss it in the manuscript.

3.3. How model performance for flood extent compares to previous modelling studies in the amazon? Please discuss it in the manuscript.

4.1. How these analyses compare to previous analyses of impact of DEM and floodplains on Amazon simulations from previous modelling studies?

4.2. It is not change in channel storage capacity that changes simulation. It is changes in channel conductance capacity.

4.3. How these analyses compare to previous analyses of channel geometry from previous modelling studies?

4.3. I'm not sure if this analysis is conclusive. It is not possible to be sure that the differences in results are related to variable Manning or if it is because a specific value of 0.03 was chosen. This value may be different from the average value of the control simulation. I suggest the computation of the average Manning from control simulation and using this value for the new simulation.

4.3. How these analyses compare to previous analyses of Manning role from previous modelling studies?

Figure 10. This figure is confusing. It's hard to understand the break in the profile. Please review it.

4.4. Line 20. See also analyses on the importance of backwater effects for amazon simulations from Paiva et al., 2013 WRR and Paiva et al., 2013 Hyd.Process. Please compare and discuss in the manuscript.

Conclusions. Line 20. Review Yamazaki et al., 2013. WRR for discussion in Catchment

vs grid based simulations.

Conclusions: I'm not sure if there are new conclusions /findings that were not addressed by the past modelling studies in the Amazon (e.g. Paiva et al., 2013, Getirana et al. 2012, Yamazaki et al., 2011, Beighley et al., 2009; Baugh et al., 2013). The past studies already pointed for the importance backwater effects and flooding, performed Sensitivity studies on the role of river geometry errors and DEM errors on amazon simulations. It is important to recognize that the analyses from this paper only reproduced similar conclusions from the past studies. And also clarify that the new contribution from this paper is mostly on updating/improving the parametrization of an specific model, i.e. MOSART model by including improvements tested or suggested by the previous studies.

---

## Author Comment (AC1) · 27 Nov 2016

Response to Referee #1

We appreciate the time and effort of the reviewer and thank the reviewer for the constructive comments. We provided replies to all the comments, and will include the corresponding changes in the revised manuscript.

In the following text, we use blue color for the reviewer comments, black color for the replies, and italics for the revisions in the manuscript.

**Comments**

[C1-1]

This manuscript describes the development of the MOSART river transport model to include a flood inundation scheme which was then tested across the Amazon basin. Excellent detail is given as to the setup of the model including the processing of the DEM and channel geometry parameters. The model is run for a time period longer than 20 years and evaluated against in-situ streamflow observations and remotely sensed satellite data of river stage and flood extent. Results from the evaluation showed good agreement in each of these aspects. A sensitivity analysis was then conducted to assess the impact of the DEM and channel geometry corrections, setting a uniform Manning's n and using a kinematic channel flow equation. Sensitivities were found in each variable due to the influence they have upon the floodplain elevation, channel capacity and flow velocity.

The manuscript's contribution to model development is the inclusion of an inundation scheme to the MOSART model, however this is not explicitly stated until page 6 therefore leaving the reader unclear about the paper's contribution for most of the introductory sections. The authors should revise the abstract to state much more clearly that this is one major contribution of the manuscript.

Reply: We thank the reviewer for the positive evaluation and suggestion. In the Abstract and Introduction of the revised manuscript, we will more clearly state the goal of our study as mainly to incorporate and document an inundation scheme in the MOSART model, which is used in Earth System Models.

[C1-2]

One aim of the manuscript is to investigate the importance of geomorphic parameters and river flow representation when modelling the Amazon basin. This is done through the results of the sensitivity analysis, however these mostly back up results from previous papers which also describe the parameterisation of large scale river models in the Amazon basin. Therefore the novel contribution from this aspect is minimised, the greatest contribution from this paper is in describing the model development of the MOSART model.

Reply: In the initial manuscript, the comparisons between our study and previous studies were not clear. We will add more discussions to compare the results of our study with those of former studies. On one hand, our work was based on the important foundation of previous studies; on the other hand, our study had some new points in terms of methodologies, model results and sensitivity analyses.

While our modeling approach and improvements do not differ conceptually from those already explored in previous studies, we attempted to generalize various methods for application over the entire Amazon Basin, which is important as MOSART is used in global Earth System Models. Our study also provided more comprehensive evaluation of our simulations and analysis of sensitivity of the simulations to various factors, which yielded some findings that have not been discussed in former studies.

[C1-3]

There is no comparison between the results from the model developed in this manuscript with results from the previous version of the model without the inundation scheme. Clearly it is not possible to compare the results of inundation extent but for a model development paper there needs to be some direct comparison between the results of the developed model and those of its predecessor. In this case it should be possible to compare the results of streamflow and river stage. I believe that the model development in this manuscript is significant and merits eventual publication, however I would suggest that it is reconsidered after major revisions so that the authors can include results from a direct comparison between the two model versions.

Reply: We thank the reviewer for the suggestion and positive evaluation.

We have conducted a new simulation "NoInund" (with the inundation scheme turned off) and compared its results with those of the control simulation "CTL" (where the inundation scheme was turned on). This comparison revealed the impacts of the inundation scheme on modeled streamflow and river stages. The comparison will be presented in a new subsection (Section 4.1 Inundation representation). In brief, we will update Figures 8, 9, and 10 to include the results of "NoInund" for comparison with simulations that include the inundation scheme in different sensitivity experiments. The three figures are attached in

the Appendix at the end of this Response. In brief, including inundation generally improves the simulation of streamflow and river stages compared to the simulation without the inundation parameterization.

**Comments**

Additionally please find below the following minor corrections:

[C1-4]

Equation 2 define v

Reply: The definition of $v$ will be added (below the newly supplemented Equation (1)).

[C1-5]

Page 7 line 3 how was it decided to combine the neighbouring catchments?

Reply: The number of catchments is comparatively large, so to show the inundation results more concisely, the catchments were combined to a few subregions. This is done by selecting seven major catchments (i.e., Xingu, Tapajos, Madeira, Purus, Jurua, Japura and Negro) as subregions or the major part of a subregion. Then the Upper-Solimoes catchments were combined as one subregion, and the northeast catchments were combined as another subregion. Lastly, the remaining five catchments were combined with their adjacent subregions.

More explanation will be added in this paragraph.

[C1-6]

Page 8 line 9 should read 'lowered to 2.5m'

Reply: The original manuscript is not clear. There should be "an amount of" added before "2.5 m". However, this sentence will be removed in the revision (please see the reply to the comment below).

[C1-7]

Page 8 line 9 why was a distinction made between shrubs which were over 5m and those which were lower - why the different treatment when correcting the SRTM?

Reply: The original manuscript is not clear. Some sentences will be revised.

The resolution of the vegetation height data is coarser than the resolution of the land cover data. Hence within one pixel of the vegetation height data, there may be more than one land cover class, which should not be assigned the same vegetation height. For shrub, any vegetation height larger than 5 m should be an overestimation (according to Junk et al. 2011), so an upper limit of 5 m is imposed. After this correction, 50% of the vegetation height was deducted from the DEM pixel covered by shrubs.

The text will be revised as: *"In the high resolution land cover dataset, shrubs were defined to be less than 5 m tall (Junk et al., 2011). So for DEM pixels with shrub, the vegetation height is determined by the vegetation height data, but with an upper limit of 5 m. After this correction, the elevations were lowered by 50% of the vegetation heights for shrub DEM pixels."*

[C1-8]

Page 8 line 13 what was the uniform value that was subtracted from areas located outside the floodplain?

Reply: The original manuscript is not clear. For the fine DEM pixels within one coarse vegetation height pixel, a unique vegetation height is used, but for different vegetation height pixels, the vegetation height can be different even for the same vegetation class.

The text will be revised as: *"a uniform vegetation height was applied for all the DEM pixels within each vegetation height pixel".*

[C1-9]

Page 7 line 15 were the elevation profiles not defined from the vegetation corrected DEM?

Reply: Yes, the elevation profiles were generated from the vegetation corrected DEM.

For clarification, the text will be revised as: *"the void-filled DEM after the correction of vegetation-caused biases was used to generate the elevation profiles of all the subbasins".*

[C1-10]

Page 9 line 13 how were the gauges distributed amongst the 10 regions? Some regions might have only had a few gauges hence the significance of the RMSE value might be low, plus this might override the significance of geomorphological factors in applying this correction

Reply: The coefficients of the basin-wide channel geometry formulae were adjusted for seven of the 10 subregions (except "Xingu", "Upper-Solimoes tributaries" and "Mainstem"; shown in Table 1). Each of the seven subregions used 3 – 13 gauges.

The channel geometry is important for inundation modeling of the "Madeira" and "Negro" subregions which have evident inundation and large area. The "Madeira" and "Negro" subregions used 12 and 13 gauges, respectively.

[C1-11]

Page 10 line 16 give an example of the literature - a reference to a textbook for example

Reply: The text will be revised as follows:

"*Following Getirana et al. (2012), $n_{max}$ and $n_{min}$ were set as 0.05 and 0.03, respectively. In addition, a few other studies of the Amazon Basin adopted a similar range of values between 0.03 and 0.05 for the Manning coefficient (Beighley et al., 2009; Paiva et al., 2013a; Yamazaki et al., 2011).*"

[C1-12]

Page 11 line 15 - can river flow in the upper tributaries really be evaluated using the gauge at Santo Antonio do Ica which is located much further downstream? The steeper gradients of the tributaries are likely to have different flow hydraulics to that in the mainstem, can the authors comment on this and provide further justification for using this gauge to make the evaluation?

Reply: We agree that the river flow in the tributaries could be quite different from that of the mainstem so the river flow in the tributaries cannot be represented by using results at this gauge. Our description in the original manuscript is not accurate so the sentence will be revised as follows:

"*Most of this subregion is controlled by the Santo antonio do ica gauge at the upper mainstem.*"

[C1-13]

Page 11 line 22 there is a positive runoff bias in the Japura basin which goes against the overall trend of negative biases in the western portion of the basin, could the authors explain what may be causing this?

Reply: There is a negative runoff bias in the subregion "Upper-Solimoes tributaries" which is on the west side of the Japura basin (Fig. 3i). On the other hand, there is a positive runoff bias in the western

part of the subregion "Negro", which is on the east side of the Japura basin (Fig. 3g). The western Negro and the Japura basin are adjacent, and both have positive runoff biases.

The runoff biases could be due to errors in precipitation inputs or errors in the land surface water fluxes calculated by the land surface model (e.g., canopy evaporation, plant transpiration, and soil evaporation).

The following sentence will be added:

*"The runoff biases could be caused by errors in the precipitation forcing or errors in the land surface water fluxes calculated by the land surface model (e.g., canopy evaporation, plant transpiration, and soil evaporation)."*

[C1-14]

Eq 7 This describes how the simulated river stages are converted into elevations, should this not therefore be included in section 2.5 which describes how the river channel geometry in the model was established?

Reply: Following the suggestion by the reviewer, the method for estimating the riverbed elevation will be moved to Section 2.5.

[C1-15]

Page 12 line 12 how were the simulated river stages shifted to coincide with the observations?

Reply: All the simulated river stages of the same subbasin were raised or lowered by a uniform height, to facilitate comparison of the timing and magnitude between the simulated river stages and the observations. A similar method was also used in Figure 7 of Coe et al. (2002) .

The text will be supplemented by *"of the same subbasin"* and *"by a uniform height"*, and now read: *"For better visual comparison, the simulated river stages of the same subbasin were shifted by a uniform height to coincide with the observations."*

[C1-16]

Page 12 line 15 should another metric be calculated alongside the correlation coefficient? In the Negro and Japura basins for example Fig 4 shows there is a very high correlation but the differences between the simulations and observations are very large. Perhaps calculating another metric might capture this?

Reply: The original manuscript was not clear. The standard deviations were calculated and used to indicate river stage fluctuations. It was discussed that the river stage fluctuations were overestimated for the subbasins of 4 gauges (i.e., Canutama[Purus], Acanaui[Japura], Serrinha[Negro] and Santo antonio do ica[Mainstem] ).

To make the text more clear, the phrase "*as well as standard deviation for simulated and observed river stages*" will be replaced with "*Moreover, the standard deviations for the simulated and observed river stages were also calculated*."

[C1-17]

Page 13 line 2 should read 'lake areas'

Reply: This will be corrected.

[C1-18]

Figure 6 the four plots should be replaced with two difference plots, one showing the difference between the simulated and observed during high water and the other during low water. This would better visualise the difference between the two simulations.

Reply: Fig. 6 will be supplemented by two panels showing the differences during high water season (Fig. 6e) and low water season (Fig. 6f). The original four panels were kept in order to show the spatial patterns of inundation. Please find the revised Fig. 6 in the Appendix of this response.

[C1-19]

Page 13 line 13 the statement that the GIEMS data and simulation agree reasonably well is very vague. Figure 6 appears to show that the simulation overestimates the extent in the lowland portion of the basin, especially at low water. This sentence should be expanded to include more details about where the differences occur.

Reply: We will add some discussions of the differences between model results and GIEMS data in the revised manuscript.

[C1-20]

Figure 7 it could be useful to plot the data by seasons e.g AMJ, JAS, OND, JFM as this might show if the errors are concentrated in a particular season e.g. low water.

Reply: Thank you for the suggestion. Similar to Figure 7 that compares the annual averaged biases in flood extent and streamflow, we plotted the averaged monthly streamflow errors and the flood extent discrepancies (i.e., the differences between simulated flood extent and the GIEMS data) during 12 years (1995 – 2006). Please find the figure at the end of the Appendix. This figure shows that the seasonal distribution of streamflow errors varies for different gauges. For example, for "(a) Altamira" and "(b) Itaituba", evident positive biases occur from January to April; for "(c) Fazenda vista alegre" and "(d) Guajara-mirim", positive biases are more evident from about May to October; for "(h) Acanaui", positive biases are more evident from about March to July. Except for three subregions (Negro, Cach de porteira-con, and Tabatinga), the seasonality of flood extent discrepancies follows the seasonality of streamflow errors very closely, indicating the important contribution of streamflow errors to flood extent biases on seasonal time scale.

[C1-21]

Figure 8, why does this figure refer to the average seasonal cycle from 1995-2006 whilst figures 9 and 10 refer to 2007 only? Does this explain why the results for the kinematic simulation are so different between figures 8 and 9 & 10? I would expect the kinematic simulation to be very different to the control simulation (as it appears in Figs 9 and 10), yet does this not appear to be the case for streamflow - could the authors explain why streamflow is not sensitive to the kinematic solution or replot Fig 8 for 2007 only so that it is directly comparable to Figs 9 and 10?

Reply: Figs 8, 9 and 10 have been replotted to show the results of the same year. We did not have observed streamflow data for year 2007 so we plotted the results of another year (2005).

The three figures show that the differences in streamflow between the simulation KW (kinematic wave method) and the simulation CTL (diffusion wave method) are not as evident as those in river stages. Previous studies have yielded similar results (e.g., Yamazaki et al. 2011: Fig. 5). The reason could be that the flow velocities in the simulation KW (which are based on riverbed slopes) are also quite different from those in the simulation CTL (which are based on friction slopes). Please see the revised figures in the Appendix. Note that as mentioned in our reply to [C1-3], these figures have also been updated to include the results from a simulation without the inundation parameterization.

[C1-22]

Figure 10 is confusing with the y-axis reset for 0-1500 km for the simulations but not for the riverbed profile. These graphs should use the same y-axis for the entire river length in order to remove the confusing jump that happens at 1500 km.

Reply: Figure 10 has been replotted to use the same y-axis for the entire river length and will be updated in the revised manuscript. We thank the reviewer for the suggestion. Please find the revised Fig. 10 in the Appendix of this response.

[C1-23]

Section 4.2, the greatest effect is shown in the Madeira basin, is this most likely because the multiplicative factor (0.36) has the greatest effect on changing the channel geometry relative to the other basins? This should be stated more explicitly in the second paragraph.

Reply: Yes, the reason is that the channel geometry changes in the Madeira subregion are larger than those of the other subregions.

To make the discussion more explicit, the following revisions will be made in the second paragraph:

(1) "*channel geometry changes*" was replaced with "*the channel cross-sectional area is multiplied by a factor of 0.36 (Table 1)*" ;

(2) Added one sentence: "*Similar phenomenon is observed at the gauge "Cach da porteira-con" in the Northeast subregion (Fig. 8h), where the channel cross-sectional area is multiplied by a factor of 0.48.*"

(3) Added "*caused by refining channel geometry*" after "*Inundation changes*".

[C1-24]

Figure 13 needs to be redone as it is very difficult to follow the decision chain that the authors are trying to imply. For example at the second box there are four options but how is a reader meant to decide between these?

Reply: The original manuscript was not clear. In general, the phenomena before and after an arrow have the cause-effect relationship.

The figure caption was revised: "*An example of the effects of channel cross-sectional geometry on the water depth of the local channel*" was replaced with "*A diagram illustrating how decreasing the width of*

*the local channel could bring about changes in the water depth of the local channel through various mechanisms. In general the phenomena before and after an arrow have the cause-effect relationship*" .

Page 18 line 26 should read 'could have an evident effect'

Reply: "*an*" was added between "*have*" and "*evident*".

**Appendix**

[Figure]

**Figure 6. Average spatial patterns of flooded fractions for all subbasins during 13 years (1995 – 2007): a) Results of the control simulation in the high-water season (AMJ – April, May and June); b) Results of the control simulation in the low-water season (OND – October, November and December); c) GIEMS observations in the high-water season; d) GIEMS observations in the low-water season; e) Differences between the control simulation and GIEMS observations in the high-water season; f) Differences between the control simulation and GIEMS observations in the low-water season.**

[Figure]

**Figure 8. Observed and modeled daily streamflow of year 2005 at 13 stream gauges. Setup of the six simulations is described in Table 2: CTL – Control simulation; NoInund – Without inundation scheme; OriDEM – Using the original DEM (with vegetation-caused biases); OriSec – Using basin-wide channel geometry formulae; n003 – Using a uniform Manning roughness coefficient (i.e., 0.03) for all the channels; KW – Using kinematic wave method to represent river flow.**

[Figure]

**Figure 9. Observed and modeled river stages at the daily scale in year 2005 for the subbasins containing or close to 11 of the 13 stream gauges. Setup of the six simulations is described in Table 2: CTL – Control simulation; NoInund – Without inundation scheme; OriDEM – Using the original DEM (with vegetation-caused biases); OriSec – Using basin-wide channel geometry formulae; n003 – Using a uniform Manning roughness coefficient (i.e., 0.03) for all the channels; KW – Using kinematic wave method to represent river flow.**

[Figure]

**Figure 10. Modeled average river surface profiles along the mainstem in the four seasons of year 2005: (a) JFM (January, February and March; the period of rising flood); (b) AMJ (April, May and June; the period of high water); (c) JAS (July, August and September; the period of falling flood); and (d) OND (October, November and December; the period of low water). Results of six simulations are shown. The four stream-gauge locations are labeled on the x-axis: Ita – Itapeua; Man – Manacapuru; J+C – Jatuarana+Careiro; Obi – Obidos. Riverbed slopes (e) and Manning roughness coefficients (f) along the mainstem are also shown. In the panel (f), the solid curve shows spatially varied Manning coefficients used in five simulations; the dotted line shows the uniform Manning coefficient of 0.03 used in the simulation "n003".**

[Figure]

Addition to Figure 7: Averaged monthly streamflow errors and the flood extent discrepancies (i.e., the differences between simulated flood extent and the GIEMS data) in the area upstream of the gauge for 10 gauges during 12 years (1995 – 2006). Streamflow of the Negro subregion (panel (i) ) is approximated by the streamflow difference between the Jatuarana+Careiro gauge and the Manacapuru gauge. The upstream area of each gauge is enclosed by gray lines (or brown dotted lines for the Guajara-mirim gauge) in the basin map.

---

## Author Comment (AC2) · 27 Nov 2016

Response to Referee #2

We appreciate the time and effort of the reviewer and thank the reviewer for the constructive comments. We addressed all the comments, and will include corresponding changes in the revised manuscript.

In the following text, we use blue font for the reviewer comments, black font for our replies, and italics for the revisions in the manuscript.

**Comments**

[C2-1]

The paper presents improvements on the parametrization of the MOSART surface water model. State of art methods are used to update river model and inundation parametrization. The model is evaluated in the Amazon basin and several simulations were performed to evaluate the role of the DEM, river geometry parameters, and backwater effects. The subject addressed by the paper is important. With new data available for regional/global hydrologic simulations, there are several new efforts to improve hydrological models. And the documentation of new improvements/updates of models, as the MOSART, fits the goal of GMD journal. Also, the study of impact of model errors and different parametrizations are important guide future model developments. The paper is generally clear. However, it seems that most of the conclusions from paper analyses were already provided by the past modelling studies in the Amazon (e.g. Paiva et al., 2013, Getirana et al. 2012, Yamazaki et al., 2011, Beighley et al., 2009; Baugh et al., 2013). For example, the past studies already pointed for the importance backwater effects and flooding, performed sensitivity studies on the role of river geometry errors and DEM errors on amazon simulations. So I guess that it would be better to present the paper as a documentation of the improvements of a specific model (MOSART) to move toward state of art methods. And to clarify that the analyses could reproduce similar conclusions from the past studies. So, as the documentation of model parametrizations fits the GMD journal scope, I think that the paper could be published. But it needs to be reviewed clarify the actual contributions, by addressing the comments above and below.

Reply: We thank the reviewer for the positive evaluation and suggestions. We will clarify the contribution mainly as the incorporation of an inundation scheme into the MOSART model. The related revisions will be added in several sections: Abstract, Introduction, Methods and data, Sensitivity study, and Conclusion and discussion.

Our initial manuscript was not clear in the comparisons with previous studies. Following the suggestions of the reviewers, we will also add more discussions on comparisons between our study and former studies. While previous studies provided the foundation for the approach we adopted in our study and

some of our results agree with those of former studies, our study has also provided some new points in terms of methodologies, model results and analyses. These are elaborated below.

1. Methods

1.1 DEM correction

We explicitly considered the spatial variation of the vegetation-caused biases in the DEM data, which alleviated biases for hydrological modeling in the entire Amazon Basin. The DEM correction was based on the spatially varied vegetation-height map and land cover dataset.

In most previous studies of hydrological modeling for the entire Amazon Basin, the DEM was lowered by a uniform height for vegetated area (Coe et al., 2008; Paiva et al., 2013a). Although spatial variation of vegetation-caused biases in DEM was also considered in previous hydrodynamic modeling studies, it was performed only in a comparatively small area of the central Amazon region (Baugh et al., 2013; Wilson et al., 2007). We generalized the approach by using land cover data and vegetation height data that have global coverage so the method can be used in the entire Amazon Basin and other regions.

1.2 Refining channel geometry

We refined the basin-wide empirical formulae for channel cross-sectional dimensions in various subregions to improve the representation of spatial variation in channel geometry (Table 1). In many former studies, the basin-wide formulae were used (Beighley et al., 2009; Coe et al., 2008; Getirana et al., 2012; Yamazaki et al., 2011).

Paiva et al. (2013a) accounted for spatial variation of channel geometry formulae and used various coefficients in their formulae for six zones of the Amazon Basin (Table 1 of Paiva et al. 2013a). But they did not compare the results of diverse subregion formulae with those of the basin-wide formulae.

 2. Sensitivity study

The sensitivity analyses of former studies primarily examined the impacts of various factors (e.g., the inundation scheme, channel geometry, Manning coefficients or backwater effects) on the total flooded area of the central Amazon region (Figs. 9 and 13 of Yamazaki et al. 2011) or the entire Amazon Basin (Fig. 10 of Paiva et al. 2013a), and streamflow and river stages of a few mainstem gauges (Figs. 13, 5a and 5b of Yamazaki et al. 2011; Fig. 10 of Paiva et al. 2013a).

Paiva et al. (2013a) examined the impacts of perturbing precipitation, elevation profiles or maximum soil water storage on modeled surface hydrology (Figs. 10 and 11 of Paiva et al. 2013a). Complementary to their study, we examined the impacts of five other factors (i.e., including the inundation scheme, correcting DEM, refining channel geometry, adjusting Manning coefficients, and considering backwater

effects) on modeled surface hydrology at various locations spread over the Amazon Basin, including inundation of 10 subregions (Fig. 11), streamflow and river stages of more than 10 gauges (at both the mainstem and tributaries) (Figs. 8 and 9), the water-surface profile along the mainstem (Fig. 10).

Our sensitivity study yielded several new points as follows.

2.1 Impacts of including the inundation scheme

This point was not explicitly discussed in the initial manuscript. Following the suggestions of both reviewers, in the revision this point was investigated and discussed.

Our investigation related to river stages was different from the former study. To our knowledge, only Yamazaki et al. (2011) explicitly examined the impacts of the inundation scheme on water depths at the gauge station (Fig. 5b of Yamazaki et al. 2011) and water depths along the mainstem (Fig. 7 of Yamazaki et al. 2011). They conducted three simulations: the diffusion wave simulation with the inundation scheme (FLD+Diff), the kinematic wave simulation with the inundation scheme (FLD+Kine), and the kinematic wave simulation without the inundation scheme (NoFLD). Therefore, while examining the impacts of the inundation scheme on water depths (or river stages), they used the kinematic wave river routing method, but we used the diffusion wave river routing method, which was more advanced (e.g., could represent the backwater effects) (Figs. 9 and 10).

2.2 Impacts of correcting DEM

The vegetation-caused biases in DEM were alleviated with various approaches in a few previous studies in the partial or entire Amazon Basin (Baugh et al., 2013; Coe et al., 2008; Getirana et al., 2012; Paiva et al., 2011, 2013a; Wilson et al., 2007; Yamazaki et al., 2011). To our knowledge, most of these studies did not examine and explicitly report the impacts of DEM correction on the modeled results. Only Baugh et al. (2013) showed the impacts of DEM correction on floodplain water levels and inundation in a comparatively small area in the central Amazon region (Figs. 2 and 5 of Baugh et al. 2013).

Our study examined and explicitly reported the impacts of alleviating vegetation-caused biases in DEM on modeled surface hydrology in the hydrological modeling for the entire Amazon Basin (Figs. 8, 9, 10 and 11). These impacts were not explicitly reported in the past.

2.3 Impacts of refining channel geometry

While examining the impacts of adjusting channel geometry on modeled surface hydrology, we used a method different from those of previous studies, where the channel widths or depths of all the channels were perturbed by a uniform percentage (or a uniform amount) (Fig. 13 of Yamazaki et al. 2011; Fig. 10 of Paiva et al. 2013a).

We refined the basin-wide formulae of channel geometry for various subregions. The channel-geometry changes were caused by the process of refining channel cross-sections and those change ratios were different for various subregions (Table 1). We compared the results of diverse subregion formulae with those of basin-wide formulae to reveal the impacts of adjusting channel geometry on modeled surface hydrology (Figs. 8, 9, 10 and 11). Our method had more physical mechanism than the former method of perturbing channel geometry uniformly in the entire basin.

2.4 Impacts of considering backwater effects

Our model results showed the impacts of backwater effects on flood extent and river stages were more prominent than those of the previous study. To our knowledge, only Yamazaki et al. (2011) explicitly reported the impacts of backwater effects on flood extent (Fig. 9 in Yamazaki et al. 2011) and river stages (Figs. 5b and 7a in Yamazaki et al. 2011). In our study, the impacts of backwater effects on flood extent (Fig. 11) and river stages (Figs. 9 and 10) were more prominent than those of Yamazaki et al. (2011). These differences may be due to the discrepancies in channel geometry or floodplain topography between the two studies.

Our model results showed that backwater effects could advance the flood peak in the Madeira subregion (Fig. 8c). To our knowledge, this phenomenon has not been discussed in the previous modeling studies in the Amazon Basin.

In summary, while our modeling approach and improvements do not differ conceptually from those already explored in previous studies, we attempted to generalize various methods for application over the entire Amazon Basin, which is important as MOSART is used in global Earth System Models. We also provided more comprehensive evaluation of our simulations and analysis of sensitivity of the simulations to various factors, which yielded some findings that have not been discussed in former studies.

[C2-2]

Introduction: I feel that the main goal of this paper should be to document improvements on the MOSART model. So it is important to provide more details in the intro section.

Reply: We agree with the reviewer's assessment. We will revise the Introduction to more explicitly state our objective for implementing and documenting an inundation parameterization in the MOSART model for global application.

[C2-3]

Page 2. Line 25. Which of these challenges were addressed by this paper in a novel way that was not done by the past efforts?

Reply: In the initial manuscript, the comparisons between our study and previous studies were not clear. As discussed in the reply to the first comment [C2-1], on one hand, our study was based on the important foundation of previous studies; on the other hand, our work also yielded some new points in terms of methodologies, model results and sensitivity analyses.

[C2-4]

Page 3. Line 9. Vegetation errors from SRTM DEM were removed globally by F.E. O'Loughlin et al. 2016 RSE. Please review and discuss it in the paper.

Reply: We thank the reviewer for directing us to this study related to our work. O'Loughlin et al. (2016) used both vegetation height data and vegetation density data to estimate vegetation-caused biases embedded in the SRTM DEM data, and for the first time created the 'Bare-Earth' global high resolution DEM from the SRTM data. They also compared their methods with the static correction method used by Baugh et al. (2013). Their study will be discussed in the revised manuscript.

[C2-5]

Page 3. Line 22. See also analyses from Paiva et al., 2013 WRR.

Reply: Paiva et al. (2013a, WRR) analyzed the sensitivities of streamflow, water depths and flooded area to the channel width and depth. The citation of this reference will be supplemented.

[C2-6]

Objectives. What is the new proposed contribution? If the contributions are limited to updating MOSART model with state of art methods, then I think that you should specify it in the objectives and introduce MOSART in the intro section.

Reply: As discussed in our replies to the first and second comments ([C2-1] and [C2-2]), we will make our objectives more clear in the Introduction section. As a reply to a similar comment from the first reviewer, we will add a new section to compare the simulations with and without the inundation parameterization to document its impacts on the overall performance of MOSART. Figures 8, 9, and 10

will be updated to include results for the simulation without inundation for comparison with various simulations that include the inundation parameterizations. The three figures are also attached in the Appendix at the end of this Response.

2.1. How the model defines what is main river network and tributary subnetwork?

Reply: In the MOSART model, each computation unit (subbasin or grid cell) has a major channel (or main channel) and a tributary subnetwork which includes tributaries within the computation unit. Please see the following figure from Li et al. (2013).

[Figure]

FIG. 1. Conceptualization of river network in MOSART. The runoff generated first enters the tributaries (surface runoff via hillslope routing and subsurface runoff without hillslope routing); it is then routed through the tributaries (here conceptualized as a single equivalent channel, as shown by the light blue dashed lines) and is finally discharged into the main channel. The value $V_h$ is the overland flow during hillslope routing, $V_t$ is the channel velocity within the tributaries, and $V_r$ is the channel velocity within the main channel.

The main channels of all the computation units constitute the main-channel network of the entire basin.

The following text will be added in the first paragraph of Section 2.1 :

"*In this model, each computation unit (subbasin or grid cell) has a major channel (or main channel) and a tributary subnetwork which includes tributaries within the computation unit.*"

Eq.1. It seems g can be removed from equation.

Reply: "*g*" will be deleted from this equation.

Continuity equation is not shown. Please show it.

Reply: The continuity equation will be added.

How these equations (kinematic and diffusive) are solved? Please provide details on the numerical methods. finite difference, finite volumes, implicit, explicit? Criteria for time step, spatial discretization? What is done to avoid mass errors.

Reply: The explicit finite difference method is used to solve the equations. The computation units can be grid cells or subbasins. The Courant condition is used for choosing the time step. In this Amazon application, the time step is one minute when the diffusion wave method is used. The cumulative mass error is less than 0.5 percent in these multi-year simulations.

The following text will be added at the end of Section 2.1:

"*In this model, the equations are solved with the explicit finite difference method. Either square grid cells or irregular subbasins can be used as computation units. The time-step size is selected to satisfy the Courant condition to ensure stable computation.*"

2.2. It is not clear how you compute river bed elevation? Is it simply the lowest DEM pixel of the catchment? How the model accounts for the fact that SRTM DEM does not see the river bed? And the fact that the river profile is not flat?

Reply: In Fig. 1b, the elevation profile and the fraction of channel area ($A_c$) can determine the elevation of the channel bank top ($E_t$). The channel bed elevation ($E_b$) is: $E_b = E_t - d$, where $d$ is the channel depth and is estimated in Section 2.5. The channel bed could be lower than the lowest DEM pixel of the catchment because the DEM does not see the channel bed.

In the elevation profile of Fig. 1b, the longitudinal profile of the channel bed is deemed to be flat, which is different from the actual condition in the real world. This assumption may bring about some error when estimating the flooded area.

Fig. 1b will be updated in the manuscript: (1) In the amended elevation profile, the channel bed elevation was lowered; (2) The bank top elevation ($E_t$) and the channel bed elevation ($E_b$) were indicated. Please find the revised Fig. 1 in the Appendix of this response.

The following text will be added in the second paragraph of Section 2.2:

"*The channel bed elevation equals the difference of the bank top elevation and the channel depth which is estimated in Sect. 2.5. The channel bed could be lower than the lowest DEM pixel of the computation unit because the DEM do not reflect the channel bed elevation.*"

[C2-12]

2.3. How the basins are defined? What is the input data? Hydrosheds? Please make it clear.

Reply: Yes, the subbasins were extracted from the HydroSHEDS DEM. The following text will be revised and moved from Section 2.4 to Section 2.3.

"*The 3-second resolution HydroSHEDS DEM data developed by United States Geological Survey (USGS) (http://hydrosheds.cr.usgs.gov/) was used in this study. The hydrologically conditioned HydroSHEDS DEM was used to generate the digital river network and subbasins.*"

[C2-13]

Pag. 6 Line 30. What is the criteria to define river length? How time step is defined? How these choices affect model errors (model numerical stability, mass errors, numerical dispersion, … ) ? Please clarify and discuss it.

Reply: We used comparatively coarse subbasins (the average area is 1091.7 $km^2$) due to computational costs. Each subbasin has a main channel. The main channel length varies with the subbasin size.

The time step was determined based on the Courant condition and some test simulations. The time step of one minute was used so that the simulations were stable. The cumulative mass error for the entire Amazon Basin was less than 0.5 percent in the simulations.

The following text will be added in the first paragraph of Section 2.3 :

"*Comparatively coarse subbasins were adopted due to the consideration of computational costs. ... ... To ensure stable computation, the time-step size was determined based on the Courant condition and some experimental simulations. The time step of one minute was used for all the simulations.*"

[C2-14]

2.4. Vegetation Errors. Was the corrected DEM validated ? Please justify and compare these methods to the global SRTM DEM product free of veg errors recently developed by F.E. O'Loughlin et al. 2016 RSE.

Reply: Our method was based on that of Baugh et al. (2013). Moreover, we also used a high resolution (3 arc-second) land cover dataset when estimating the vegetation-caused biases in the SRTM DEM data.

O'Loughlin et al. (2016) used both vegetation height data and vegetation density data to estimate vegetation-caused biases in the SRTM DEM data. They also compared their methods to that of Baugh et al. (2013), which was named the static correction method. They stated that the static correction method was effective, but moderately worse than their methods. Their study will be discussed in the revised manuscript.

[C2-15]

2.6. Line 16. What literature was used to define Manning at 0.03 and 0.05? I feel that the parametrization of Manning needs more justification (past studies or calibration). How these choices will impact model results?

Reply: Previous studies were cited to justify our choice of the Manning coefficients. The sensitivity study part (Section 4.3) discusses the effects of the Manning coefficients on model results. Refining the Manning coefficients could improve streamflow hydrographs. The increase of the Manning coefficient could affect flood extent, streamflow and river stages in local, upstream or downstream subbasins.

The text will be revised as follows:

"*Following Getirana et al. (2012), $n_{max}$ and $n_{min}$ were set as 0.05 and 0.03, respectively. In addition, a few other studies of the Amazon Basin adopted a similar range of values between 0.03 and 0.05 for the Amazon Basin (e.g., Beighley et al., 2009; Paiva et al., 2013a; Yamazaki et al., 2011).*"

[C2-16]

2.7. Line 6. Why average manning of 0.03 ? You should use the average Manning from the reference simulation or use other approach to isolate the effect of variable vs constant manning.

Reply: We conducted a new simulation using the Manning coefficient of 0.04 (the average of 0.03 and 0.05) for all the channels (abbreviated as 'n004'). We compared the Nash–Sutcliffe efficiency coefficients (NSEs) of streamflow between the simulation 'n004' and the control simulation (which used varied Manning coefficients) (revised Table 4). The control simulation performed better than the simulation 'n004' at a majority of the 13 stream gauges.

Section 2.7 and Table 2 will be revised to add the description of the simulation 'n004'.

[C2-17]

2.7. What is optimal combination? Was any calibration performed?

Reply: The original manuscript was not clear. It meant that in the control simulation, the preferred methodologies were used at each aspect. We did not try to calibrate parameters to improve the modeled results.

The text will be expanded to be more specific as follows:

"*In the control simulation (abbreviated as "CTL"), the preferred methodologies for five aspects were used: (1) the inundation scheme was turned on; (2) vegetation-caused biases in the DEM data were alleviated; (3) the basin-wide channel geometry formulae were refined for different subregions; (4) the Manning roughness coefficient varied with the channel size; (5) the diffusion wave method was used to represent river flow in channels. The results of the control simulation were used in the model evaluation (Sect. 3).*"

[C2-18]

3.1. How the model performance compare to past modelling studies in the Amazon? Please discuss it in the manuscript.

Reply: The modeled streamflow results were compared with a few previous studies. The following text will be added to the second paragraph of Section 3.1:

"*In general, the simulated streamflow results are comparable to those of a few previous studies (e.g., Getirana et al., 2012; Yamazaki et al., 2011) and slightly worse than those of Paiva et al. (2013a).*"

3.2. Equation 7. Do you use this equation to estimate a parameter for simulation? If yes, this explanation should appear o section 2. Why this approach were selected? How it compares to previous studies? How this choice impact model results? See Paiva et al., 2013 for an analyses of impact of bed elevation errors on simulations.

Reply: The relative riverbed elevations from this equation can be deemed as parameters for channel routing computation. Actually, riverbed slopes ($S_0$ in Equation (2)) are directly used in the channel routing computation in this study. This approach is the same as those of previous studies (Beighley et al., 2009; Getirana et al., 2012; Paiva et al., 2011, 2013a; Yamazaki et al., 2011).

The Equation (7) and the related descriptions will be moved to the methodology part (Section 2.5).

In this study, the method for estimating the riverbed slope may be different from those of previous studies. The riverbed slopes of this study were directly derived from the DEM. Some previous studies first alleviated errors caused by water depths or vegetation heights, then used the corrected DEM to derive riverbed slopes (e.g., Paiva et al., 2011). So the method of our study has less physical mechanism and may have more uncertainties than those of some previous studies.

Paiva et al. (2013a) studied the sensitivities of streamflow, water depths and flooded area to riverbed elevations. In scenario simulations, riverbed elevations were perturbed by 3m, 1m, -1m, or -3m. In our understanding, the riverbed elevations of the entire basin were raised or lowered by a uniform value in any single simulation. So this treatment did not affect the riverbed slopes used in channel routing computation. Actually this treatment reduced or increased the channel depths, which decreased or enlarged the channel conveyance capacities. In our study, the impacts of channel cross-sectional geometry on surface hydrology were studied in a different way (Section 2.5 and 4.3).

3.2. How model performance for river elevation compares to previous modelling studies in the amazon? Please discuss it in the manuscript.

Reply: The simulated river-stage results were compared to some previous studies. The following text will be added to the third paragraph of Section 3.2:

" *Overall, in terms of the timing and magnitude of fluctuations, the modeled river stages of this study are comparable with some previous investigations* (Coe et al., 2008; Getirana et al., 2012; Paiva et al., 2013a). "

3.3. How model performance for flood extent compares to previous modelling studies in the amazon? Please discuss it in the manuscript.

Reply: The modeled inundation results were compared to a few previous studies. The following text will be added at the end of Section 3.3:

" *The flood extent results were compared with those of a few previous studies which also used the GIEMS data* (Getirana et al., 2012; Paiva et al., 2013a; Yamazaki et al., 2011). *As mentioned above, the GIEMS data had non-negligible uncertainties. So the comparison to this data should not be deemed as a criterion for judging different modeling studies. The spatial inundation patterns of this study were slightly better than those of Getirana et al. (2012), and comparable to those of Yamazaki et al. (2011) and Paiva et al. (2013a). In terms of monthly total flooded areas, this study, Getirana et al. (2012) and Paiva et al. (2013a) were comparable at the whole-basin scale; this study and Getirana et al. (2012) were closer to the GIEMS data than Paiva et al. (2013a) at the subregion scale.* "

4.1. How these analyses compare to previous analyses of impact of DEM and floodplains on Amazon simulations from previous modelling studies?

Reply: The vegetation-caused biases in DEM were alleviated with various approaches in a few previous studies. To our understanding, most of those studies did not explicitly report the effects of the DEM correction on the modeled results except Baugh et al. (2013). The following text will be added to Section 4.2 Correction of DEM (previous Section 4.1):

"*The vegetation-caused biases in DEM data were alleviated with various approaches in a few previous studies of modeling the surface water dynamics in the Amazon Basin* (Baugh et al., 2013; Coe et al., 2008; Getirana et al., 2012; Paiva et al., 2011, 2013a; Wilson et al., 2007; Yamazaki et al., 2011). *Most of these studies did not examine and explicitly report the effects of the DEM correction on the modeled results. Baugh et al. (2013) demonstrated that alleviating vegetation-caused biases in DEM could improve the modeled water levels and inundation over floodplains adjacent to a 280-km reach of the central Amazon.*"

The impacts of using the inundation scheme on modeled surface hydrology were examined and reported in a few studies: (1) Yamazaki et al. (2011) showed the effects on streamflow, water depths, and flow velocities at the Obidos gauge; and the effects on the mainstem water-surface profile; (2) Getirana et al.

(2012) demonstrated the effects on streamflow of a few mainstem gauges; (3) Paiva et al. (2013a) reported the effects on streamflow at the Obidos and Manacapuru gauges.

In the revised manuscript, we will add a new subsection (Section 4.1 Inundation representation) to report the impacts of using the inundation scheme on modeled surface water dynamics including: (1) Streamflow at 13 mainstem and tributary gauges; (2) River stages at the 13 gauges; (3) the mainstem water-surface profile.

[C2-23]

 4.2. It is not change in channel storage capacity that changes simulation. It is changes in channel conductance capacity.

Reply: "*channel storage capacity*" will be revised to be "*channel conveyance capacity*" throughout the manuscript.

[C2-24]

 4.2. How these analyses compare to previous analyses of channel geometry from previous modelling studies?

Reply: Some previous studies (e.g., Paiva et al., 2013a; Yamazaki et al., 2011) also investigated the sensitivities of modeled surface hydrology to channel geometry. They pointed out the importance of channel geometry that motivated similar analysis in our study. At the same time, the methods and results of our study had some new points: (1) channel-geometry changes were caused by the process of refining channel cross-sections and those changes were spatially varied (Table 1); (2) we examined the effects of channel-geometry changes on modeled surface hydrology at spatially distributed locations (i.e., different subregions, tributary and mainstem gauges, and the mainstem); (3) some of our result-analyzing approaches were different from those of former studies.

The following text will be added to the end of "Section 4.3 Adjustment of channel geometry" (previous Section 4.2):

   " *The sensitivities of modeled surface hydrology to channel geometry were also investigated by some former studies (e.g., Paiva et al., 2013a; Yamazaki et al., 2011). Yamazaki et al. (2011) perturbed the channel width or depth by a uniform percentage for all the channels and examined the effects of these channel-geometry changes on streamflow of the Obidos gauge and the flooded area over the central Amazon region. Paiva et al. (2013a) perturbed the channel width by a uniform percentage or perturbed the channel-bottom level by a uniform height, which was equivalent to perturbing the channel depth by a*

*uniform value, and investigated the effects of these channel-geometry changes on streamflow of the Obidos gauge, channel water depths of the Manacapuru gauge, and the total flooded area of the entire Amazon Basin. These two studies showed the sensitivities of modeled surface hydrology to channel geometry, and the interactions between streamflow, water depths and inundation. These previous studies pointed out the importance of channel geometry and provided motivation for similar analysis. In this study, channel-geometry changes were caused by the process of refining channel cross-sections and those changes were spatially varied (Table 1). We examined the effects of channel-geometry changes on inundation of 10 subregions, streamflow of 13 gauges, river stages near 11 gauges, as well as the mainstem water-surface profile. In addition, the effects of channel-geometry changes on modeled surface water dynamics were analyzed with approaches of which some were different from those of the former studies. "*

[C2-25]

 4.3. I'm not sure if this analysis is conclusive. It is not possible to be sure that the differences in results are related to variable Manning or if it is because a specific value of 0.03 was chosen. This value may be different from the average value of the control simulation. I suggest the computation of the average Manning from control simulation and using this value for the new simulation.

Reply: We conducted a new simulation "n004" which used a constant Manning roughness coefficient of 0.04 (i.e., the average of 0.03 and 0.05) for all the channels. We compared the streamflow Nash–Sutcliffe efficiency coefficients (NSEs) of three simulations (CTL, n004 and n003). The beginning part of "Section 4.4 Varying the Manning coefficients" (previous Section 4.3) will be revised as follows:

 " *The streamflow Nash–Sutcliffe efficiency coefficients (NSEs) of the simulation "CTL" were compared with those of the simulations "n003" and "n004" (Table 4). The NSEs of the simulation "CTL" are higher than those of the simulation "n004" at 10 of the 13 gauges (except Fazenda vista alegre, Itapeua and Manacapuru), and higher than those of the simulation "n003" at 12 of the 13 gauges (except Obidos). These results suggest that the spatially varied Manning coefficients are more appropriate than the uniform Manning coefficient of 0.03 or 0.04 for the simulations of this study.*

 *The spatially varied Manning coefficients range from 0.03 to 0.05 and are equal to or larger than the Manning coefficient of 0.03. The results of the simulation "CTL" are compared to those of the simulation "n003" to reveal the effects of Manning coefficient increases on modeled surface water dynamics. "*

4.3. How these analyses compare to previous analyses of Manning role from previous modelling studies?

Reply: A few former studies (e.g., Paiva et al., 2013a; Yamazaki et al., 2011) also investigated the sensitivities of simulated surface hydrology to the Manning roughness coefficient. They revealed the importance of the Manning coefficient and motivated similar analysis in our study. At the same time, the approaches and analyses of this study had some new points: (1) the Manning coefficient increase depended on the channel depth; (2) we examined the effects of Manning coefficient changes on modeled surface hydrology at spatially diverse locations (i.e., different subregions, tributary and mainstem gauges, and the mainstem).

The following text will be added to the end of "Section 4.4 Varying the Manning coefficients" (previous Section 4.3):

  " *A few previous studies for the Amazon Basin (e.g., Paiva et al., 2013a; Yamazaki et al., 2011) conducted numerical experiments to reveal the sensitivities of modeled surface hydrology to the Manning coefficient. Yamazaki et al. (2011) perturbed the Manning coefficient by a uniform percentage for all the channels and examined the effects of the Manning coefficient change on streamflow of the Obidos gauge and the flooded area over the central Amazon region. Using a similar approach, Paiva et al. (2013a) investigated the effects of the Manning coefficient change on streamflow of the Obidos gauge, channel water depths of the Manacapuru gauge, and the total flooded area of the entire Amazon Basin. These studies revealed that increasing the Manning coefficient could raise the river stage, enlarge the flooded area, and reduce and delay the flood peak. In this study, instead of being perturbed uniformly, the Manning coefficient varied with the channel depth. We examined the effects of Manning coefficient changes on flood extent of 10 subregions, streamflow of 13 gauges, river stages near 11 gauges, and the mainstem water-surface profile.* "

Figure 10. This figure is confusing. It's hard to understand the break in the profile. Please review it.

Reply: Figure 10 has been replotted to avoid the break and to use the same y-axis for the entire river length. This figure will be updated in the revised manuscript. Please find the revised Fig. 10 in the Appendix of this response.

4.4. Line 20. See also analyses on the importance of backwater effects for amazon simulations from Paiva et al., 2013 WRR and Paiva et al., 2013 Hyd.Process. Please compare and discuss in the manuscript.

Reply: Paiva et al. (2013b, HP) demonstrated the important impacts of backwater effects on streamflow of the mainstem and tributaries, and discussed the important role of backwater effects in the inundation dynamics and river stages of the Amazon Basin. Paiva et al. (2013a, WRR) showed the important impacts of backwater effects on streamflow of two mainstem gauges.

We examined the impacts of backwater effects on flood extent in 10 subregions, streamflow of 13 gauges, river stages near 11 gauges, and the mainstem water-surface profile.

The following text will be added or revised:

"*Paiva et al. (2013b) used the dynamic wave method to represent river flow in the Solimoes River basin, which is the western upstream portion of the Amazon Basin. They discussed the important role of backwater effects in the inundation dynamics of the Amazon. In this study, we examined the impacts of backwater effects on flood extent in the 10 subregions constituting the Amazon Basin (Fig. 11), and demonstrated the spatial pattern of flood extent changes caused by backwater effects (Figs. 12j and 12k).*"

"*These backwater effects on hydrographs agree with the results of Paiva et al. (2013)*" will be replaced with "*These results agree with Paiva et al. (2013a, 2013b), which demonstrate the important role of the backwater effects on streamflow of the mainstem and tributaries of the Amazon Basin.*"

"*In addition, to our knowledge, this phenomenon of backwater effects on the streamflow timing has not been discussed in the previous modeling studies in the Amazon Basin.*"

"*The result of this study also agrees with Paiva et al. (2013b) which discussed the backwater effects on river stages in the Solimoes River basin.*"

Conclusions. Line 20. Review Yamazaki et al., 2013. WRR for discussion in Catchment vs grid based simulations.

Reply: Yamazaki et al. (2013, WRR) used a special computation unit, which had characteristics of both catchment unit and grid unit. Using their computation unit could preserve the river flow pathway better

than using the grid unit. Their computation units were more even than catchment units in terms of area. Their method will be discussed in the revised manuscript.

[C2-30]

Conclusions: I'm not sure if there are new conclusions /findings that were not addressed by the past modelling studies in the Amazon (e.g. Paiva et al., 2013, Getirana et al. 2012, Yamazaki et al., 2011, Beighley et al., 2009; Baugh et al., 2013). The past studies already pointed for the importance backwater effects and flooding, performed Sensitivity studies on the role of river geometry errors and DEM errors on amazon simulations. It is important to recognize that the analyses from this paper only reproduced similar conclusions from the past studies. And also clarify that the new contribution from this paper is mostly on updating/improving the parametrization of an specific model, i.e. MOSART model by including improvements tested or suggested by the previous studies.

Reply: We thank the reviewer for the constructive suggestions.

In the initial manuscript, the comparisons between our work and previous studies were not clear. The manuscript will be improved at this aspect. As discussed in our reply to the first comment [C2-1], on one hand, our work was based on the important foundation of previous studies; at the same time, our investigation also had a few new points in terms of methodologies, simulation results and sensitivity analyses.

Following the suggestion of both reviewers, the contribution of incorporating the inundation scheme into the MOSART model will be described more clearly than before in the revised manuscript.

**References:**

[revised manuscript text omitted]

---

## Author Comment (AC3) · 27 Nov 2016

Dear Dr. Neal,

We have submitted point-by-point response to the two reviewers' comments for our manuscript.

Both referees provided very constructive comments. We have addressed all the comments and planned to include the corresponding changes in the revised manuscript. The major modifications are summarized as follows:

[Figure]

1. As suggested by the referees, we will clarify the main contribution of our study as incorporating an inundation scheme in the MOSART model, which is used in Earth System Models. To document our effort, we conducted a new simulation called "NoInund" (with the inundation scheme turned off) and compared its results with those of the control simulation "CTL" (where the inundation scheme was turned on). This comparison revealed the effects of the inundation scheme on modeled surface hydrology, and will be presented in a new subsection (Section 4.1 Inundation representation).

2. We included more discussions to compare our study with previous studies, which provided the foundation for the approach we have taken. Although some of our results agree with those of former studies, we have also provided some new insights in terms of methodologies, model results and analyses. The manuscript will be revised to be more clear on this point.

3. We also addressed the other comments of the referees to make the manuscript clearer, more precise, or more complete than before.

4. As a result of the above revisions, five of the 13 figures and two of the four tables will be updated in the revised manuscript.

We appreciate your time and attention for this manuscript.

Sincerely,

L. Ruby Leung, Ph.D. Laboratory Fellow Atmospheric Sciences and Global Change Division Pacific Northwest National Laboratory Richland, Washington State, USA

---

## Author Response (AR1)

Final response for the following manuscript:

Luo, X., Li, H.-Y., Leung, L. R., Tesfa, T. K., Getirana, A., Papa, F., and Hess, L. L.: Modeling surface water dynamics in the Amazon Basin using MOSART-Inundation-v1.0: Impacts of geomorphological parameters and river flow representation, Geosci. Model Dev. Discuss., doi:10.5194/gmd-2016-210, in review, 2016.

December 21, 2016

Dr. Jeffrey Neal

Topical Editor of Geoscientific Model Development
School of Geographical Sciences
University of Bristol

Dear Dr. Neal,

We would like to submit the final response and the revised manuscript.

Both referees provided very constructive comments. We have addressed all the comments and made the corresponding changes in the revised manuscript. The major modifications are summarized as follows:

1. As suggested by the referees, we clarified the main contribution of our study as incorporating an inundation scheme in the MOSART model, which is used in Earth System Models. To document our effort, we conducted a new simulation called "NoInund" (with the inundation scheme turned off) and compared its results with those of the control simulation "CTL" (where the inundation scheme was turned on). This comparison revealed the effects of the inundation scheme on modeled surface hydrology, and was presented in a new subsection (Section 4.1 Representing floodplain inundation).

2. We included more discussions to compare our study with previous studies, which provided the foundation for the approaches we have taken. Although some of our results agree with those of former studies, we have also provided some new insights in terms of methodologies, model results and analyses. The manuscript was revised to be clearer on this point.

3. We also addressed the other comments of the referees to make the manuscript clearer, more precise, or more complete than before.

4. As a result of the above revisions, five of the 13 figures and two of the four tables were updated in the revised manuscript.

We appreciate your time and effort for this manuscript.

Sincerely,

L. Ruby Leung   Ph.D.
Laboratory Fellow
Atmospheric Sciences and Global Change Division
Pacific Northwest National Laboratory
Richland, Washington State, USA

**Response to Referee #1**

We appreciate the time and effort of the reviewer and thank the reviewer for the constructive comments. We provided replies to all the comments, and made the corresponding changes in the revised manuscript.

In the following text, we use blue color for the reviewer comments, black color for the replies, and italics for the revisions in the manuscript.

**Comments**

[C1-1]

This manuscript describes the development of the MOSART river transport model to include a flood inundation scheme which was then tested across the Amazon basin. Excellent detail is given as to the setup of the model including the processing of the DEM and channel geometry parameters. The model is run for a time period longer than 20 years and evaluated against in-situ streamflow observations and remotely sensed satellite data of river stage and flood extent. Results from the evaluation showed good agreement in each of these aspects. A sensitivity analysis was then conducted to assess the impact of the DEM and channel geometry corrections, setting a uniform Manning's n and using a kinematic channel flow equation. Sensitivities were found in each variable due to the influence they have upon the floodplain elevation, channel capacity and flow velocity.

The manuscript's contribution to model development is the inclusion of an inundation scheme to the MOSART model, however this is not explicitly stated until page 6 therefore leaving the reader unclear about the paper's contribution for most of the introductory sections. The authors should revise the abstract to state much more clearly that this is one major contribution of the manuscript.

Reply: We thank the reviewer for the positive evaluation and suggestion. In the Abstract and Introduction of the revised manuscript, we more clearly stated the goal of our study as mainly to incorporate and document an inundation scheme in the MOSART model, which is used in Earth System Models.

[C1-2]

One aim of the manuscript is to investigate the importance of geomorphic parameters and river flow representation when modelling the Amazon basin. This is done through the results of the sensitivity analysis, however these mostly back up results from previous papers which also describe the parameterisation of large scale river models in the Amazon basin. Therefore the novel contribution from this aspect is minimised, the greatest contribution from this paper is in describing the model development of the MOSART model.

Reply: In the initial manuscript, the comparisons between our study and previous studies were not clear. We added more discussions to compare the results of our study with those of former studies. On one hand, our work was based on the important foundation of previous studies; on the other hand, our study had some new points in terms of methodologies, model results and sensitivity analyses.

While our modeling approach and improvements do not differ conceptually from those already explored in previous studies, we attempted to generalize various methods for application over the entire Amazon Basin, which is important as MOSART is used in global Earth System Models. Our study also provided more comprehensive examination of our simulations and analysis of sensitivity of the simulations to various factors, which yielded some findings that have not been discussed in former studies, or are different from those of former studies.

[C1-3]

There is no comparison between the results from the model developed in this manuscript with results from the previous version of the model without the inundation scheme. Clearly it is not possible to compare the results of inundation extent but for a model development paper there needs to be some direct comparison between the results of the developed model and those of its predecessor. In this case it should be possible to compare the results of streamflow and river stage. I believe that the model development in this manuscript is significant and merits eventual publication, however I would suggest that it is reconsidered after major revisions so that the authors can include results from a direct comparison between the two model versions.

Reply: We thank the reviewer for the suggestion and positive evaluation.

We have conducted a new simulation "NoInund" (with the inundation scheme turned off) and compared its results with those of the control simulation "CTL" (where the inundation scheme was turned on). This comparison revealed the impacts of the inundation scheme on modeled streamflow and river stages. The comparison was presented in a new subsection (Section 4.1 Representing floodplain inundation). In brief, we updated Figures 8, 9, and 10 to include the results of "NoInund" for comparison with

simulations that include the inundation scheme in different sensitivity experiments. In brief, including inundation generally improves the simulation of streamflow and river stages compared to the simulation without the inundation parameterization.

**Comments**

Additionally please find below the following minor corrections:

[C1-4]

Equation 2 define v

Reply: The definition of $v$ was added as follows (below the newly supplemented Equation (1)) :

*"… where $v$ is the flow velocity [unit: m s$^{-1}$];… "*

[C1-5]

Page 7 line 3 how was it decided to combine the neighbouring catchments?

Reply: The number of catchments is comparatively large, so to show the inundation results more concisely, the catchments were combined to a few subregions. More explanation of the combining procedure was added in this paragraph. The revised text read:

*" Twenty eight large tributary catchments were first delineated and then aggregated to nine tributary subregions. Initially, seven major catchments (i.e., Xingu, Tapajos, Madeira, Purus, Jurua, Japura and Negro) were selected as subregions or the major part of a subregion; Then the Upper-Solimoes catchments were combined as one subregion, the northeast catchments were combined as another subregion, and the remaining five large catchments were incorporated into their adjacent subregions. This way, nine tributary subregions were delineated. Lastly, all the small tributary catchments and the area draining directly to the mainstem were aggregated to be the tenth subregion (i.e., the mainstem subregion). "*

[C1-6]

Page 8 line 9 should read 'lowered to 2.5m'

Reply: The original manuscript is not clear. There should be "an amount of" added before "2.5 m". However, this sentence was removed in the revision (please see the reply to the comment below).

[C1-7]

Page 8 line 9 why was a distinction made between shrubs which were over 5m and those which were lower - why the different treatment when correcting the SRTM?

Reply: The original manuscript is not clear. Some sentences were revised.

The resolution of the vegetation height data is coarser than the resolution of the land cover data. Hence within one pixel of the vegetation height data, there may be more than one land cover class, which should not be assigned the same vegetation height. For shrubs, any vegetation height larger than 5 m should be an overestimation (according to Junk et al. 2011), so an upper limit of 5 m is imposed. After this correction, 50% of the vegetation height was deducted from the DEM pixel covered by shrubs.

The text was revised as: *"In the high resolution land cover dataset, shrubs were defined to be less than 5 m tall (Junk et al., 2011). So for DEM pixels with shrubs, the vegetation height was determined by the vegetation height data, but with an upper limit of 5 m. After this correction, the elevations were lowered by 50% of the vegetation heights for shrub DEM pixels."*

[C1-8]

Page 8 line 13 what was the uniform value that was subtracted from areas located outside the floodplain?

Reply: The original manuscript is not clear. For the fine DEM pixels within one coarse vegetation height pixel, a unique vegetation height is used, but for different vegetation height pixels, the vegetation height can be different even for the same vegetation class.

The text was revised as: *"... , a uniform vegetation height was applied for all the DEM pixels within each vegetation height pixel"*.

[C1-9]

Page 7 line 15 were the elevation profiles not defined from the vegetation corrected DEM?

Reply: Yes, the elevation profiles were generated from the vegetation corrected DEM.

In the revised manuscript, this point was clarified by the following sentence near the beginning of the second paragraph in Section 2.4 "Vegetation-caused biases in DEM" :

*" Before being used for producing elevation profiles, the void-filled HydroSHEDS DEM was processed to alleviate the biases caused by vegetation. "*

[C1-10]

Page 9 line 13 how were the gauges distributed amongst the 10 regions? Some regions might have only had a few gauges hence the significance of the RMSE value might be low, plus this might override the significance of geomorphological factors in applying this correction

Reply: The coefficients of the basin-wide channel geometry formulae were adjusted for seven of the 10 subregions (except "Xingu", "Upper-Solimoes tributaries" and "Mainstem"; shown in Table 1). Each of the seven subregions used 3 – 13 gauges.

The channel geometry is important for inundation modeling of the "Madeira" and "Negro" subregions which have evident inundation and large area. The "Madeira" and "Negro" subregions used 12 and 13 gauges, respectively.

[C1-11]

Page 10 line 16 give an example of the literature - a reference to a textbook for example

Reply: The text was revised as follows:

"*Following Getirana et al. (2012), $n_{\max}$ and $n_{\min}$ were set as 0.05 and 0.03, respectively. In addition, a few other studies of the Amazon Basin adopted similar values around the range of 0.03 – 0.05 for the Manning coefficient (Beighley et al., 2009; Paiva et al., 2013a; Yamazaki et al., 2011).*"

[C1-12]

Page 11 line 15 - can river flow in the upper tributaries really be evaluated using the gauge at Santo Antonio do Ica which is located much further downstream? The steeper gradients of the tributaries are likely to have different flow hydraulics to that in the mainstem, can the authors comment on this and provide further justification for using this gauge to make the evaluation?

Reply: We agree that the river flow in the tributaries could be quite different from that of the mainstem so the river flow in the tributaries cannot be represented by using results at this gauge. Our description in the original manuscript is not accurate so the sentence was revised as follows:

*"Most of this subregion is controlled by the Santo antonio do ica gauge at the upper mainstem."*

[C1-13]

Page 11 line 22 there is a positive runoff bias in the Japura basin which goes against the overall trend of negative biases in the western portion of the basin, could the authors explain what may be causing this?

Reply: There is a negative runoff bias in the subregion "Upper-Solimoes tributaries" which is on the west side of the Japura basin (Fig. 3i). On the other hand, there is a positive runoff bias in the western part of the subregion "Negro", which is on the east side of the Japura basin (Fig. 3g). The western Negro and the Japura basin are adjacent, and both have positive runoff biases.

The runoff biases could be due to errors in precipitation inputs or errors in the land surface water fluxes calculated by the land surface model (e.g., canopy evaporation, plant transpiration, and soil evaporation).

The following sentence was added:

*"The runoff biases could be caused by errors in the precipitation forcing dataset or errors in the land surface water fluxes calculated by the land surface model (e.g., canopy evaporation, plant transpiration, and soil evaporation)."*

[C1-14]

Eq 7 This describes how the simulated river stages are converted into elevations, should this not therefore be included in section 2.5 which describes how the river channel geometry in the model was established?

Reply: Following the suggestion by the reviewer, the method for estimating the riverbed elevation was moved to Section 2.5.

[C1-15]

Page 12 line 12 how were the simulated river stages shifted to coincide with the observations?

Reply: All the simulated river stages of the same subbasin were raised or lowered by a uniform height, to facilitate comparison of the timing and magnitude between the simulated river stages and the observations. A similar method was also used in Figure 7 of Coe et al. (2002) .

The text was supplemented by *"of the same subbasin"* and *"by a uniform height"*, and now read: *"For better visual comparison, the simulated river stages of the same subbasin were shifted by a uniform height to coincide with the observations."*

[C1-16]

Page 12 line 15 should another metric be calculated alongside the correlation coefficient? In the Negro and Japura basins for example Fig 4 shows there is a very high correlation but the differences between the simulations and observations are very large. Perhaps calculating another metric might capture this?

Reply: The original manuscript was not clear. The standard deviations were calculated and used to indicate river stage fluctuations. It was discussed that the river stage fluctuations were overestimated for the subbasins of 4 gauges (i.e., Canutama[Purus], Acanaui[Japura], Serrinha[Negro] and Santo antonio do ica[Mainstem] ).

To make the text more clear, the phrase "*as well as standard deviation for simulated and observed river stages*" was replaced with "*Moreover, the standard deviations for the simulated and observed river stages were also calculated.*"

[C1-17]

Page 13 line 2 should read 'lake areas'

Reply: This was corrected as suggested.

[C1-18]

Figure 6 the four plots should be replaced with two difference plots, one showing the difference between the simulated and observed during high water and the other during low water. This would better visualise the difference between the two simulations.

Reply: Fig. 6 was supplemented by two panels showing the differences during high water season (Fig. 6e) and low water season (Fig. 6f). The original four panels were kept in order to show the spatial patterns of inundation.

Page 13 line 13 the statement that the GIEMS data and simulation agree reasonably well is very vague. Figure 6 appears to show that the simulation overestimates the extent in the lowland portion of the basin, especially at low water. This sentence should be expanded to include more details about where the differences occur.

Reply: We added some discussions of the similarities and differences between model results and GIEMS data in the revised manuscript (as follows).

*"Both the observations and the simulated results show evident inundation in the regions near the middle and lower mainstem. The observed inundation in the upper Madeira subregion and middle Negro subregion is partially captured by the model. The comparison also shows spatially varying differences between the modeled and observed flood extent (Figs. 6e and 6f). The modeled flood extent exceeds the observations in the lower Madeira subregion near the mainstem and around the major reaches in the middle Negro subregion. At the same time, the modeled flood extent is lower than the observations for some subbasins in the mainstem, upper Madeira, Upper-Solimoes and middle Negro subregions."*

[C1-20]

Figure 7 it could be useful to plot the data by seasons e.g AMJ, JAS, OND, JFM as this might show if the errors are concentrated in a particular season e.g. low water.

Reply: Thank you for the suggestion. Similar to Figure 7 that compares the annual averaged biases in flood extent and streamflow, we plotted the averaged monthly streamflow errors and the flood extent discrepancies (i.e., the differences between simulated flood extent and the GIEMS data) during 12 years (1995 – 2006). Please find the figure in the Appendix. This figure shows that the seasonal distribution of streamflow errors varies for different gauges. For example, for "(a) Altamira" and "(b) Itaituba", evident positive biases occur from January to April; for "(c) Fazenda vista alegre" and "(d) Guajara-mirim", positive biases are more evident from about May to October; for "(h) Acanaui", positive biases are more evident from about March to July. Except for three subregions (Negro, Cach de porteira-con, and Tabatinga), the seasonality of flood extent discrepancies follows the seasonality of streamflow errors very closely, indicating the important contribution of streamflow errors to flood extent biases on seasonal time scale.

Figure 8, why does this figure refer to the average seasonal cycle from 1995-2006 whilst figures 9 and 10 refer to 2007 only? Does this explain why the results for the kinematic simulation are so different between figures 8 and 9 & 10? I would expect the kinematic simulation to be very different to the control simulation (as it appears in Figs 9 and 10), yet does this not appear to be the case for streamflow - could the authors explain why streamflow is not sensitive to the kinematic solution or replot Fig 8 for 2007 only so that it is directly comparable to Figs 9 and 10?

Reply: Figs. 8, 9 and 10 have been replotted to show the results of the same year. We did not have observed streamflow data for year 2007 so we plotted the results of another year (2005).

The three figures show that the differences in streamflow between the simulation KW (kinematic wave method) and the simulation CTL (diffusion wave method) are not as evident as those in river stages. Previous studies have yielded similar results (e.g., Fig. 5 of Yamazaki et al., 2011). The reason could be that the flow velocities in the simulation KW (which are based on riverbed slopes) are also quite different from those in the simulation CTL (which are based on friction slopes).

[C1-22]

Figure 10 is confusing with the y-axis reset for 0-1500 km for the simulations but not for the riverbed profile. These graphs should use the same y-axis for the entire river length in order to remove the confusing jump that happens at 1500 km.

Reply: Figure 10 was replotted to use the same y-axis for the entire river length. We thank the reviewer for the suggestion.

[C1-23]

Section 4.2, the greatest effect is shown in the Madeira basin, is this most likely because the multiplicative factor (0.36) has the greatest effect on changing the channel geometry relative to the other basins? This should be stated more explicitly in the second paragraph.

Reply: Yes, the reason is that the channel geometry changes in the Madeira subregion are larger than those of the other subregions.

To make the discussion more explicit, the following revisions were made in the second paragraph:

(1) "*channel geometry changes*" was replaced with "*the channel cross-sectional area is multiplied by a factor of 0.36 (Table 1)*" ;

(2) Added one sentence: '*Similar phenomenon is observed at the gauge "Cach da porteira-con" in the Northeast subregion (Fig. 8h), where the channel cross-sectional area is multiplied by a factor of 0.48.*'

(3) Added "*caused by refining channel geometry*" after "*Inundation changes*".

[C1-24]

Figure 13 needs to be redone as it is very difficult to follow the decision chain that the authors are trying to imply. For example at the second box there are four options but how is a reader meant to decide between these?

Reply: The original manuscript was not clear. In general, the phenomena before and after an arrow have the cause – effect relationship.

The figure caption was revised: "*An example of the effects of channel cross-sectional geometry on the water depth of the local channel*" was replaced with "*A diagram illustrating that decreasing the width of the local channel could bring about changes in the water depth of the local channel through various mechanisms. In general the phenomena before and after an arrow have the cause – effect relationship*".

[C1-25]

Page 18 line 26 should read 'could have an evident effect'

Reply: "*an*" was added between "*have*" and "*evident*".

**Response to Referee #2**

We appreciate the time and effort of the reviewer and thank the reviewer for the constructive comments. We addressed all the comments, and included corresponding changes in the revised manuscript.

In the following text, we use blue font for the reviewer comments, black font for our replies, and italics for the revisions in the manuscript.

**Comments**

[C2-1]

The paper presents improvements on the parametrization of the MOSART surface water model. State of art methods are used to update river model and inundation parametrization. The model is evaluated in the Amazon basin and several simulations were performed to evaluate the role of the DEM, river geometry parameters, and backwater effects. The subject addressed by the paper is important. With new data available for regional/global hydrologic simulations, there are several new efforts to improve hydrological models. And the documentation of new improvements/updates of models, as the MOSART, fits the goal of GMD journal. Also, the study of impact of model errors and different parametrizations are important guide future model developments. The paper is generally clear. However, it seems that most of the conclusions from paper analyses were already provided by the past modelling studies in the Amazon (e.g. Paiva et al., 2013, Getirana et al. 2012, Yamazaki et al., 2011, Beighley et al., 2009; Baugh et al., 2013). For example, the past studies already pointed for the importance backwater effects and flooding, performed sensitivity studies on the role of river geometry errors and DEM errors on amazon simulations. So I guess that it would be better to present the paper as a documentation of the improvements of a specific model (MOSART) to move toward state of art methods. And to clarify that the analyses could reproduce similar conclusions from the past studies. So, as the documentation of model parametrizations fits the GMD journal scope, I think that the paper could be published. But it needs to be reviewed clarify the actual contributions, by addressing the comments above and below.

Reply: We thank the reviewer for the positive evaluation and suggestions. We clarified the contribution mainly as the incorporation of an inundation scheme into the MOSART model. The related revisions were added in several sections: Abstract, Introduction, Methods and data, Sensitivity study, and Summary and discussion.

Our initial manuscript was not clear in the comparisons with previous studies. Following the suggestions of the reviewers, we added more discussions on comparisons between our study and former studies.

While previous studies provided the foundation for the approaches we adopted in our study and some of our results agree with those of former studies, our study has also provided some new points in terms of methodologies, model results and analyses. These are elaborated below.

1. Methods

1.1 DEM correction

We explicitly considered the spatial variability of the vegetation-caused biases in the DEM data, which alleviated biases for hydrologic modeling in the entire Amazon Basin. The DEM correction was based on a map of spatially varying vegetation heights and a land cover dataset.

In most previous studies of hydrologic modeling for the entire Amazon Basin, the DEM was lowered by a uniform height for vegetated area (Coe et al., 2008; Paiva et al., 2013a). Although spatial variability of vegetation-caused biases in DEM was also considered in previous hydrodynamic modeling studies, they were performed only in a comparatively small area of the central Amazon region (Baugh et al., 2013; Wilson et al., 2007). We generalized the approach by using land cover data and vegetation height data that have global coverage so the method can be used in the entire Amazon Basin and other regions.

1.2 Refining channel geometry

We refined the basin-wide empirical formulae for channel cross-sectional dimensions in various subregions to improve the representation of spatial variability in channel geometry (Table 1). In many former studies, the basin-wide formulae were used (Beighley et al., 2009; Coe et al., 2008; Getirana et al., 2012; Yamazaki et al., 2011).

Paiva et al. (2013a) accounted for spatial variability of channel geometry formulae and used various coefficients in their formulae for six zones of the Amazon Basin (Table 1 of Paiva et al. 2013a). But they did not compare the results of diverse subregion formulae with those of the basin-wide formulae.

 2. Sensitivity study

The sensitivity analyses of former studies primarily examined the impacts of various factors (e.g., the inundation scheme, channel geometry, Manning coefficients or backwater effects) on the total flooded area of the central Amazon region (Figs. 9 and 13 of Yamazaki et al. 2011) or the entire Amazon Basin (Fig. 10 of Paiva et al. 2013a), and streamflow and river stages of a few mainstem gauges (Figs. 13, 5a and 5b of Yamazaki et al. 2011; Fig. 10 of Paiva et al. 2013a).

In a more comprehensive manner, we examined the impacts of five factors (i.e., the inundation scheme, correcting DEM, channel geometry, Manning coefficients, and backwater effects) on modeled surface hydrology at various locations spread over the Amazon Basin, including inundation of 10 subregions

(Fig. 11), streamflow and river stages of more than 10 gauges (at both the mainstem and tributaries) (Figs. 8 and 9), and the water surface profile along the mainstem (Fig. 10).

Paiva et al. (2013a) examined the impacts of perturbing precipitation, elevation profiles or maximum soil water storage on modeled surface hydrology (Figs. 10 and 11 of Paiva et al. 2013a). We did not investigate these three factors.

Our sensitivity study yields several findings which are new or different from former studies (as follows).

2.1 Impacts of including the inundation scheme

This point was not explicitly discussed in the initial manuscript. Following the suggestions of both reviewers, in the revision this point was investigated and discussed.

Our investigation related to river stages was different from the former study. To our knowledge, only Yamazaki et al. (2011) explicitly examined the impacts of the inundation scheme on water depths at the gauge station (Fig. 5b of Yamazaki et al. 2011) and water depths along the mainstem (Fig. 7 of Yamazaki et al. 2011). They conducted three simulations: the diffusion wave simulation with the inundation scheme (FLD+Diff), the kinematic wave simulation with the inundation scheme (FLD+Kine), and the kinematic wave simulation without the inundation scheme (NoFLD). Therefore, while examining the impacts of the inundation scheme on water depths (or river stages), they used the kinematic wave river routing method, but we used the diffusion wave river routing method, which was more advanced (e.g., could represent the backwater effects) (Figs. 9 and 10).

2.2 Impacts of correcting DEM

The vegetation-caused biases in DEM were alleviated with various approaches in a few previous studies in the partial or entire Amazon Basin (Baugh et al., 2013; Coe et al., 2008; Getirana et al., 2012; Paiva et al., 2011, 2013a; Wilson et al., 2007; Yamazaki et al., 2011). To our knowledge, most of these studies did not examine and explicitly report the impacts of DEM correction on the modeled results. Only Baugh et al. (2013) showed the impacts of DEM correction on floodplain water levels and inundation in a comparatively small area in the central Amazon region (Figs. 2 and 5 of Baugh et al. 2013).

Our study examined and explicitly reported the impacts of alleviating vegetation-caused biases in DEM on modeled surface hydrology in the hydrologic modeling for the entire Amazon Basin (Figs. 8, 9, 10 and 11). These basin-wide impacts were not explicitly reported in the past.

2.3 Impacts of refining channel geometry

While examining the impacts of adjusting channel geometry on modeled surface hydrology, we used a method different from those of previous studies, where the channel widths or depths of all the channels

were perturbed by a uniform percentage (or a uniform amount) (Fig. 13 of Yamazaki et al. 2011; Fig. 10 of Paiva et al. 2013a).

We refined the basin-wide formulae of channel geometry for various subregions. The channel-geometry changes were caused by the process of refining channel cross-sections and those change ratios were different for various subregions (Table 1). We compared the results of diverse subregion formulae with those of basin-wide formulae to reveal the impacts of adjusting channel geometry on modeled surface hydrology (Figs. 8, 9, 10 and 11). Our method had more physical mechanism than the former method of perturbing channel geometry uniformly in the entire basin.

2.4 Impacts of considering backwater effects

Our model results showed the impacts of backwater effects on flood extent and river stages were more prominent than those of the previous study. To our knowledge, only Yamazaki et al. (2011) explicitly reported the impacts of backwater effects on flood extent (Fig. 9 in Yamazaki et al. 2011) and river stages (Figs. 5b and 7a in Yamazaki et al. 2011). In our study, the impacts of backwater effects on flood extent (Fig. 11) and river stages (Figs. 9 and 10) were more prominent than those of Yamazaki et al. (2011). These differences may be due to the discrepancies in channel geometry or floodplain topography between the two studies.

Our model results showed that backwater effects could advance the flood peak in the Madeira River (Fig. 8c). To our knowledge, this phenomenon has not been discussed in the previous modeling studies in the Amazon Basin.

In summary, while our modeling approach and improvements do not differ conceptually from those already explored in previous studies, we attempted to generalize various methods for application over the entire Amazon Basin, which is important as MOSART is used in global Earth System Models. We also provided more comprehensive examination of our simulations and analysis of sensitivity of the simulations to various factors, which yielded some findings that have not been discussed in former studies, or are different from those of former studies.

[C2-2]

Introduction: I feel that the main goal of this paper should be to document improvements on the MOSART model. So it is important to provide more details in the intro section.

Reply: We agree with the reviewer's assessment. We revised the Introduction to more explicitly state our objective for implementing and documenting an inundation parameterization in the MOSART model

for global application, and handling a few challenges facing the continental-scale hydrologic modeling in the Amazon Basin.

Page 2. Line 25. Which of these challenges were addressed by this paper in a novel way that was not done by the past efforts?

Reply: In the initial manuscript, the comparisons between our study and previous studies were not clear. As discussed in the reply to the first comment [C2-1], on one hand, our study was based on the important foundation of previous studies; on the other hand, our work also yielded some new points in terms of methodologies, model results and sensitivity analyses.

Page 3. Line 9. Vegetation errors from SRTM DEM were removed globally by F.E. O'Loughlin et al. 2016 RSE. Please review and discuss it in the paper.

Reply: We thank the reviewer for directing us to this study related to our work. The DEM correction for hydrologic modeling is discussed in this paragraph. O'Loughlin et al. (2016) did not conduct hydrologic modeling, so the discussion of their study was not added here, but in Section 2.4 "Vegetation-caused biases in DEM" of the revised manuscript (as follows).

*" O'Loughlin et al. (2016) estimated the vegetation-caused biases in the SRTM DEM data based on vegetation height data, canopy density data and the distribution of five climatic zones (i.e., Tropical, Arid, Temperate, Cold and Polar). They created the first global 'Bare-Earth' high resolution (3 arc-seconds) DEM from the SRTM DEM data. They compared their method with the static correction method (i.e., estimating the vegetation-caused bias as the product of vegetation height and a fixed percentage) used by Baugh et al. (2013) and this study, and noted that the static correction method was effective but moderately worse than their method. "*

Page 3. Line 22. See also analyses from Paiva et al., 2013 WRR.

Reply: Paiva et al. (2013a, WRR) analyzed the sensitivities of streamflow, water depths and flooded area to the channel width and depth (in their Fig. 10). The citation of this reference was supplemented.

Objectives. What is the new proposed contribution? If the contributions are limited to updating MOSART model with state of art methods, then I think that you should specify it in the objectives and introduce MOSART in the intro section.

Reply: As discussed in our replies to the first and second comments ([C2-1] and [C2-2]), we made our objectives more clear in the Introduction section. As a reply to a similar comment from the first reviewer, we added a new section (Section 4.1 Representing floodplain inundation) to compare the simulations with and without the inundation parameterization to document its impacts on the overall performance of MOSART. Figures 8, 9, and 10 were updated to include results for the simulation without inundation for comparison with various simulations that include the inundation parameterizations.

2.1. How the model defines what is main river network and tributary subnetwork?

Reply: In the MOSART model, each computation unit (subbasin or grid cell) has a major channel (or main channel) and a tributary subnetwork which includes tributaries within the computation unit. Please see the following figure from Li et al. (2013).

[Figure]

FIG. 1. Conceptualization of river network in MOSART. The runoff generated first enters the tributaries (surface runoff via hillslope routing and subsurface runoff without hillslope routing); it is then routed through the tributaries (here conceptualized as a single equivalent channel, as shown by the light blue dashed lines) and is finally discharged into the main channel. The value $V_h$ is the overland flow during hillslope routing, $V_t$ is the channel velocity within the tributaries, and $V_r$ is the channel velocity within the main channel.

The main channels of all the computation units constitute the main-channel network of the entire basin.

The following text was added in the first paragraph of Section 2.1 :

*"In the MOSART model, each computation unit (subbasin or grid cell) has a major channel (or main channel) and a tributary subnetwork that represents the combined equivalent transport capacity of all the tributaries within the computation unit."*

Eq.1. It seems g can be removed from equation.

Reply: "*g*" was deleted from this equation.

Continuity equation is not shown. Please show it.

Reply: The continuity equation was added (Equation (1) in the revised manuscript).

How these equations (kinematic and diffusive) are solved? Please provide details on the numerical methods. finite difference, finite volumes, implicit, explicit? Criteria for time step, spatial discretization? What is done to avoid mass errors.

Reply: The explicit finite difference method is used to solve the equations. The computation units can be grid cells or subbasins. The Courant condition is used for choosing the time step. In this Amazon application, the time step is one minute when the diffusion wave method is used. The cumulative mass error is less than 0.5 percent in these multi-year simulations.

The following text was added at the end of Section 2.1:

*"In this model, the equations are solved with the explicit finite difference method. Either square grid cells or irregular subbasins can be used as computation units. The time-step size is chosen to satisfy the Courant condition to ensure stable computation (Cunge et al., 1980)."*

2.2. It is not clear how you compute river bed elevation? Is it simply the lowest DEM pixel of the catchment? How the model accounts for the fact that SRTM DEM does not see the river bed? And the fact that the river profile is not flat?

Reply: In Fig. 1b, the elevation profile and the fraction of channel area ($A_c$) can determine the elevation of the channel bank top ($E_t$). The channel bed elevation ($E_b$) is: $E_b = E_t - d$, where $d$ is the channel depth and is estimated in Section 2.5. The channel bed could be lower than the lowest DEM pixel of the catchment because the DEM does not see the channel bed.

In the elevation profile of Fig. 1b, the longitudinal profile of the channel bed is deemed to be flat, which is different from the actual condition in the real world. This assumption may bring about some error when the flooded area is estimated.

Fig. 1b was updated in the manuscript: (1) In the amended elevation profile, the channel bed elevation was lowered; (2) The bank top elevation ($E_t$) and the channel bed elevation ($E_b$) were indicated.

The following text was added in the second paragraph of Section 2.2:

"*The channel bed elevation equals the difference of the bank top elevation and the channel depth which is estimated in Sect. 2.5. The channel bed could be lower than the lowest DEM pixel of the computation unit because the DEM does not reflect the channel bed elevation.*"

[C2-12]

2.3. How the basins are defined? What is the input data? Hydrosheds? Please make it clear.

Reply: Yes, the subbasins were extracted from the HydroSHEDS DEM. The following text was revised and moved from Section 2.4 to Section 2.3.

"*The 3 arc-seconds HydroSHEDS DEM data developed by United States Geological Survey (USGS) (http://hydrosheds.cr.usgs.gov/) was used in this study. The hydrologically conditioned HydroSHEDS DEM was used to generate the digital river network and subbasins.*"

[C2-13]

Pag. 6 Line 30. What is the criteria to define river length? How time step is defined? How these choices affect model errors (model numerical stability, mass errors, numerical dispersion, … ) ? Please clarify and discuss it.

Reply: We used comparatively coarse subbasins (the average area is 1091.7 km$^2$) due to computational costs. Each subbasin has a main channel. The main channel length varies with the subbasin size.

The time step was determined based on the Courant condition and some test simulations. The time step of one minute was used so that the simulations were stable. The cumulative mass error for the entire Amazon Basin was less than 0.5 percent in the simulations.

The following text was added in the first paragraph of Section 2.3 :

"*Relatively coarse resolution subbasins were adopted as MOSART-Inundation is intended for global earth system modeling, which is constrained by computational cost. ... ... To ensure stable computation, the time-step size was determined based on the Courant condition and numerical tests. The time step of one minute was used for all the simulations.*"

[C2-14]

2.4. Vegetation Errors. Was the corrected DEM validated ? Please justify and compare these methods to the global SRTM DEM product free of veg errors recently developed by F.E. O'Loughlin et al. 2016 RSE.

Reply: Our method was based on that of Baugh et al. (2013). Moreover, we also used a high resolution (3 arc-seconds) land cover dataset when estimating the vegetation-caused biases in the HydroSHEDS DEM data.

The discussion on the study of O'Loughlin et al. (2016) was added in Section 2.4 "Vegetation-caused biases in DEM" (as follows).

" *O'Loughlin et al.(2016) estimated the vegetation-caused biases in the SRTM DEM data based on vegetation height data, canopy density data and the distribution of five climatic zones (i.e., Tropical, Arid, Temperate, Cold and Polar). They created the first global 'Bare-Earth' high resolution (3 arc-seconds) DEM from the SRTM DEM data. They compared their method with the static correction method (i.e., estimating the vegetation-caused bias as the product of vegetation height and a fixed percentage) used by Baugh et al. (2013) and this study, and noted that the static correction method was effective but moderately worse than their method.* "

[C2-15]

2.6. Line 16. What literature was used to define Manning at 0.03 and 0.05? I feel that the parametrization of Manning needs more justification (past studies or calibration). How these choices will impact model results?

Reply: Previous studies were cited to justify our choice of the Manning coefficients. The sensitivity study part (Section 4.4) discusses the effects of the Manning coefficients on model results. Refining the Manning coefficients could improve streamflow hydrographs. The increase of the Manning coefficient could affect flood extent, streamflow and river stages in local, upstream or downstream subbasins.

The text was revised as follows:

"*Following Getirana et al. (2012), $n_{max}$ and $n_{min}$ were set as 0.05 and 0.03, respectively. In addition, a few other studies of the Amazon Basin adopted similar values around the range of 0.03 – 0.05 for the Manning coefficient (e.g., Beighley et al., 2009; Paiva et al., 2013a; Yamazaki et al., 2011).*"

[C2-16]

2.7. Line 6. Why average manning of 0.03 ? You should use the average Manning from the reference simulation or use other approach to isolate the effect of variable vs constant manning.

Reply: The uniform Manning coefficient of 0.03 is used for two reasons: (1) the uniform Manning coefficient of 0.03 was used by Yamazaki et al. (2011); (2) it is the lowest value in the range 0.03 – 0.05. The spatially varying Manning coefficients are from 0.03 to 0.05 in the control simulation. So comparing the control simulation and the simulation "n003" (which uses the uniform Manning coefficient of 0.03) can reveal the impacts of Manning coefficient increases on modeled surface hydrology.

Following the reviewer's suggestion, we conducted a new simulation using the Manning coefficient of 0.04 (the average of 0.03 and 0.05) for all the channels (abbreviated as 'n004'). We added the Nash–Sutcliffe efficiency coefficients (NSEs) of streamflow of the simulation 'n004' into Table 4.

The description of the six contrasting scenario simulations was expanded and moved to Section 4 "Sensitivity study" in the revised manuscript. The description of the simulations "n003" and "n004" was added in the fourth paragraph of Section 4 (as follows).

"*A few previous studies at the Amazon Basin used a constant Manning coefficient for all the channels (e.g., 0.04 was used by Beighley et al., 2009; and 0.03 was used by Yamazaki et al., 2011). A constant Manning coefficient of 0.03 and 0.04 was used in the fifth and sixth simulations, respectively (abbreviated as "n003" and "n004").*"

The description on the simulation comparison in Section 4.4 was revised (as follows).

'*The streamflow Nash–Sutcliffe efficiency coefficients (NSEs) of "CTL" were compared with those of "n003" and "n004" (Table 4). The NSEs of "CTL" are higher than those of "n004" at 10 of the 13*

*gauges (except Fazenda vista alegre, Itapeua and Manacapuru) and higher than those of "n003" at 12 of the 13 gauges (except Obidos). These results suggest that the spatially varying Manning coefficients are more appropriate than the uniform Manning coefficient of 0.03 or 0.04 for the simulations of this study.*

*The spatially varying Manning coefficients range from 0.03 to 0.05 and are equal to or larger than the Manning coefficient of 0.03. The spatially varying Manning coefficients result in larger flood extent than the uniform coefficient of 0.03 (Fig. 11).  '*

[C2-17]

2.7. What is optimal combination? Was any calibration performed?

Reply: The original manuscript was not clear. It meant that in the control simulation, the preferred methodologies were used at each aspect. We did not try to calibrate parameters to improve the modeled results.

The text was expanded to be more specific in Section 2.7 "Control simulation" (as follows).

*"The aforementioned factors could have important impacts on modeling surface hydrology of the Amazon Basin. We configured a control simulation (abbreviated as "CTL") using the preferred methodologies for five aspects: (1) the inundation scheme was turned on; (2) vegetation-caused biases in the DEM data were alleviated; (3) the basin-wide channel geometry formulae were refined for different subregions; (4) the Manning coefficient varied with the channel size; (5) the diffusion wave method was used to represent river flow in channels. The control simulation was run for 14 years (1994 – 2007) and the results of 13 years (1995 – 2007) were evaluated against gauged streamflow data and remotely sensed river stage and inundation data."*

[C2-18]

3.1. How the model performance compare to past modelling studies in the Amazon? Please discuss it in the manuscript.

Reply: The modeled streamflow results were compared with a few previous studies. The following text was added to the second paragraph of Section 3.1:

*"In general, the simulated streamflow results are comparable to those of a few previous studies (e.g., Getirana et al., 2012; Yamazaki et al., 2011) and slightly worse than those of Paiva et al. (2013a)."*

3.2. Equation 7. Do you use this equation to estimate a parameter for simulation? If yes, this explanation should appear o section 2. Why this approach were selected? How it compares to previous studies? How this choice impact model results? See Paiva et al., 2013 for an analyses of impact of bed elevation errors on simulations.

Reply: The relative riverbed elevations from this equation can be deemed as parameters for channel routing computation. Actually, riverbed slopes ($S_0$ in Equation (2)) are directly used in the channel routing computation in this study. This approach is the same as those of previous studies (Beighley et al., 2009; Getirana et al., 2012; Paiva et al., 2011, 2013a; Yamazaki et al., 2011).

The Equation (7) and the related descriptions was moved to the methodology part (Section 2.5).

In this study, the method for estimating the riverbed slope may be different from those of previous studies. The riverbed slopes of this study were directly derived from the DEM. Some previous studies first alleviated errors caused by water depths or vegetation heights, then used the corrected DEM to derive riverbed slopes (e.g., Paiva et al., 2011). So the method of our study has less physical mechanism and may have more uncertainties than those of some previous studies.

Paiva et al. (2013a) studied the sensitivities of streamflow, water depths and flooded area to riverbed elevations. In scenario simulations, riverbed elevations were perturbed by 3m, 1m, -1m, or -3m. In our understanding, the riverbed elevations of the entire basin were raised or lowered by a uniform value in any single simulation. So this treatment did not affect the riverbed slopes used in channel routing computation. Actually this treatment reduced or increased the channel depths, which decreased or enlarged the channel conveyance capacities. In our study, the impacts of channel cross-sectional geometry on surface hydrology were studied in a different way (Sections 2.5 "Channel geometry" and 4.3 "Refining channel geometry").

3.2. How model performance for river elevation compares to previous modelling studies in the amazon? Please discuss it in the manuscript.

Reply: The simulated river-stage results were compared to some previous studies. The following text was added to the third paragraph of Section 3.2:

" *Overall, in terms of the timing and magnitude of fluctuations, the modeled river stages of this study are comparable to those reported in some previous investigations (Coe et al., 2008; Getirana et al., 2012; Paiva et al., 2013a).* "

3.3. How model performance for flood extent compares to previous modelling studies in the amazon? Please discuss it in the manuscript.

Reply: The modeled inundation results were compared to a few previous studies. The following text was added at the end of Section 3.3:

"*Although the GIEMS data have non-negligible uncertainties, it is useful to check how our results may differ from those of previous studies using the GIEMS data as the common benchmark. Overall compared to the GIEMS data, the spatial inundation patterns of this study were slightly better than those of Getirana et al. (2012), and comparable to those of Yamazaki et al. (2011) and Paiva et al. (2013a). In terms of monthly total flooded areas, Getirana et al. (2012), Paiva et al. (2013a) and this study were comparable at the whole-basin scale, while the results from Getirana et al. (2012) and this study were closer to the GIEMS data than those of Paiva et al. (2013a) at the subregion scale.* "

4.1. How these analyses compare to previous analyses of impact of DEM and floodplains on Amazon simulations from previous modelling studies?

Reply: The vegetation-caused biases in DEM were alleviated with various approaches in a few previous modeling studies in the Amazon Basin. To our knowledge, most of those studies did not explicitly report the effects of the DEM correction on the modeled surface hydrology except Baugh et al. (2013). The following text was added to Section 4.2 "Correcting DEM" (previous Section 4.1 "Correction of DEM"):

"*The vegetation-caused biases in DEM data were alleviated with various approaches in a few previous studies modeling the surface hydrology in the Amazon Basin (Baugh et al., 2013; Coe et al., 2008; Getirana et al., 2012; Paiva et al., 2011, 2013a; Wilson et al., 2007; Yamazaki et al., 2011). Most of these studies did not examine and explicitly report the effects of the DEM correction on the modeled results. Baugh et al. (2013) demonstrated that alleviating vegetation-caused biases in DEM could improve the modeled water levels and inundation over floodplains adjacent to a 280-km reach of the central Amazon (in their Figs. 2 and 5).*"

Following the suggestion of both reviewers, in the revised manuscript we added a new section (Section 4.1 "Representing floodplain inundation") to report the impacts of using the inundation scheme on modeled surface hydrology. We also compared our methodology and results with those of a few previous studies (as follows).

*" Some previous studies also examined and reported the impacts of representing the floodplain inundation on the modeled surface hydrology in the Amazon Basin (Getirana et al., 2012; Paiva et al., 2013a; Yamazaki et al., 2011). Yamazaki et al. (2011) showed the impacts of floodplain inundation on the streamflow, water depths, and flow velocities at the Obidos gauge (in their Fig. 5) and the mainstem water surface profile (in their Fig. 7). Getirana et al. (2012) demonstrated the effects of floodplain inundation on streamflow of a few mainstem gauges (in their Fig. 16). When investigating the impacts of floodplain inundation on surface hydrology, these two studies used the kinematic wave river routing method that could not represent the important backwater effects in the Amazonia, while we used the diffusion wave river routing method that captured backwater effects. Backwater effects were also represented in the dynamic wave river routing method used by Paiva et al. (2013a) when they studied the impacts of floodplain inundation on streamflow of a few major tributary or mainstem gauges including Obidos and Manacapuru (in their Table 2 and Fig. 14). Besides streamflow, in this study we also examined and revealed the prominent impacts of floodplain inundation on the river stages near 11 major gauges or along the mainstem. "*

[C2-23]

 4.2. It is not change in channel storage capacity that changes simulation. It is changes in channel conductance capacity.

Reply: "*channel storage capacity*" was revised to be "*channel conveyance capacity*" throughout the manuscript.

[C2-24]

 4.2. How these analyses compare to previous analyses of channel geometry from previous modelling studies?

Reply: Some previous studies in the Amazon Basin (e.g., Paiva et al., 2013a; Yamazaki et al., 2011) also investigated the sensitivities of modeled surface hydrology to channel geometry. They pointed out the importance of channel geometry that motivated the analysis in our study. At the same time, the methods and results of our study had some new points: (1) channel-geometry changes were caused by the process of refining channel cross-sections and those changes were spatially varying (Table 1); (2) we examined

the effects of channel-geometry changes on modeled surface hydrology at spatially distributed locations (i.e., the 10 subregions, more than 10 tributary and mainstem gauges, and the mainstem); (3) some of our result-analyzing approaches were different from those of former studies.

The following text was added to the end of Section 4.3 "Refining channel geometry" (previous Section 4.2 "Adjustment of channel geometry"):

*" The sensitivities of modeled surface hydrology to channel geometry were also investigated by some former studies (e.g., Paiva et al., 2013a; Yamazaki et al., 2011). Yamazaki et al. (2011) perturbed the channel width or depth by a uniform percentage for all the channels and examined the effects of these channel-geometry changes on streamflow of the Obidos gauge and the flooded area over the central Amazon region (in their Fig. 13). Paiva et al. (2013a) perturbed the channel width by a uniform percentage or perturbed the channel-bottom level by a uniform height, which was equivalent to perturbing the channel depth by a uniform value, and investigated the effects of these channel-geometry changes on streamflow of the Obidos gauge, channel water depths of the Manacapuru gauge, and the total flooded area of the entire Amazon Basin (in their Fig. 10). These two studies showed the sensitivities of modeled surface hydrology to channel geometry, as well as the interactions between streamflow, water depths and inundation. They pointed out the importance of channel geometry and provided a foundation to this study. Here, channel-geometry changes were caused by the process of refining the channel cross-sections, and the changes varied spatially (Table 1). We examined the effects of channel-geometry changes on inundation of 10 subregions, streamflow of 13 gauges, river stages near 11 gauges, as well as the mainstem water surface profile. In addition, the effects of channel-geometry changes on modeled surface water dynamics were analyzed with approaches of which some were different from those of the former studies. "*

[C2-25]

4.3. I'm not sure if this analysis is conclusive. It is not possible to be sure that the differences in results are related to variable Manning or if it is because a specific value of 0.03 was chosen. This value may be different from the average value of the control simulation. I suggest the computation of the average Manning from control simulation and using this value for the new simulation.

Reply: Following the reviewer's suggestion, we conducted a new simulation "n004" which used a constant Manning roughness coefficient of 0.04 (i.e., the average of 0.03 and 0.05) for all the channels. We compared the streamflow Nash–Sutcliffe efficiency coefficients (NSEs) of three simulations ("CTL", "n004" and "n003"). In Section 4.4 "Varying Manning roughness coefficients" (previous Section 4.3 "Varying the Manning coefficients"), the second paragraph and the beginning of the third paragraph were revised as follows:

*" The streamflow Nash–Sutcliffe efficiency coefficients (NSEs) of "CTL" were compared with those of "n003" and "n004" (Table 4). The NSEs of "CTL" are higher than those of "n004" at 10 of the 13 gauges (except Fazenda vista alegre, Itapeua and Manacapuru) and higher than those of "n003" at 12 of the 13 gauges (except Obidos). These results suggest that the spatially varying Manning coefficients are more appropriate than the uniform Manning coefficient of 0.03 or 0.04 for the simulations of this study.*

*The spatially varying Manning coefficients range from 0.03 to 0.05 and are equal to or larger than the Manning coefficient of 0.03. The spatially varying Manning coefficients result in larger flood extent than the uniform coefficient of 0.03 (Fig. 11). ... ..."*

[C2-26]

4.3. How these analyses compare to previous analyses of Manning role from previous modelling studies?

Reply: A few former studies in the Amazon Basin (e.g., Paiva et al., 2013a; Yamazaki et al., 2011) also investigated the sensitivities of simulated surface hydrology to the Manning roughness coefficient. They revealed the importance of the Manning coefficient and motivated the analysis in our study. At the same time, the approaches and analyses of this study had some new points: (1) the Manning coefficient increase depended on the channel depth; (2) we examined the effects of Manning coefficient changes on modeled surface hydrology at spatially diverse locations (i.e., the 10 subregions, more than 10 tributary and mainstem gauges, and the mainstem).

The following text was added to the beginning of Section 4.4 "Varying Manning roughness coefficients" (previous Section 4.3 "Varying the Manning coefficients"):

*" A few studies for the Amazon Basin (e.g., Paiva et al., 2013a; Yamazaki et al., 2011) revealed some sensitivities of surface hydrology to the Manning coefficient. Yamazaki et al. (2011) perturbed the Manning coefficient by a uniform percentage for all the channels and examined the effects on streamflow of the Obidos gauge and the flooded area over the central Amazon region (in their Fig. 13). Using a similar approach, Paiva et al. (2013a) investigated the effects of the Manning coefficient on streamflow of the Obidos gauge, channel water depths of the Manacapuru gauge, and the total flooded area of the entire Amazon Basin (in their Fig. 10). These studies revealed that increasing the Manning coefficient could raise the river stage, expand the flooded area, and reduce and delay the flood peak. Instead of a uniform perturbation, we varied the Manning coefficient with the channel depth and examined the effects on flood extent of 10 subregions, streamflow of 13 gauges, river stages near 11 gauges, and the mainstem water surface profile. "*

[C2-27]

Figure 10. This figure is confusing. It's hard to understand the break in the profile. Please review it.

Reply: Figure 10 was replotted to avoid the break and to use the same y-axis for the entire river length.

[C2-28]

4.4. Line 20. See also analyses on the importance of backwater effects for amazon simulations from Paiva et al., 2013 WRR and Paiva et al., 2013 Hyd.Process. Please compare and discuss in the manuscript.

Reply: Paiva et al. (2013b, HP) demonstrated the important impacts of backwater effects on streamflow of the mainstem and tributaries, and discussed the important role of backwater effects in the inundation dynamics and river stages of the Amazon Basin. Paiva et al. (2013a, WRR) showed the important impacts of backwater effects on streamflow of eight mainstem or tributary gauges.

In a more comprehensive manner, we examined the impacts of backwater effects on flood extent in 10 subregions, streamflow of 13 gauges, river stages near 11 gauges, and the mainstem water surface profile.

The following text was added or revised:

"*Paiva et al. (2013b) used the dynamic wave method to represent river flow in the Solimoes River basin, which is the western upstream portion of the Amazon Basin. They discussed the important role of backwater effects in the inundation dynamics of the Amazon. In this study, we examined the impacts of backwater effects on flood extent in the 10 subregions constituting the Amazon Basin (Fig. 11), and demonstrated the spatial pattern of flood extent changes caused by backwater effects (Figs. 12j and 12k).*"

"*These backwater effects on hydrographs agree with the results of Paiva et al. (2013)*" was revised as "*These results agree with Paiva et al. (2013a, 2013b) which demonstrated the important role of the backwater effects in streamflow of the mainstem and tributaries of the Amazon Basin (Table 2 and Fig. 14 of Paiva et al., 2013a; Table 2 and Figs. 3, 4 and 9 of Paiva et al., 2013b).*"

"*In addition, to our knowledge, this phenomenon of backwater effects on the streamflow timing has not been discussed in previous modeling studies in the Amazon Basin.*"

"*In addition, the result of this study agreed with Paiva et al. (2013b), which discussed the backwater effects on river stages in the Solimoes River basin.*"

Conclusions. Line 20. Review Yamazaki et al., 2013. WRR for discussion in Catchment vs grid based simulations.

Reply: Yamazaki et al. (2013, WRR) used a special computation unit, which had characteristics of both catchment unit and grid unit. Using their computation unit could preserve the river flow pathway better than using the grid unit. Their computation units were more even than catchment units in terms of area. The citation of their study was supplemented in the revised manuscript.

[C2-30]

Conclusions: I'm not sure if there are new conclusions /findings that were not addressed by the past modelling studies in the Amazon (e.g. Paiva et al., 2013, Getirana et al. 2012, Yamazaki et al., 2011, Beighley et al., 2009; Baugh et al., 2013). The past studies already pointed for the importance backwater effects and flooding, performed Sensitivity studies on the role of river geometry errors and DEM errors on amazon simulations. It is important to recognize that the analyses from this paper only reproduced similar conclusions from the past studies. And also clarify that the new contribution from this paper is mostly on updating/improving the parametrization of an specific model, i.e. MOSART model by including improvements tested or suggested by the previous studies.

Reply: We thank the reviewer for the constructive suggestions.

In the initial manuscript, the comparisons between our work and previous studies were not clear. The manuscript was improved at this aspect during the revising procedure. As discussed in our reply to the first comment [C2-1], on one hand, our work was based on the important foundation of previous studies; at the same time, our investigation also had a few new points in terms of methodologies, simulation results and sensitivity analyses.

Following the suggestion of both reviewers, the contribution of incorporating the inundation scheme into the MOSART model was described more clearly than before in the revised manuscript.

**References**

[revised manuscript text omitted]

[5] University of California, Santa Barbara, California 93106, United States

*Correspondence to*: L. Ruby Leung (Ruby.Leung@pnnl.gov)

**Abstract**

In the Amazon Basin, floodplain inundation is a one key important component of Ssurface water dynamics and plays an important role in water, energy and carbon cycles of the Amazon Basin. The Model for Scale Adaptive River Transport (MOSART) was extended with Aa macro-scale inundation scheme was integrated with a surface water transport model which couldto represent floodplain inundation. and tThe extended model, named as "MOSART-Inundation", was applied used to simulate surface hydrology in this vast basinof the entire Amazon Basin. Previous hydrological modeling studies in the Amazon Basin identified and used some methodologies to dealaddressed with a few challenges facingin simulating surface hydrology of this basin, including uncertainties of floodplain topography and channel geometry, and the representation of river flow in mild slope reaches with mild slopes. We made efforts to addressed handleThis study further addressesd four aspects of the these challenges. First, 
[revised manuscript text omitted]

---

## Author Response (AR2)

February 17, 2017

Dr. Jeffrey Neal

Topical Editor of Geoscientific Model Development
School of Geographical Sciences
University of Bristol

Dear Dr. Neal,

We would like to thank you for the constructive comments and suggestions. We have addressed all of them and made corresponding revisions to the manuscript. Please find attached our replies to the comments, and the revised manuscript.

Thank you for your time and effort for this manuscript.

Sincerely,

L. Ruby Leung   Ph.D.

Laboratory Fellow
Atmospheric Sciences and Global Change Division
Pacific Northwest National Laboratory
Richland, Washington State, USA

**Reply to Editor**

In the following text, we use blue color for the Editor's comments, black color for the replies, and italics for the revisions in the manuscript.

**Abstract**

The abstract is very long and this section below does not say anything except that you compared the results to 13 gauges and inundation from GIEMS.

"The streamflow hydrographs were reproduced fairly well for the majority of 13 major stream gauges. The river-stage hydrographs were modeled reasonably well for the 11 subbasins containing or close to 11 of the 13 stream gauges. The inundation estimates were comparable to the GIEMS observations."

I would reduce the abstract length

Reply: The abstract has been shortened by removing the above three sentences. A few other revisions have also been made. The abstract has been reduced from 447 words to 347 words.

**Introduction**

The introduction needs a thorough proof read. I spotted numerous minor errors.

Reply: We have carefully checked the introduction and made many minor revisions. Please see the marked-up manuscript for the revisions.

**Methods**

The description of the inundation solver is very brief for a model description and it is not at all clear from the text how this actually works in conjunction with Figure 1. Rather it appears to simply list three equations (continuity, momentum and Manning's).

Reply: Section 2.2 "Macro-scale inundation scheme" has been revised and expanded to better elaborate on the inundation scheme. Please see Section 2.2 of the revised manuscript for the revisions.

Furthermore, the Courant condition is not sufficient to guarantee stability of a diffusive wave model (Hunter, N. M., M. S. Horritt, P. D. Bates, M. D. Wilson, and M. G. F. Werner (2005), An adaptive time step solution for raster-based storage cell modelling of floodplain inundation, Adv. Water Resour., 28, 975– 991, doi:10.1016/j.advwatres.2005.03.007.). GMD will publish model descriptions actually without the need for substantially new scientific insight, however the description needs to be more comprehensive. How was the stable time step computed? Usually this is a combination of the smallest computational unit length in the model and the maximum simulated depth. For a diffusive wave model you might need to additionally consider the friction and lowest slope simulated (e.g. hunter et al). The time step seems very long for a diffusive wave model where very shallow gradients are being simulated. The model clearly works so there is perhaps no reason to make a big issue of this, however the time step appears more a pragmatic compromise based on computational cost and the scheme will have problems at higher resolution that should be noted.

Reply: The original descriptions were not accurate. In Section 2.1 "MOSART model", the description has been expanded to be more comprehensive and accurate as follows.

" *The Courant-Friedrichs-Lewy (CFL) condition can be used to obtain a preliminary estimate of the time-step size (Cunge et al., 1980). In order to satisfy the CFL condition, the time-step size should be reduced with decreasing computation-unit length or increasing water depth. However, the CFL condition may not be sufficient to guarantee a stable numerical simulation (e.g., Hunter et al., 2005). In practice, the time-step size is determined through sensitivity tests to ensure numerical stability.* "

In Section 2.3 "Application in the Amazon Basin", the description was revised to be more accurate as follows.

" *To ensure stable computation, the time-step size was determined based on the Courant-Friedrichs-Lewy condition and sensitivity tests. The time step of one minute was used for all the simulations.* "

The one-minute time step is comparable to (or shorter than) those used in a few previous similar studies. For example, Yamazaki et al. (2011) and Yamazaki et al. (2012) use a time step of 20 minutes and 5 minutes, respectively. They also use the diffusive wave method in their one-dimensional river routing model for the Amazon Basin. Our time step is shorter than those of Yamazaki et al. (2011, 2012). The primary reason may be the difference in computation unit resolution. Yamazaki et al. use comparatively even computation units (i.e., unit catchments) which are around 600 square kilometers. Our computation units (i.e., subbasins) are quite uneven and the smallest computation units could be around one square kilometer.

In section 2.2 do you consider floodplain flows and if so how. Is a compound channel used or do you restrict flow to the channel only and use the floodplain for storage? If floodplain flow is allowed how was Manning's n assigned. I think my confusion here propagates from the model description.

Reply: The flows between floodplains of adjacent computation units are not considered. The flow is restricted to the main channels. The following clarification was added in the first paragraph of Section 2.2 "Macro-scale inundation scheme" :

*" The lateral flow between adjacent computation units is restricted to the main channel, namely it is assumed that there is no water exchange between the floodplains of different computation units. "*

The first paragraph of section 2.4 makes no sense to me. Specifically what is the conditioned DEM? Is it the HydroSHEDS DEM before or after you have applied the vegetation correction or something else? Are these non-negligible errors in the final data used for the modelling?

Reply: The original description was not clear. The conditioned DEM refers to the hydrologically conditioned DEM in which the flow direction is consistent with the expected flow of water over the terrain. The hydrologically conditioned DEM data were used to generate the digital river network. However, we used the vegetation corrected void-filled DEM data rather than the hydrologically conditioned DEM data to generate the elevation profiles. The following sentences have been added in the first paragraph of Section 2.4 "Vegetation-caused biases in DEM", and other minor revisions have been made.

*" In the previous section, it is mentioned that the digital river network and subbasins were derived from the hydrologically conditioned HydroSHEDS DEM. … … Therefore, the void-filled HydroSHEDS DEM, which was not altered by the conditioning process, is more appropriate for use to generate the elevation profiles. "*

The topography errors in the hydrologically conditioned HydroSHEDS DEM did not affect the modeling of inundation because the void-filled HydroSHEDS DEM was used to generate the elevation profiles for the inundation computation.

For equation 7 how is the slope found?

Reply: The riverbed slopes were estimated based on the DEM. The following explanation has been added below Equation (7).

*" The riverbed slopes were extracted from the DEM and could contain uncertainties since the DEM did not reflect the actual riverbed elevations. "*

Discussion

Page 30 "backwater effects on streamflow" section. I would not claim that the effect of backwatering on streamflow timing hasn't been discussed in previous studies. For me this is a process that happens in all subcritical flow systems (most lowland rivers) and would be expected once you decide that the backwater effect is important.

Reply: The following sentence has been removed :

*" In addition, to our knowledge, this phenomenon of backwater effects on the streamflow timing has not been discussed in previous modeling studies in the Amazon Basin. "*

The following relevant remark in Section 5 "Summary and discussion" has also been deleted :

*" …, of which the last was not reported in previous studies. "*

In the discussion point 5 do you mean increase inundation extent rather than advance inundation which would suggest an earlier arrival when including backwater. Please check for clarity.

Reply: Backwater effects could make the friction slope steeper and hence increase the flow velocity, which resulted in an earlier flow peak. The sentence was expanded to be more clear as follows:

*" …, as well as increase the flow velocity, which leads to an earlier timing of streamflow peak (e.g., Fig. 8c). "*

I'm not sure what is added by the paragraph starting "Building on previous studies (Baugh…" It seems to just say some of our conclusions agree and some do not with previous studies. I would either be specific here or just remove the paragraph. There are a few places where you provide quite general impressions of the results that would be stronger if supported by the statistics you calculated on model performance.

Reply: This paragraph was intended as a summary of the comparisons between our study and previous studies. However, the comparisons were already described in Section 4 "Sensitivity study". This paragraph has been removed.

Reply: The following sentence:

[revised manuscript text omitted]